# DiffusionGuard: A Robust Defense Against Malicious Diffusion-based Inpainting

**June Suk Choi[1], Kyungmin Lee[1], Jongheon Jeong[2],**
**Saining Xie[3], Jinwoo Shin[1], Kimin Lee[1]**
[1]KAIST, [2]Korea University, [3]NYU

## Abstract

Recent advances in diffusion models have introduced a new era of text-guided image manipulation, enabling users to create realistic edited images with simple textual prompts. However, there is significant concern about the potential misuse of these methods, especially in creating misleading or harmful content. Although recent defense strategies, which introduce imperceptible adversarial noise to induce model failure, have shown promise, they remain ineffective against more sophisticated manipulations, such as editing with a mask. In this work, we propose DiffusionGuard, a robust and effective defense method against unauthorized edits by diffusion-based image editing models, even in challenging setups. Through a detailed analysis of these models, we introduce a novel objective that generates adversarial noise targeting the early stage of the diffusion process. This approach significantly improves the efficiency and effectiveness of adversarial noises. We also introduce a mask-augmentation technique to enhance robustness against various masks during test time. Finally, we introduce a comprehensive benchmark designed to evaluate the effectiveness and robustness of methods in protecting against privacy threats in realistic scenarios. Through extensive experiments, we show that our method achieves stronger protection and improved mask robustness with lower computational costs compared to the strongest baseline. Additionally, our method exhibits superior transferability and better resilience to noise removal techniques compared to all baseline methods. Our source code is publicly available at our project page: https://choi403.github.io/diffusionguard.

## 1 Introduction

Text-to-image diffusion models trained on large-scale datasets have demonstrated impressive results in generating high-quality images from text prompts (Betker et al., 2023; Sauer et al., 2024; Saharia et al., 2022b). These models have expanded beyond simple image generation to support text-guided image editing (Wang et al., 2023; Brooks et al., 2023; Yenphraphai et al., 2024), enabling users to modify existing images with both ease and precision. For instance, Imagen Editor (Wang et al., 2023) allows users to manipulate images using masks and textual descriptions. This facilitates detailed and intuitive adjustments to specific areas of an image. Similarly, Image Sculpting (Yenphraphai et al., 2024) identifies 3D objects in photos. With this tool, users can then manipulate these objects directly, unlocking new possibilities for altering images. These approaches improve the user-friendliness of image editing tools, which significantly enhance the creative process by allowing for precise modifications based on textual input.

However, alongside these advances, there exists a significant concern regarding the potential misuse of text-guided image editing models. With their ability to create highly realistic and convincing content, image editing models can be exploited for malicious purposes such as generating fake news, spreading disinformation, and creating deceptive visual content. For example, with open-sourced text-to-image models (Rombach et al., 2023), one could easily manipulate a photo to falsely depict a celebrity being arrested, as shown in the bottom row of Fig. 1. As text-to-image models become more powerful, it is paramount to address these risks and implement safeguards to prevent misuse.

To mitigate the potential risks associated with the misuse of text-to-image diffusion models, protection methods based on adversarial noises have shown promise recently (Liang et al., 2023; Liang &

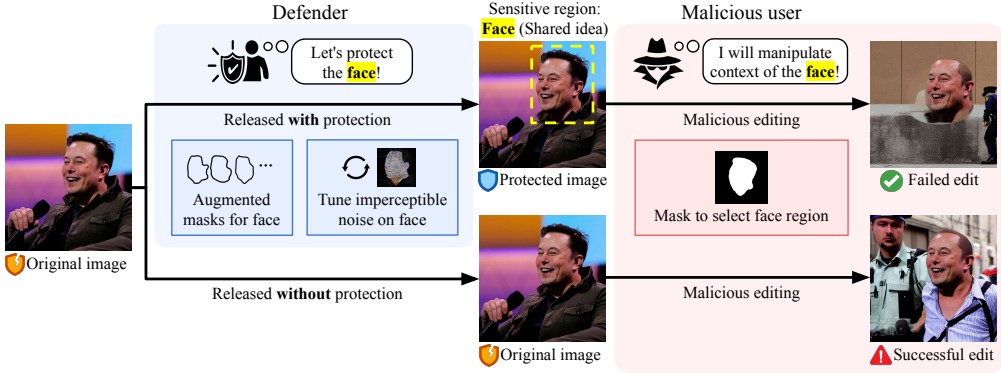

Malicious editing prompt: `"A man being arrested by the police"`

Figure 1: **Protecting against the misuse of text-to-image models using DiffusionGuard.** (Bottom row) Images without protection are vulnerable to malicious editing, such as altering the background while preserving the face to create fake images (*e.g.*, a celebrity being arrested). (Top row) DiffusionGuard protects the image by focusing on the face, a defining feature of personal identity. It disrupts diffusion models and results in failed edits when attackers attempt malicious changes.

Wu, 2023; Salman et al., 2024; Xue et al., 2024). These techniques involve adding imperceptible adversarial noise to the original image, which is designed to cause the model to fail in generating high-quality images (see Fig. 1). By publishing images with this adversarial noise, the cost and difficulty of malicious editing are significantly increased. However, current methods do not provide robust protection against real-life scenarios, such as editing with freely chosen masks by adversaries, which can bypass the protection. This issue is especially problematic as adversaries may select the smallest possible region containing sensitive identities (*e.g.*, a person's face), thereby minimizing the effectiveness of these protection methods.

**Contributions**   In this work, we introduce DiffusionGuard, a robust and effective defense method against text-guided image editing models in challenging setups, such as editing with user-selected masks. Specifically, we propose a novel objective to generate adversarial noises targeting the early stage of the diffusion process. Through our analysis, we observe that editing models tend to generate key regions within the mask during these initial diffusion steps. Therefore, by directing adversarial perturbations at the early stages, we prevent the models from maintaining the key regions, which are crucial for high-quality editing. Additionally, we propose a mask-augmentation method to find robust adversarial perturbations that are effective against mask inputs of various shapes.

For concrete evaluation, we introduce InpaintGuardBench, a challenging evaluation benchmark designed to assess defense methods against image editing models. InpaintGuardBench comprises images paired with handcrafted masks of diverse shapes and text prompts for editing, enabling a comprehensive evaluation of robustness against various misuse scenarios. We conduct human surveys and measure qualitative metrics to evaluate DiffusionGuard. Through extensive experiments, we demonstrate both qualitatively and quantitatively that DiffusionGuard is effective, and most importantly, robust against changes in mask inputs. This makes it exceptionally useful in real-life scenarios. Moreover, our method proves to be more compute-efficient and performs well even in low noise budget setups compared to existing baselines (Liang et al., 2023; Salman et al., 2024).

## 2   PRELIMINARIES

This section provides an overview of text-to-image diffusion models, emphasizing inpainting models and adversarial examples against them.

### 2.1   DIFFUSION MODELS

We consider denoising diffusion models (Sohl-Dickstein et al., 2015; Ho et al., 2020; Dhariwal & Nichol, 2021) in discrete time. Suppose $\mathbf{x} \sim p_{\text{data}}(\mathbf{x})$ represents the data distribution. A diffusion model defines a sequence of latent variables with noise scheduling functions $\alpha_t, \sigma_t$ such that the log signal-to-noise ratio $\lambda_t = \log(\alpha_t^2/\sigma_t^2)$ decreases with $t$. The forward process of a diffusion

model is described by gradually adding noise to the data $\mathbf{x}$, where the marginal distribution is given as $q(\mathbf{x}_t|\mathbf{x}) = \mathcal{N}(\mathbf{x}_t; \alpha_t\mathbf{x}, \sigma_t^2\mathbf{I})$. For sufficiently large $\lambda_T$, $\mathbf{x}_T$ becomes indistinguishable from pure Gaussian noise, and for sufficiently small $\lambda_0$, $\mathbf{x}_0$ is nearly identical to the data distribution. The reverse process starts from random noise $\mathbf{x}_T$, and sequentially denoises it to generate $\mathbf{x}_0$, which matches the training distribution.

**Text-to-Image diffusion models**  Text-to-image (T2I) diffusion models (Rombach et al., 2023; Saharia et al., 2022b; Betker et al., 2023) are a class of diffusion models specifically designed to generate images conditioned on text prompts. These models incorporate text embeddings extracted from pre-trained text encoders like T5 (Raffel et al., 2020) or CLIP (Radford et al., 2021) to guide the image generation process. Given a pair of image $\mathbf{x}$ and text $y_{\texttt{text}}$, these models commonly employ a noise prediction model $\epsilon_\theta(\mathbf{x}_t; t)$ and are trained using a noise prediction loss as follows:

$$\mathcal{L}_{\texttt{diff}}(\theta; \mathbf{x}) = \mathbb{E}_{t\sim\mathcal{U}(1,T), \epsilon\sim\mathcal{N}(\mathbf{0},\mathbf{I})}\big[\omega(\lambda_t)\|\epsilon_\theta(\mathbf{x}_t; y_{\texttt{text}}, t) - \epsilon\|_2^2\big], \tag{1}$$

where $\omega(\lambda_t)$ is a weighting function of a timestep $t$.

**Text-guided inpainting models**  In addition to T2I generation, it is of a great interest to edit a desired region of a given image with text prompts. To this end, T2I image inpainting models (Nichol et al., 2022; Saharia et al., 2022c;a) propose to fine-tune pretrained T2I diffusion models to leverage its rich generative prior. In specific, inpainting models are fine-tuned by adding conditions of source image $\mathbf{x}_{\texttt{src}}$ and binary mask $M$ that designates the region to infill to the noise prediction loss in Eq. 1. During fine-tuning, random regions of an image are masked, and the source image and a mask are concatenated to the noisy latent $\mathbf{x}_t$ as an input of the diffusion model. The training objective of diffusion-based inpainting models is given as follows:

$$\mathcal{L}_{\texttt{Inpaint}}(\theta; \mathbf{x}_{\texttt{src}}, M) = \mathbb{E}_{t\sim\mathcal{U}(1,T), \epsilon\sim\mathcal{N}(\mathbf{0},\mathbf{I})}\big[\omega(\lambda_t)\|\epsilon_\theta(\mathbf{x}_t; y_{\texttt{text}}, t, M, \mathbf{x}_{\texttt{src}}) - \epsilon\|_2^2\big]. \tag{2}$$

## 2.2 Adversarial examples against diffusion models

Adversarial examples are deliberately fabricated data to manipulate model behaviors (Szegedy et al., 2014; Biggio et al., 2013), often with malicious intent. Given a clean image $\mathbf{x}$ and a model, an adversarial example adds a perturbation $\delta$ to $\mathbf{x}$ so that $\mathbf{x} + \delta$ deceives the model. These perturbations are typically crafted to be imperceptible to human eyes, *e.g.*, via constrained optimization using $\ell_\infty$ bound $\|\delta\|_\infty \leq \eta$ for some $\eta > 0$. In this paper, we consider crafting an adversarial example for text-guided image editing models, where we aim to find a perturbation $\delta$ of source image that enforces the editing models to generate low-quality images. A line of research (Liang et al., 2023; Liang & Wu, 2023; Xue et al., 2024; Salman et al., 2024) has investigated adversarial examples of this purpose, using them as a protective measure against unauthorized image editing. These works either perturb each individual step of the denoising process to maximize the diffusion model training loss (*i.e.*, Eq. 1), or force diffusion models to generate a specifically undesirable image as follows:

$$\delta = \underset{||\delta||_\infty \leq \eta}{\arg\min} \mathbb{E}_{\epsilon\sim\mathcal{N}(\mathbf{0},\mathbf{I})}\big[\|\widehat{\mathbf{x}}(\mathbf{x}_{\texttt{src}} + \delta; \epsilon, y_{\texttt{text}}, M) - \mathbf{x}_{\texttt{target}}\|_2^2\big], \tag{3}$$

where $\widehat{\mathbf{x}}$ is a generated image given source image $\mathbf{x}_{\texttt{src}} + \delta$, prompt $y_{\texttt{text}}$, and mask $M$.

## 3 Main method

In this section, we outline DiffusionGuard, a method designed to protect images against inpainting methods in challenging scenarios (Sec. 3.1). First, based on the unique behaviors of inpainting models, we develop a novel objective to target the early stages of the reverse diffusion process (Sec. 3.2). Next, we propose a mask-augmentation method to find a robust adversarial perturbation that remains effective against mask inputs of various shapes (Sec. 3.3).

## 3.1 Problem setup

Previous protection methods (Liang et al., 2023; Liang & Wu, 2023; Xue et al., 2024) typically consider a *global* perturbation $\delta$ applied across the entire image, *i.e.*, $\mathbf{x} + \delta$, as described in Sec. 2.2. However, such methods become ineffective against diffusion inpainting models, where a binary mask

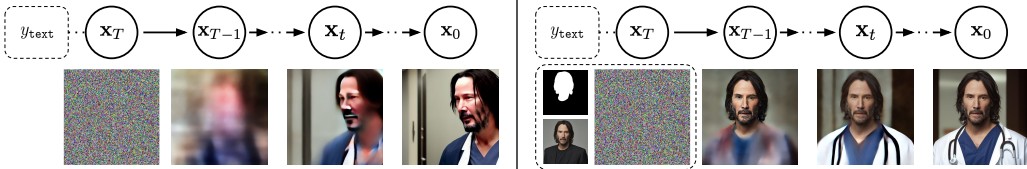

(a) Denoising process of standard diffusion models     (b) Denoising process of inpainting diffusion models

Figure 2: **Denoising diffusion process of standard and inpainting diffusion models.** (a) Standard text-to-image models typically generate only coarse features in the early stages of the denoising process. (b) In contrast, inpainting models, which are fine-tuned versions of these standard models, produce fine details (*e.g.*, face) from the very first denoising step ($T - 1$).

$M$ is additionally applied to the source image, *i.e.*, it processes *masked* source images $(\mathbf{x} + \delta) \odot M$. This realistic setup poses a unique challenge for adversarial defense, given that now only the part of adversarial noise that intersects with the mask $M$ can affect the model's behavior.

**Threat model**    We assume that a *malicious user* attempts to successfully edit an image protected by adversarial perturbations applied by a *defender*. This malicious user can freely choose the mask input $M$, and text prompt $y_{\text{text}}$ for editing. Because it is challenging to develop a defense method against any arbitrary mask, we consider a feasible yet practical scenario where both defender and malicious user share a common understanding of the *sensitive region* in the source image; typically, in a portrait, this could be the face or the body of a person, while in other contexts, it might be a specific object. We assume that the defender uses this sensitive region as a training mask $M_{\text{tr}}$ in generating adversarial noises. Meanwhile, a malicious user can utilize a different mask $M_{\text{te}}$ but based on the same conceptual sensitive region.

## 3.2   PERTURBING THE EARLY STAGES OF THE DIFFUSION PROCESS

In this section, we introduce a novel objective that specifically exploits a unique behavior we have observed in inpainting models. As shown in Fig. 2a, it is well-known that during the denoising process of diffusion models, coarse features (such as the image outline) emerge first, while fine details are generated in the later stages (Ho et al., 2020; Hertz et al., 2023). However, we have found that this pattern does not hold for inpainting models. As illustrated in Fig. 2b, these models first produce fine details (*e.g.*, facial features) even at the first denoising step.

This unique behavior likely originates from the additional inputs given during the fine-tuning process of inpainting models. Unlike standard diffusion models that only receive random noises as input, inpainting models are fine-tuned to utilize two additional inputs by modifying the input channel of the noise prediction model $\epsilon_\theta$. Specifically, these models take a binary mask $M_{\text{tr}}$, and a masked source image $\mathbf{x}_{\text{src}} \odot M_{\text{tr}}$ as inputs. Inpainting models are fine-tuned using a reconstruction loss (Eq. 2), which encourages them to *copy and paste* the unmasked region of the image, leading to the emergent behavior observed in Fig. 2b.

Inspired by the unique behavior of inpainting models, we develop a novel objective that targets the initial step of the denoising process. Suppose we have a source image $\mathbf{x}_{\text{src}}$ to protect, an inpainting model $\epsilon_\theta$, and a binary mask $M_{\text{tr}}$ which designates the part of the image to keep while rest of the image is recreated. We aim to find an adversarial perturbation $\delta$ that maximizes the $\ell_2$ norm of *the initial predicted noise* only (*e.g.*, see Fig. 3):

$$\delta = \underset{||\delta||_\infty \leq \eta}{\arg\max} \, \|\epsilon_\theta(\mathbf{x}_T; y_{\text{text}}, T, M_{\text{tr}}, \mathbf{x}_{\text{src}} + \delta)\|_2^2, \tag{4}$$

where $T$ corresponds to the initial denoising step and $\mathbf{x}_T$ is random noise. Our proposed objective focuses on targeting the early stage of the diffusion process, in contrast to prior methods that target the entire diffusion process Liang et al. (2023) or the output images Salman et al. (2024). This approach makes generating adversarial noise both efficient and effective because only one forward pass through the noise prediction model is necessary. Additionally, unlike previous methods that aim to maximize reconstruction loss (Eq. 1) or minimize the distance to an arbitrary target image (Eq. 3), we propose to increase the norm of the noise. We have found that this maximum norm objective is more effective than previous approaches (see Fig. 6a for supporting results).

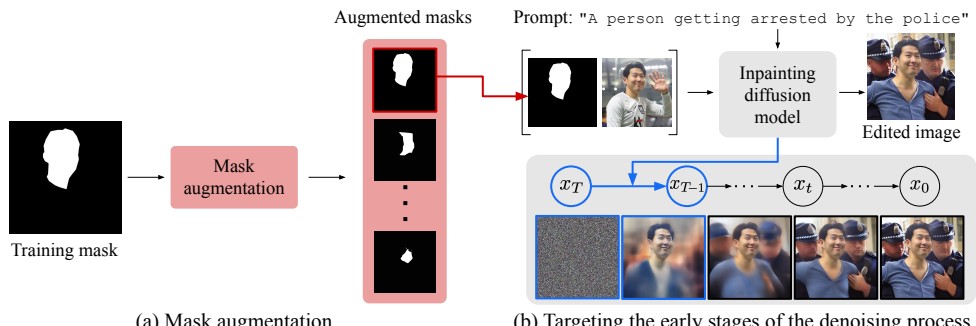

Figure 3: **Overview of DiffusionGuard.** We propose (a) mask augmentation for improving robustness, and (b) early state perturbation loss for generating effective noises.

### 3.3 MASK-ROBUST ADVERSARIAL PERTURBATION

In practice, malicious users may utilize a mask that differs from the mask $M_{\tt tr}$ that is seen during the generation of adversarial noise. Therefore, it is crucial to find robust perturbations that are effective across various mask shapes. To achieve this, we propose a mask augmentation $\mathcal{A}(\cdot)$ that generates a new binary mask with a similar shape to $M_{\tt tr}$. Specifically, we first obtain the points along the contours of $M_{\tt tr}$ using contour detection. We then adjust these points inward by a random offset to define a new contour. The area inside this new contour is filled to form the augmented mask (see Fig. 3).

To make these modifications more realistic, we apply smoothing to the random offsets. The full procedure of mask augmentation is summarized in Algorithm 1.

Using the proposed mask augmentation function $\mathcal{A}(\cdot)$, we generate a robust $\eta$-bounded adversarial perturbation $\delta$ by maximizing the following loss over the set $\mathcal{M}$ of augmented masks $\mathcal{A}(M_{\tt tr})$:

$$\delta = \underset{||\delta||_\infty \leq \eta}{\arg\max} \, \mathcal{L}_{\tt adv}(\theta; \mathbf{x} + \delta, M_{\tt tr}) = \mathbb{E}_{M \sim \mathcal{A}(M_{\tt tr})}\big[\big\|\boldsymbol{\epsilon}_\theta(\mathbf{x}_T; y_{\tt text}, T, M, \mathbf{x} + \delta)\big\|_2^2\big], \quad (5)$$

where $\mathcal{L}_{\tt adv}$ is from our adversarial loss in Eq. 4.[1] In practice, we optimize $\delta$ by stochastically sampling masks from $\mathcal{M}$ during the adversarial noise generation. Each iteration, we sample a mask $M \sim \mathcal{A}(M_{\tt tr})$ and perform a projected gradient descent (PGD) step (Madry et al., 2018) to update $\delta$:

$$\delta \leftarrow \text{Proj}_{||\delta||_\infty \leq \eta}\left(\delta - \gamma \cdot \text{sign}(\nabla_\delta \mathcal{L}_{\tt adv})\right), \quad (6)$$

where $\gamma$ is the step size and $\text{Proj}_{||\delta||_\infty \leq \eta}(\cdot)$ projects $\delta$ onto the $\ell_\infty$ ball of radius $\eta$. By iteratively updating $\delta$ using different augmented masks, we effectively minimize the expected adversarial loss over the set of masks $\mathcal{M}$. This stochastic optimization approach allows us to find a perturbation $\delta$ that is robust to various mask shapes similar to $M_{\tt tr}$.

## 4 EXPERIMENTS

### 4.1 INPAINTGUARDBENCH: INPAINTING-SPECIALIZED PROTECTION BENCHMARK

**Benchmark dataset** To thoroughly validate protection effectiveness and mask robustness, we construct a benchmark specialized for masked inpainting models. Our benchmark, named Inpaint-GuardBench, consists of 42 images, each associated with five unique masks. Out of these, one mask per image is generated using SAM (Kirillov et al., 2023), a state-of-the-art segmentation method, and the remaining four masks are handcrafted using the most common tools employed by end-users. The most commonly used tool for drawing a mask is the *circle brush*, where users select the region to keep by painting over the image. This method is employed by popular inpainting tools such as OpenAI DALL-E inpainting (Ramesh et al., 2022) and Stable Diffusion web UI,[2] the most widely used open-source GUI for diffusion models. We also incorporate simple handcrafted mask shapes

---

[1]Note that this applies to any mask-dependent adversarial loss (Salman et al., 2024), see Appendix E.3.2.
[2]https://github.com/AUTOMATIC1111/stable-diffusion-webui

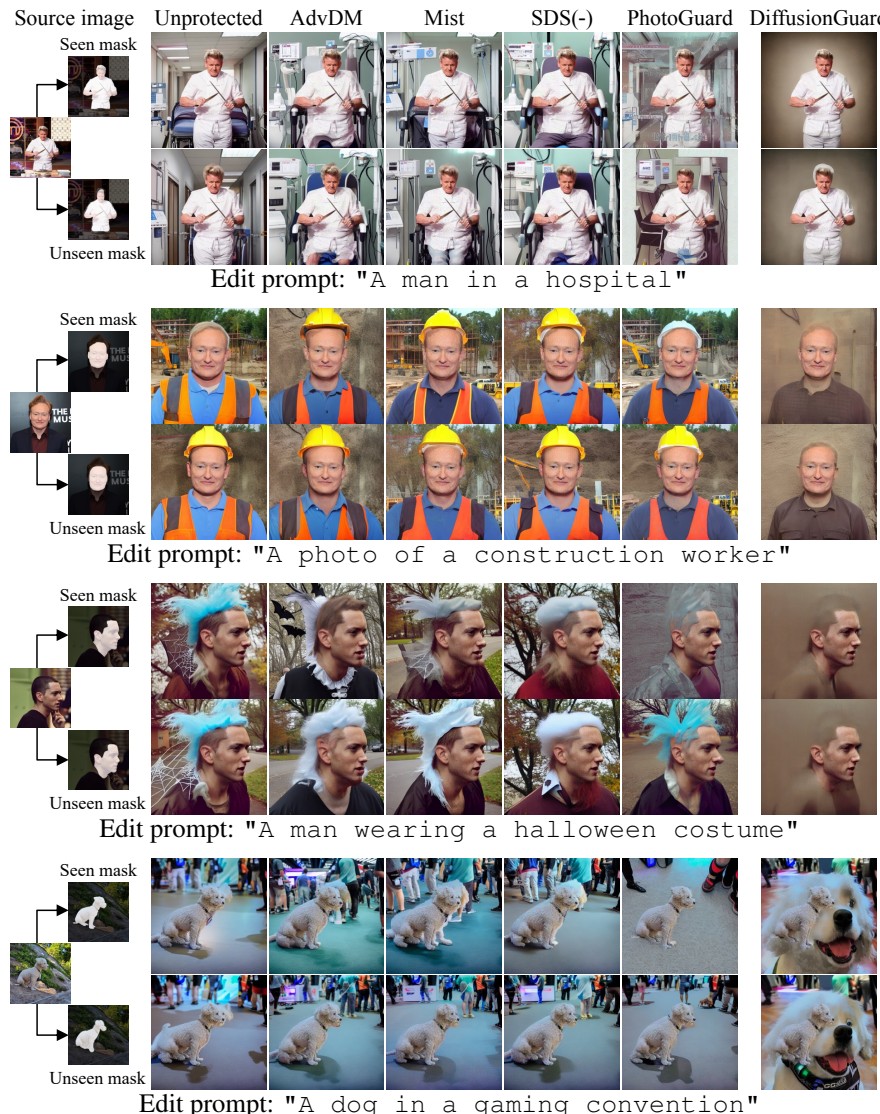

Figure 4: **Qualitative comparison between DiffusionGuard and baseline methods.** DiffusionGuard demonstrates greater protective effectiveness (*i.e.*, the ability to prevent editing diffusion models from generating images well-aligned with the edit prompt) compared to all baseline methods. Additionally, DiffusionGuard exhibits better robustness, maintaining its protective effectiveness despite changes in the mask shape.

such as rectangles and circles. Our benchmark contains 42 images, divided into three categories: 32 celebrity portraits, 5 inanimate objects, and 5 animals. We consider 10 edit prompts for each image, resulting in a total of 2,100 edit tasks (42 images, 5 masks, and 10 prompts).

**Baselines** We compare our method to various baseline methods. Our primary baseline is PhotoGuard (Salman et al., 2024), a protection method specialized for inpainting models. It adds perturbation within the mask region, optimized to cause diffusion models to generate incorrect images, as detailed in Eq. 3. We also consider AdvDM (Liang et al., 2023), a protection method that targets standard text-to-image diffusion models. This approach involves perturbing the entire image to disrupt the denoising process, aiming to maximize reconstruction loss (Eq. 1). Furthermore, we consider Mist (Liang & Wu, 2023) and SDS(-) (Xue et al., 2024), both of which build on top of AdvDM to improve the method. These three methods originally propose to add perturbation to the entire image, without considering specific mask regions. In our main experiments, we report the results obtained from the original approaches of these methods. However, we also consider variants

Table 1: **Results on InpaintGuardBench.** Our method achieves strong protection in both `Seen` and `Unseen` set, across all metrics. The lower number represents better protective effectiveness, indicating failed edits. All methods were optimized using constraint of $\|\delta\|_\infty = {}^{16}\!/_{255}$.

| Method | PSNR ↓ | CLIP Dir. Sim. ↓ | ImageReward ↓ | CLIP Sim. ↓ |
|---|---|---|---|---|
| | `Seen` (1 Mask, Train set) | | | |
| Unprotected | N/A | 24.40 | -1.365 | 30.15 |
| PhotoGuard (Salman et al., 2024) | 12.87 | 21.17 (△-3.23) | -1.537 (△-0.172) | 27.89 (△-2.26) |
| DiffusionGuard | **12.60** | **18.95 (△-5.45)** | **-1.807 (△-0.442)** | **26.55 (△-3.60)** |
| | `Unseen` (4 Masks, Test set) | | | |
| Unprotected | N/A | 24.29 | -1.315 | 30.72 |
| PhotoGuard (Salman et al., 2024) | 14.53 | 23.30 (△-0.99) | -1.357 (△-0.042) | 30.30 (△-0.42) |
| DiffusionGuard | **13.19** | **21.84 (△-2.45)** | **-1.557 (△-0.242)** | **29.05 (△-1.67)** |
| AdvDM (Liang et al., 2023) | 13.37 | 24.27 (△-0.02) | -1.361 (△-0.046) | 30.97 (△+0.25) |
| Mist (Liang & Wu, 2023) | 14.51 | 23.93 (△-0.36) | -1.307 (△+0.008) | 30.79 (△+0.07) |
| SDS(-) (Xue et al., 2024) | 14.32 | 23.85 (△-0.44) | -1.237 (△+0.078) | 30.78 (△+0.06) |

of these methods that introduce perturbations only within the mask region $M_{\mathrm{tr}}$, and the results of this modified approach are reported in Appendix E.3.1.

**Setup and evaluation metrics**   As the target model, we use Stable Diffusion Inpainting (Rombach et al., 2023), an open-sourced inpainting diffusion model. For generating adversarial noises, we use the SAM-generated mask as the training ("seen") mask. We then evaluate the effectiveness of the generated adversarial perturbations on all 5 masks, including the handcrafted 4 "unseen" masks.

For evaluation, we employ quantitative metrics to measure the fidelity of the prompt and the quality of the image. We use the following metrics:

- CLIP directional similarity (Gal et al., 2022): This measures the alignment between the deviation in the image (from the source to the edited result) and the deviation in the text (from the source caption to the edit instruction). The source caption, which describes the source image, is generated using the BLIP-Large model (Li et al., 2022).

- CLIP similarity (Radford et al., 2021): This measures the text fidelity of the edited image using the cosine similarity between their CLIP embeddings.

- ImageReward (Xu et al., 2023): A human-aligned vision-language model (Li et al., 2022) fine-tuned on a human preference dataset, assesses both the resulting image quality and text fidelity.

Additionally, we measure the PSNR between the edited results of unprotected and protected images, as done by (Salman et al., 2024), to quantify the differences in the edited result compared to the unprotected version.

## 4.2   MAIN RESULTS

We compare DiffusionGuard with the baseline methods (PhotoGuard, AdvDM, Mist, SDS(-)) on InpaintGuardBench. For all experiments, we ensure a fair comparison by running the protection methods for an equal amount of GPU time.

**Qualitative comparison**   As shown in Fig. 4, DiffusionGuard demonstrates superior protective effectiveness, causing the edit result to become nearly devoid of any recognizable content in most examples. In contrast, the baseline methods generate images that are more realistic and well-aligned with the prompts. Notably, the protected results of DiffusionGuard effectively prevent the diffusion inpainting model from 'recognizing' the object, as illustrated in the final example where a different dog is drawn in place of the original. Additionally, DiffusionGuard demonstrates its robustness against mask changes, in contrast to PhotoGuard (Salman et al., 2024), a protection method utilizing mask information, which loses the protective effectiveness even with small deviations in mask shape.

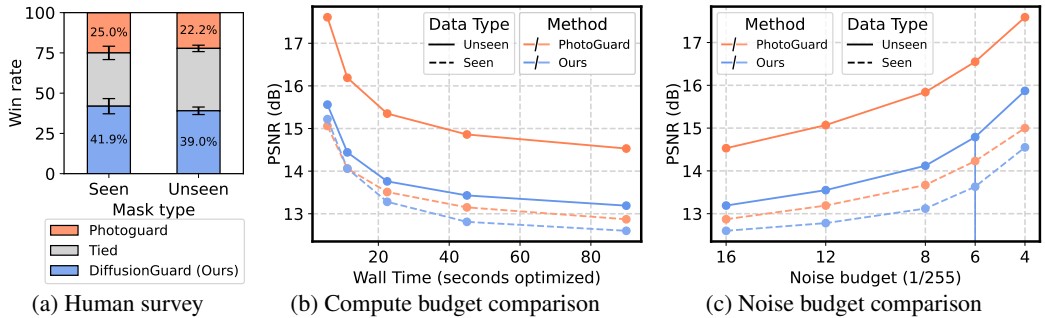

(a) Human survey          (b) Compute budget comparison          (c) Noise budget comparison

Figure 5: **(a) Human survey results.** We visualize the win rates of DiffusionGuard and Photo-Guard (Salman et al., 2024) strength grouped into `Seen` and `Unseen` groups. **(b) Comparison under limited compute budget.** PSNR values are presented per running time. **(c) Comparison under limited noise budget.** With varying noise threshold values, we measure PSNR values of each method. For (b) and (c), the protection is stronger if the PSNR value is lower.

**Quantitative results**   Table 1 reports the quantitative metrics for DiffusionGuard and the baseline methods.[3] DiffusionGuard exhibits strongest protection among all methods for both `Seen` and `Unseen` masks. Note that DiffusionGuard outperforms the baseline methods in both mask categories by a significant margin, in line with the results presented in Fig. 4.

**Human evaluation**   We also conduct a human evaluation as follows: for each edit instance, defined by a triplet of source image, mask shape, and edit instruction, we display two edit results using DiffusionGuard and PhotoGuard in random order. Human raters then select which result is better or tie (*i.e.*, the two editing results are similar) for all 2100 pairs. We ask human evaluators to make decisions based on both image quality and edit prompt fidelity simultaneously. We calculate the win rates of a protection method by counting how often its result was *not chosen* (*i.e.*, deemed worse). As shown in Fig. 5a, DiffusionGuard results in a superior win rate against PhotoGuard in both `Seen` and `Unseen` sets of masks, with approximately a 17% higher win rate gap over the baseline.

### 4.3   Comparison under resource-restricted scenarios

In this section, we compare our method against baselines in two resource-restricted scenarios. First, we evaluate each method with varying running times to compare computational efficiency. Second, we test each method under limited noise budget by setting the noise threshold $\|\delta\|_\infty$ to $4/255, 6/255, 8/255, 12/255$, and $16/255$ in order to compare them under tighter noise constraints.

**Comparison under limited compute budget**   Fig. 5b shows that DiffusionGuard is more effective than PhotoGuard when both are optimized for an equal number of steps. Specifically, our method with compute budget of 11 seconds achieves a similar PSNR of PhotoGuard at 90 seconds. Additionally, when comparing the performance gaps between the `Unseen` and `Seen` mask categories for each method, the gap is notably smaller for DiffusionGuard (blue). These results demonstrate that our method is faster, cheaper, and more effective than PhotoGuard.

**Comparison under limited noise budget**   Fig. 5c shows that DiffusionGuard consistently achieves stronger performance (*i.e.*, lower PSNR) under a tighter noise budget. In particular, our method with a noise budget of $6/255$ is similar to PhotoGuard even when PhotoGuard uses a higher budget of $16/255$. These results show that DiffusionGuard maintains high levels of protection even with reduced perturbations (*i.e.*, less visible). This makes it suitable for real-life applications where generating less detectable noise and preserving the original image quality are crucial.

### 4.4   Ablation study

We conduct a comprehensive analysis on the effects of loss functions in adversarial noise generation, mask augmentation, and the efficacy of using an inpainting-specialized method.

---

[3]Baselines optimized without mask information are omitted from `Seen` group of the table for fair comparison.

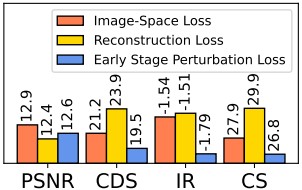 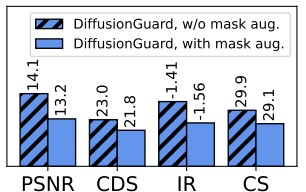 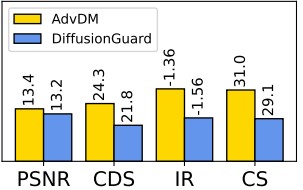

(a) Optimization loss comparison  (b) Mask augmentation comparison  (c) Against mask-free protection

Figure 6: Ablation study reporting PSNR, CLIP directional similarity (CDS), ImageReward (IR), and CLIP similarity (CS). Lower metric indicates better performance, indicating failed edits. **(a) Loss functions in adversarial noise generation.** We visualize Seen set results of the three loss functions with the same training mask. **(b) Effect of mask augmentation.** Using Unseen set, we present the effect of mask augmentation on the effectiveness of DiffusionGuard. **(c) Comparison to mask-free protection.** We visualize the protection effectiveness of using mask-free protection (AdvDM (Liang et al., 2023)) and mask-dependent protection (DiffusionGuard) on the Unseen mask set.

**Loss functions in adversarial noise generation**  To verify the effectiveness of the early stage perturbation loss (Eq. 4) in generating adversarial noises, we compare it with image-space loss (used in PhotoGuard (Salman et al., 2024)) and reconstruction loss (used in AdvDM (Liang et al., 2023)). To isolate the effects of mask augmentation, we use a single fixed mask $M_{tr}$ for generation and evaluate using the Seen set of InpaintGuardBench. As shown in Fig. 6a, our early stage perturbation loss consistently outperforms both the other losses across most metrics.

**Effect of mask augmentation**  Additionally, we evaluate how much mask augmentation improves the protection strength on the Unseen mask set by measuring the performance of DiffusionGuard with and without mask augmentation. Fig. 6b shows that mask augmentation consistently improves all metrics for the Unseen set of InpaintGuardBench, clearly demonstrating its effectiveness in enhancing mask robustness. We provide qualitative examples in Appendix E.1.

**Comparison with mask-free protection**  We compare DiffusionGuard with a mask-free protection method that applies a global perturbation over the entire image. As a baseline, we use AdvDM, a mask-free protection method based on reconstruction loss (Eq. 1). Fig. 6c presents the results on the Unseen set of InpaintGuardBench, showing that DiffusionGuard, by focusing on the mask region, provides significantly stronger protection than AdvDM across all metrics. We also remark that the noise perceptibility is much lower with DiffusionGuard. This is because the per-pixel noise threshold $\|\delta\|_\infty$ is identical between the two methods, but DiffusionGuard adds $\delta$ over a smaller, focused region, whereas in AdvDM $\delta$ occupies the entire image, making it more visible.

### 4.5 RESILIENCE AGAINST PURIFICATION METHODS

We also evaluate the robustness of DiffusionGuard and the baseline methods against noise purification tools. Specifically, we applied commonly used purification tools such as JPEG (Wallace, 1992) compression, crop-and-resize, and AdverseCleaner (Zhang, 2023), an algorithmic adversarial perturbation remover, to the protected images. As visualized in Fig. 7a, DiffusionGuard (blue) shows superior protection strength, maintaining its effectiveness even after noise removal. This demonstrates its reliable protection in real-world scenarios, where malicious users may attempt to circumvent defenses by eliminating noise. We include the full results for the purification experiments in Appendix F.

### 4.6 BLACK-BOX TRANSFER

Finally, we verify whether our method and the baselines can be black-box transferred to a different model. Our adversarial perturbations are optimized against Stable Diffusion Inpainting (SD Inpainting), which is based on the pre-trained model weights of SD 1.2. In this section, we test these perturbations on SD 2 Inpainting, which is based on the weights of SD 2.0 (Rombach et al., 2023). Since SD 2.0 was trained from scratch and does not build on SD 1.2, it represents a different model family, making this evaluation a test of black-box transfer. Fig. 7b shows the edited results for DiffusionGuard and PhotoGuard (Salman et al., 2024) using 4 different editing prompts ("A man in a hospital", "A man in a gym", "A man getting on a bus", "Photo of

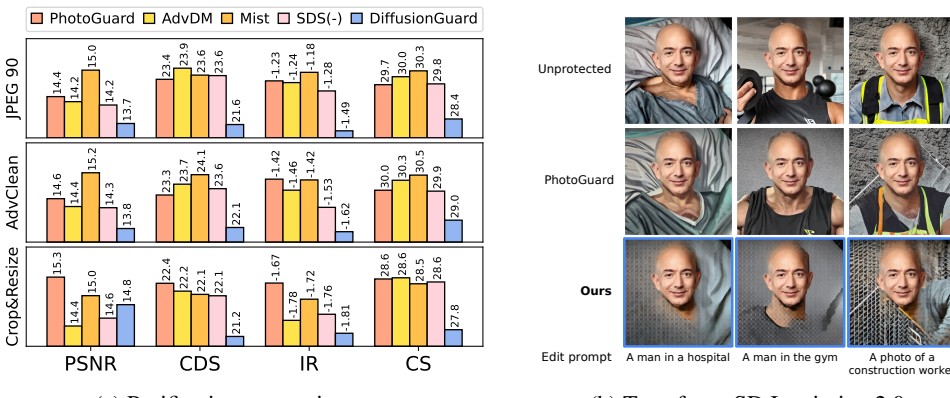

| (a) Purification comparison | (b) Transfer to SD Inpainting 2.0 |

Figure 7: Comparison of DiffusionGuard and the baseline methods after noise removal and black-box transferring. **(a) Purification.** We visualize the Seen set results for each method after removing protection using three different purification methods. Lower metric value represents stronger protection. **(b) Black-box transfer.** We visualize the protection effectiveness of DiffusionGuard and PhotoGuard (Salman et al., 2024) by demonstrating editing results after black-box transfer to SD Inpainting 2.0 on Unseen mask set, using 4 different editing prompts.

a construction worker" from left to right). As presented, DiffusionGuard maintains its effectiveness even when transferred to another model, resulting in failed edits, while PhotoGuard loses its protection and results in successful edits that are aligned with the editing prompt. We include the full evaluation results and comparisons with other baselines in Appendix G.

## 5 RELATED WORK

**Safety concerns of generative diffusion models** Generative diffusion models have raised public safety concerns, particularly regarding the potential misuse for generating realistic but harmful content. There have been various studies that seek to mitigate these concerns by developing safety measures against them. Some works focus on watermarking the images to identify manipulated images (Cui et al., 2023; Fernandez et al., 2023; Peng et al., 2023; Zhao et al., 2023). A line of studies explore ways to remove potentially harmful concepts (Gandikota et al., 2023; 2024; Heng & Soh, 2023). SPM (Lyu et al., 2024) introduced one-dimensional adapters for precise and transferable concept erasure, while MACE (Lu et al., 2024) massively scaled concept erasure to 100+ concepts. Recent methods, such as SepME (Zhao et al., 2024), have further explored efficient multi-concept erasure. Another line of research explores and identifies biases embedded in diffusion models and ways to mitigate them (Friedrich et al., 2023; Choi et al., 2024; Shen et al., 2024). These safety issues need continued focus and attention to ensure the responsible use of generative diffusion models.

**Adversarial examples against diffusion-based image editing** Several works have explored adversarial examples against diffusion-based image editing to protect images from being manipulated by these models. AdvDM (Liang et al., 2023) proposed to maximize diffusion model training loss to create adversarial examples. Xue et al. (2024) further improved AdvDM by applying score distillation sampling (Poole et al., 2023). However, most prior works did not specialize for inpainting models specifically. PhotoGuard (Salman et al., 2024) considered a defense against inpainting models but did not evaluate the protection under a challenging setup such as mask robustness.

## 6 CONCLUSION

This work presents DiffusionGuard, a robust and effective defense method against malicious diffusion-based image editing. We introduce a novel adversarial objective in order to target the vulnerability of inpainting models and disrupt the early stages of the denoising process, where key regions are generated. Furthermore, we identify key limitations in existing protection methods, such as their inability to handle various shapes of masks, and address this with mask augmentation to develop a robust protection method. By leveraging these strategies, our method achieves stronger protection and improved mask robustness with lower computational costs, when compared to several baselines. Additionally, DiffusionGuard demonstrates robustness against various purification methods, and proves effective in black-box transfer settings.

## ETHICS STATEMENT

Text-to-image diffusion models have demonstrated superior capabilities in generating and editing images based on text prompts and additional user inputs such as masks, offering significant potential across creative industries, including art, graphics, and entertainment. However, it comes with the risk of misuse, such as creating fake or deceptive visual contents. For example, malicious users can manipulate images to spread misinformation, deliberately fabricate events, or misrepresent individuals using the generated fake content.

Our work presents a defense mechanism against the malicious usage of text-to-image models, in specific, we focus on protecting sensitive areas of images from unauthorized modifications. While our method provides robust defenses, we recognize a remaining risk that advanced adversaries may find ways to circumvent protections. As such, we highlight that the necessity of ethical guidelines and legal frameworks to prevent any malicious use.

Therefore, we have made a concerted effort to evaluate and document the limitations of our approach through rigorous testing. Specifically, we conducted purification experiments (*e.g.*, Sec. 4.5 and Appendix F) to assess the resilience of our method against attempts to remove adversarial noise, as well as transfer experiments (*e.g.*, Sec. 4.6 and Appendix G) to evaluate its generalizability under a black-box transferring scenario. These experiments demonstrate the robustness of our method even when attackers employ noise-removal techniques or apply it to different models.

Nevertheless, we acknowledge the possibility of future threats and emphasize the future research on strengthening these defenses. We encourage open discussion and regulatory measures to mitigate the broader ethical implications of advancements in text-to-image models.

## REPRODUCIBILITY STATEMENT

To ensure the reproducibility of our results, we provide detailed descriptions of our methods, datasets, and evaluation criteria in Sec. 4, Appendix C, and Appendix D. Specifically, we describe how we generate adversarial perturbations, how we evaluate the protective effectiveness and the robustness of these perturbations against different test-time mask shapes, and the setup for our experiments. Additionally, we attach the executable code for adversarial perturbation generation, mask augmentation, and the evaluation suites in the supplementary materials. Full experimental details regarding setups, used inpainting models, and the evaluation metrics are in Sec. 4, Appendix C and Appendix D.

## ACKNOWLEDGMENTS

We thank Dongjun Lee, Dongyoon Hahm, Haeone Lee, and Daewon Chae for helping conduct the human survey experiments. We also appreciate Juyong Lee and Seojin Kim for providing feedback and reviewing the paper. This work was supported by Artificial Intelligence Industrial Convergence Cluster Development project funded by the Ministry of Science and ICT(MSIT, Korea)&Gwangju Metropolitan City, Artificial Intelligence Graduate School Program (KAIST) (RS-2019-II190075), the National Research Foundation of Korea (NRF) (No. RS-2024-00414822), Institute of Information & communications Technology Planning & Evaluation (IITP) (No.RS-2021-II212068, Artificial Intelligence Innovation Hub), and Development and Study of AI Technologies to Inexpensively Conform to Evolving Policy on Ethics (RS-2022-II220184, 2022-0-00184) grant funded by the Korea government (MSIT). Jongheon Jeong acknowledges support from the Institute of Information & communications Technology Planning & Evaluation (IITP) grant funded by the Korea government (MSIT) (No. RS-2019-II190079, Artificial Intelligence Graduate School Program (Korea University)), and the Culture, Sports and Tourism R&D Program through the Korea Creative Content Agency grant funded by the Ministry of Culture, Sports and Tourism (No. RS-2024-00345025, International Collaborative Research and Global Talent Development for the Development of Copyright Management and Protection Technologies for Generative AI).

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

# Appendix:

# DiffusionGuard: A Robust Defense Against Malicious Diffusion-based Image Editing

## A    MASK AUGMENTATION ALGORITHM

The full procedure of mask augmentation is summarized in Algorithm 1.

---

**Algorithm 1** Mask augmentation via contour shrinking

---

**Require:** Training mask $M_{\mathtt{tr}}$, perturbation range $\zeta$, smoothing parameter $s$, iterations $N$

1:  $M \leftarrow M_{\mathtt{train}}$
2:  **for** $i \leftarrow 1$ **to** $N$ **do**
3:      $P \leftarrow \mathtt{findContours}(M)$
4:      $P_{\mathrm{orig}} \leftarrow P$
5:      $X_{\mathrm{offset}}, Y_{\mathrm{offset}} \sim \mathcal{U}(-\zeta, \zeta) \, \forall(x_i, y_i) \in P$           ▷ Random offsets
6:      $X_{\mathrm{offset}}, Y_{\mathrm{offset}} \leftarrow \mathtt{GaussianFilter}(X_{\mathrm{offset}}, s), \mathtt{GaussianFilter}(Y_{\mathrm{offset}}, s)$ ▷ Smooth out
7:          **for** each point $(x_i, y_i) \in P$ **do**
8:              $(x_i, y_i) \leftarrow (x_i + X_{\mathrm{offset}}[i], y_i + Y_{\mathrm{offset}}[i])$
9:          **for** each point $(x_i, y_i) \in P$ **do**       ▷ Ensure $P$ stays within the original mask
10:            **if** $M_{\mathtt{tr}}[y_i, x_i] = 0$ **then**            ▷ Point is outside the mask
11:                $(x_i^{\mathrm{closest}}, y_i^{\mathrm{closest}}) \leftarrow$ closest point to $(x_i, y_i)$ on $P_{\mathrm{orig}}$
12:                $(x_i, y_i) \leftarrow (x_i^{\mathrm{closest}}, y_i^{\mathrm{closest}})$
13:        $M \leftarrow$ mask from new contour $P$
14: **return** M

---

## B    INPAINTGUARDBENCH

To assess the ability of a protection method to prevent unauthorized adversaries from editing an image in a challenging yet practical scenario as outlined in Sec. 3.1, we construct a benchmark out of various images, mask shapes, and edit prompt instructions.

### B.1    DATASET

#### B.1.1    IMAGES

To take into account realistic scenarios of privacy threat posed by inpainting models, we collect 42 images consisting of 32 images of celebrities and 10 images of non-human objects. The 32 celebrity images were collected from the web, and consist of 20 front-view images and 12 side-view images of racial and domain diversity. Out of each, 30 images are focused on faces, and 2 images focus on the body of the person. 10 non-human images were sourced from the DreamBooth (Ruiz et al., 2023) dataset. Out of them, 5 images contain animals, and 5 images include inanimate objects. We visualize all images that we have used in Fig. 8 and all masks that we have used in Fig. 9.

#### B.1.2    MASKS

In order to measure the robustness of a protection method against mask variations, we prepare 5 masks per image. For the first mask, we obtain a training mask $M_{\mathtt{tr}}$, which determines the *sensitive region* (*e.g.* face, body or object) using an automated segmentation tool (Kirillov et al., 2023) (see Sec. 3.1 for more details about the definition of the sensitive region). This mask is used for training in both DiffusionGuard and the baselines. For the remaining 4 masks, we handcraft 4 additional masks that contain the same sensitive region. The handcrafted masks are drawn using either circle brush or simple shapes such as rectangles or circles. Circle brush is a simple yet the most commonly used user interface (UI) to draw a mask, and it is used by popular inpainting tools such as DALL-E 3 ChatGPT integration (Betker et al., 2023), DALL-E 2 playground (Ramesh et al., 2022), or Stable Diffusion web UI.[4]

---

[4] https://github.com/AUTOMATIC1111/stable-diffusion-webui

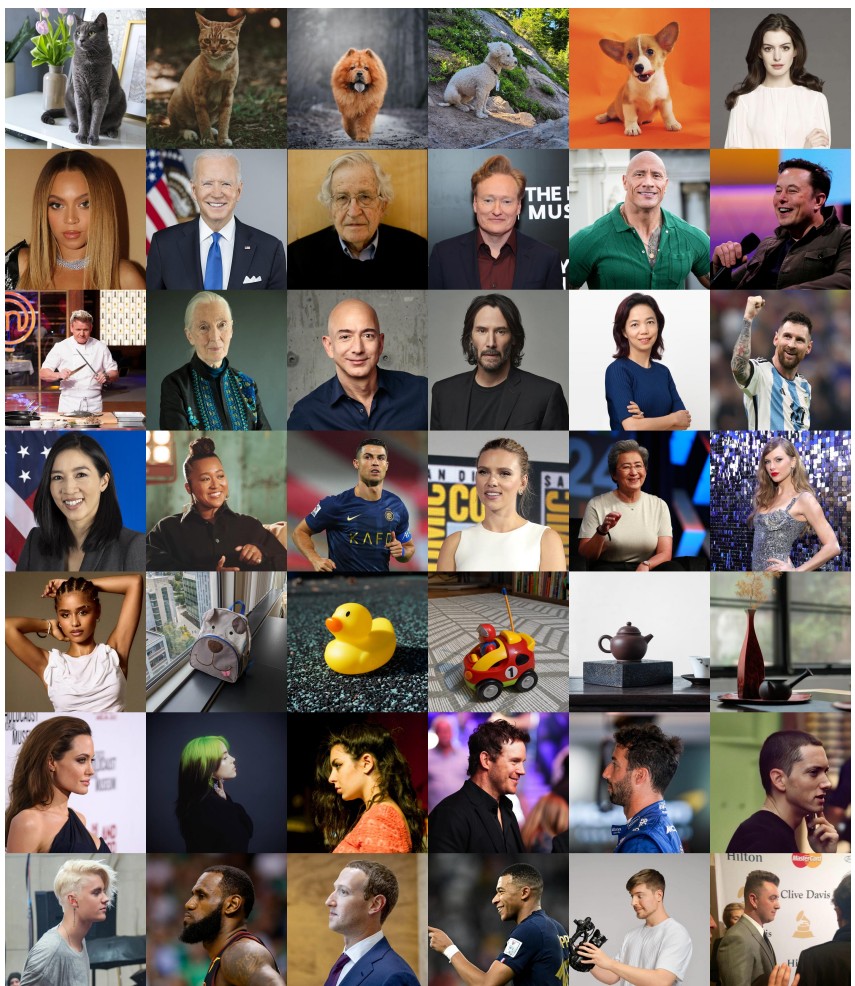

Figure 8: **All images used in InpaintGuardBench.** Best seen zoomed in.

### B.1.3 EDIT TEXT PROMPTS

Finally, we use 10 different editing text prompts in order to take into account the robustness of each protection method against different editing prompt choices. All prompts are available in Table 2 and Table 3.

| A [man/woman] in a hospital |
| --- |
| A [man/woman] riding a motorcycle |
| A [man/woman] walking in the street |
| A [man/woman] driving a car |
| A [man/woman] dancing in a club |
| A [man/woman] dressed up in halloween costume |
| A [man/woman] in the gym |
| A [man/woman] in a gaming convention |
| A photo of a construction worker |
| A [man/woman] getting on a bus |

Table 2: All prompts for portrait images.

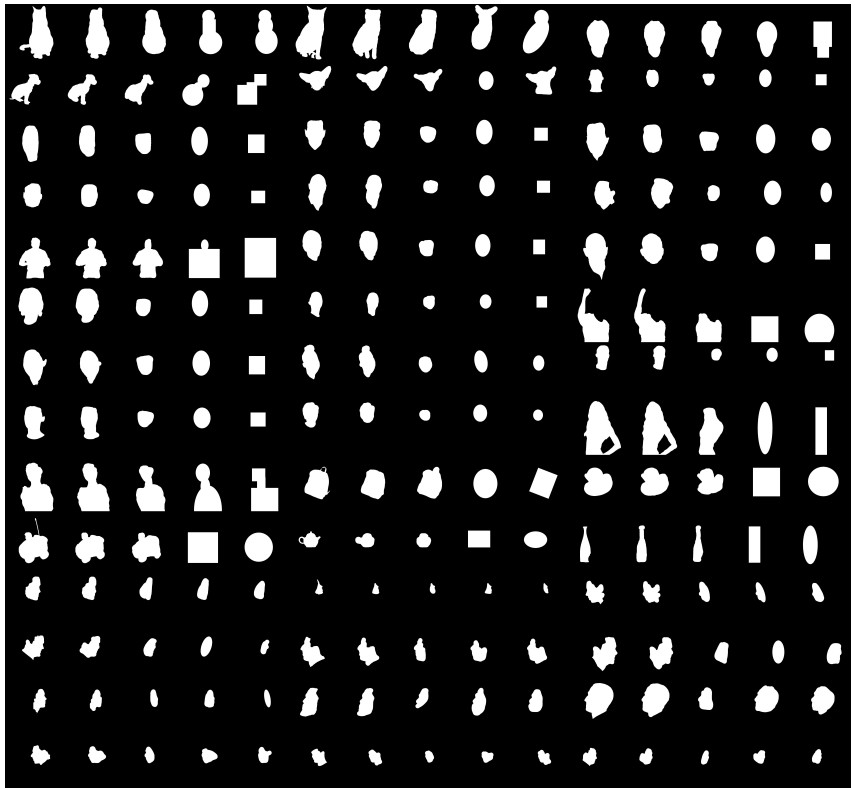

Figure 9: **All masks used in InpaintGuardBench.** Best seen zoomed in.

| A [object] in a hospital |
|---|
| A [object] on a motorcycle |
| A [object] in the street |
| A [object] in a car |
| A [object] in a club |
| A [object] in halloween |
| A [object] in the gym |
| A [object] in a gaming convention |
| A photo of [object] at a construction site |
| A [object] on a bus |

Table 3: All prompts for non-portrait images.

## C  EVALUATION DETAILS

### C.1  QUANTITATIVE METRICS

In order to quantitatively measure the protection strength of each method, we employ multiple metrics in order to measure both edit instruction fidelity and edit image quality (*i.e.* how realistic the generated image is). Because these metrics measure the degree of alignment, and our goal is to stop adversaries from obtaining desirable edits, these metrics should be lower if the protection is better.

### C.1.1  CLIP SIMILARITY

Contrastive Language-Image Pre-training (CLIP) (Radford et al., 2021) is a set of vision and text encoder trained together to align vision and text representations. To measure edit instruction fidelity, we calculate the cosine similarity between the textual description $\text{CLIP}_{\texttt{text}}(y_{\texttt{edit}})$ and the actual edited image representation $\text{CLIP}_{\texttt{image}}(\mathbf{x}_{\texttt{edit}})$, where $\mathbf{x}_{\texttt{edit}}$ is the edit result image, and CLIP is the

CLIP encoder. Higher similarity scores indicate that the edit more closely aligns with the desired instruction. This metric helps us evaluate how accurately the edits reflect the specified changes.

### C.1.2 CLIP DIRECTIONAL SIMILARITY

CLIP directional similarity (Gal et al., 2022) is a metric specifically intended to measure the performance of a text-guided image editing model. Specifically, CLIP directional similarity measures the alignment between the deviation in the text space (from the source caption to the edit instruction) and the deviation in the image space (from the source to the edited result). The source caption is a caption that describes the source image and in our case, it is obtained using BLIP-Large model (Li et al., 2022), which is an open-source captioning model. The formulation of CLIP directional similarity can be written as follows:

$$\text{CLIP directional similarity} = \frac{(\mathbf{e}_{\text{image, edit}} - \mathbf{e}_{\text{image, source}}) \cdot (\mathbf{e}_{\text{text, edit}} - \mathbf{e}_{\text{text, source}})}{\|\mathbf{e}_{\text{image, edit}} - \mathbf{e}_{\text{image, source}}\| \|\mathbf{e}_{\text{text, edit}} - \mathbf{e}_{\text{text, source}}\|}.$$

### C.1.3 IMAGEREWARD

ImageReward (Xu et al., 2023) is a human-aligned vision-language model and a reward model, which is fine-tuned on a human preference dataset. As stated and used by several works (Ye et al., 2024; Fan et al., 2023; Black et al., 2024), ImageReward is suitable for evaluating edit prompt fidelity as well as overall image quality, and shows improvement especially in terms of the ability to measure prompt-image alignment.

### C.1.4 PSNR

Peak Signal-to-Noise Ratio (PSNR) is a widely used metric to assess the similarity between two images by calculating the ratio between the maximum possible power of a signal and the power of corrupting noise that affects the quality of its representation. In our context, PSNR is used to measure the similarity between the edit result of an unprotected clean image $\mathbf{x}_{\texttt{edit}}$ and a protected image $\mathbf{x}_{\texttt{src}} + \delta$. This serves as an indicator of how much the protection alters the edited result compared to the edited result of a clean image. PSNR is defined as follows:

$$\text{PSNR}(\mathbf{x}_{\text{edit, protected}}, \mathbf{x}_{\text{edit, unprotected}}) = 20 \cdot \log_{10}\left(\frac{\text{MAX}(\mathbf{x}_{\text{edit, unprotected}})}{\sqrt{\text{MSE}(\mathbf{x}_{\text{edit, unprotected}}, \mathbf{x}_{\text{edit, protected}})}}\right)$$

where $\text{MAX}(\mathbf{x}_{\text{edit, unprotected}})$ is the maximum possible pixel value of the unprotected edited result image, and MSE is the mean squared error. Lower PSNR values indicate that the edited result of the protected image is different from the edited result of the unprotected image, indicating that the protection alters the edited result of the image.

### C.2 HUMAN SURVEY

In order to assess the edited result of the protected images perceived by human eyes, we perform a human survey with the 2,100 edit instances from InpaintGuardBench. We collected 10,500 labels using the Amazon Mechanical Turk (Crowston, 2012) platform, from 5 unique human annotators for each edit instance. An edit instance is defined by a triplet of (source image, mask, edit instruction), with fixed random seed value. We draw one edit instance from each of the two methods that are compared and present them to the rater in a shuffled order. Then, the rater is instructed to choose the method with *better* edited result in terms of the criteria, or whether it is tie. For detailed explanation about the human survey criteria, refer to Appendix C.2.1.

We created the labeling interface using HTML and CSS, which is the default method accepted by Amazon Mechanical Turk. We shuffled the 2,100 edit instances in a random order and split it into 42 batches, each with 50 edit instances. Then, we distributed the 42 batches on Amazon Mechanical Turk with 5 unique annotators assigned for each batch, resulting in 10,500 annotations in total.

When aggregating the results, we applied majority voting to the collected 5 votes per each edit instance and counted all ambiguous, tied cases as ties. Additionally, we report the 95% confidence interval of the voting results, as visualized as error bars in Fig. 5a.

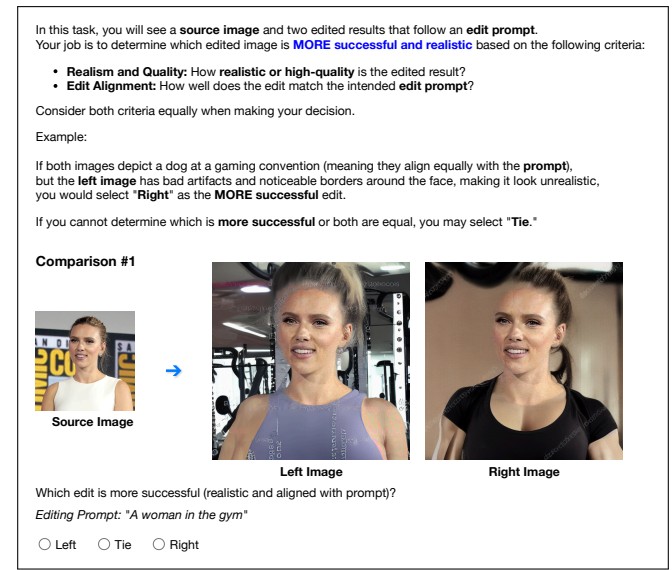

Figure 10: **The human survey labeling user interface, with the instruction given to the annotators included.**

### C.2.1 HUMAN SURVEY CRITERIA

The purpose of the protection is to prevent adversaries from achieving desired edit results that are aligned with their edit instructions, and are natural and realistic enough to spread malicious information. In order to assess this, we ask raters to choose the edit result that is *better* in terms of the following criteria and count the cases where a given method was not chosen (*i.e.* had worse results) as a winning case. The actual instruction given to the raters are visualized in Fig. 10.

- Overall image quality: Raters are instructed to assess how *natural*, *realistic*, and *high-quality* the edited image is.
- Edit prompt fidelity: Raters are instructed to assess how *aligned* the edit result image and the edit prompt are.

Due to the need for a large quantity of labels, we optimized the annotation process based on feedback from human annotators. One major issue was confusion when asked to choose a *worse* image, leading to slower labeling and less accurate annotations. As a result, we decided to ask annotators to select the *better* edit instance, counting the cases where a method was not chosen as the winner. Details on the win rate computation are provided in Appendix C.2.

### C.2.2 BASELINE FOR HUMAN SURVEY

For the baseline, we choose PhotoGuard (Salman et al., 2024) as our baseline, as (1) PhotoGuard achieves the best result overall in terms of quantitative metrics as presented in Table 1, Fig. 6, and Fig. 4, which is also visually notable, and (2) PhotoGuard proposed to target the diffusion model in a mask-dependent manner, which is more aligned with our setup outlined in Sec. 3.1, allowing a fairer comparison in contrast to other baselines, which are not necessarily mask-specific.

## D    EXPERIMENTAL DETAILS

In this section, we outline the experimental details of our experimental setup for reproducibility. We conduct all our experiments on a single NVIDIA H100 80GB HBM3 GPU. For fair comparison, we match the time taken for running PGD optimization to 90 seconds in all comparisons throughout the paper. Additionally, we fix the random seed for a reliable comparison of the edited results of different methods, and we also follow the same projected gradient descent (PGD) (Madry et al., 2018) optimization configuration proposed by each method. For the generation of adversarial perturbation, we fix the input text prompt to an empty string ("") to maximize generalization to any test-time prompt. After protection is done, each image is edited using DDIM (Song et al., 2021) sampler with 50 inference steps, following the default implementation of Stable Diffusion Inpainting (Rombach et al., 2023).

All demonstrations as well as quantitative measurements (except Fig. 2) are done after a post-processing procedure, in which the masked region from the source image is copied and pasted over the generated result, following the visualization practice of the baseline works (Liang et al., 2023; Salman et al., 2024; Xue et al., 2024). To provide insights as to what the raw generated results look like without the post-processing, we include generated results of the images protected using DiffusionGuard before and after post-processing in Fig. 11. As illustrated, the region of the edited result of the protected image inside the mask area looks noisy and blurry before post-processing (third column). Post-processing overlays the mask region of the source image onto the generated result (fourth column).

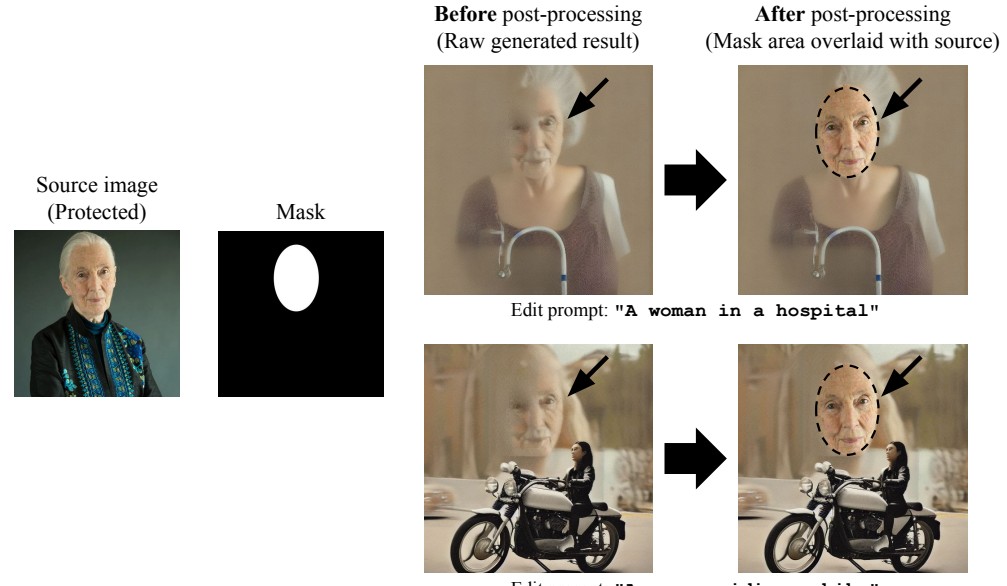

Figure 11: **Comparison of edited results of images protected using DiffusionGuard before and after visualization post-processing.** Best seen zoomed in.

## E    MORE EXPERIMENTAL RESULTS

### E.1    MORE EDITING RESULTS

In this section, we include additional editing results using DiffusionGuard. We attach the additional editing results in Fig. 30, Fig. 31, Fig. 32, Fig. 33.

### E.2    ADDITIONAL ANALYSIS UNDER SCENARIOS WITH LIMITED RESOURCES

We compare our method to the baseline method in Sec. 4 by optimizing the adversarial perturbation using under two different scenarios with limited resources. The two scenarios were using tighter

Table 4: **Results of each protection method using various noise strength threshold values.** Our method achieves strong protection, exhibiting **best** protection strength in every evaluation even under tighter noise budget, in both Seen and Unseen set, across all metrics. Noticeably, the gap between our method and the baselines grows as noise budget gets tighter, showing that our method is more robust under limited noise budget scenario. Lower metric is better, indicating failed edits.

| $|\delta|_\infty$ | Method | PSNR ↓ | CDS ↓ | IR ↓ | CS ↓ | PSNR ↓ | CDS ↓ | IR ↓ | CS ↓ |
|---|---|---|---|---|---|---|---|---|---|
| | | Seen (1 Mask, Train set) | | | | Unseen (4 Masks, Test set) | | | |
| ${}^{16}/_{255}$ | PhotoGuard | 12.87 | 21.17 | -1.537 | 27.89 | 14.53 | 23.30 | -1.357 | 30.30 |
| | AdvDM | 13.62 | 23.77 | -1.438 | 30.04 | 13.37 | 24.27 | -1.361 | 30.97 |
| | Mist | 14.22 | 24.25 | -1.368 | 30.27 | 14.51 | 23.93 | -1.307 | 30.79 |
| | SDS(-) | 14.44 | 23.42 | -1.337 | 29.89 | 14.32 | 23.85 | -1.237 | 30.78 |
| | Ours | **12.60** | **18.95** | **-1.807** | **26.55** | **13.19** | **21.84** | **-1.557** | **29.05** |
| ${}^{12}/_{255}$ | PhotoGuard | 13.19 | 21.79 | -1.499 | 28.42 | 15.07 | 23.32 | -1.365 | 30.26 |
| | AdvDM | 13.99 | 24.10 | -1.404 | 30.19 | 14.02 | 24.27 | -1.302 | 30.84 |
| | Mist | 14.87 | 23.96 | -1.374 | 30.04 | 15.05 | 24.13 | -1.281 | 30.75 |
| | SDS(-) | 15.07 | 23.85 | -1.333 | 29.92 | 14.88 | 23.96 | -1.207 | 30.64 |
| | Ours | **12.78** | **20.08** | **-1.762** | **27.15** | **13.55** | **22.13** | **-1.499** | **29.25** |
| ${}^{8}/_{255}$ | PhotoGuard | 13.67 | 22.25 | -1.475 | 28.76 | 15.84 | 23.64 | -1.337 | 30.29 |
| | AdvDM | 14.89 | 24.15 | -1.386 | 30.02 | 15.01 | 24.18 | -1.302 | 30.71 |
| | Mist | 15.84 | 24.10 | -1.383 | 30.02 | 16.01 | 23.90 | -1.313 | 30.50 |
| | SDS(-) | 15.66 | 24.09 | -1.369 | 29.87 | 15.58 | 23.92 | -1.250 | 30.58 |
| | Ours | **13.12** | **20.53** | **-1.674** | **27.43** | **14.12** | **22.65** | **-1.455** | **29.48** |
| ${}^{6}/_{255}$ | PhotoGuard | 14.23 | 23.18 | -1.432 | 29.35 | 16.55 | 23.57 | -1.341 | 30.26 |
| | AdvDM | 15.83 | 24.49 | -1.409 | 30.22 | 15.88 | 24.08 | -1.308 | 30.64 |
| | Mist | 16.54 | 24.42 | -1.367 | 30.01 | 16.75 | 24.01 | -1.321 | 30.52 |
| | SDS(-) | 16.22 | 24.14 | -1.341 | 29.82 | 16.31 | 23.92 | -1.279 | 30.45 |
| | Ours | **13.63** | **21.65** | **-1.594** | **28.05** | **14.79** | **22.93** | **-1.415** | **29.66** |
| ${}^{4}/_{255}$ | PhotoGuard | 15.00 | 23.77 | -1.397 | 29.58 | 17.59 | 23.78 | -1.315 | 30.40 |
| | AdvDM | 17.37 | 24.61 | -1.398 | 30.10 | 17.62 | 24.21 | -1.297 | 30.66 |
| | Mist | 17.54 | 24.41 | -1.392 | 29.97 | 17.98 | 24.00 | -1.327 | 30.50 |
| | SDS(-) | 17.36 | 24.09 | -1.361 | 29.94 | 17.35 | 23.95 | -1.298 | 30.53 |
| | Ours | **14.55** | **22.68** | **-1.533** | **28.68** | **15.87** | **23.16** | **-1.387** | **29.92** |

noise strength threshold (*i.e.* noise budget), and using constrained optimization time (*i.e.* compute budget).

Verifying whether a protection method is still effective under such constraints is crucial, as they are closely related to the usefulness in realistic scenarios. For instance, a stricter noise budget plays a crucial role in generating adversarial perturbations that are both subtle and imperceptible to human eyes or automated detection systems. This ensures that the noises remain less noticeable and at the same time preserve the quality and integrity of the original image, making the protection method more applicable in practical scenarios. Tighter compute budget, on the other hand, is vital for developing a cost-effective defense measure as the defender likely has to protect a large number of images in real-life scenarios, where computational resources are limited and deploying efficient defenses quickly becomes critical for maintaining security without compromising the protective effectiveness.

### E.2.1 LIMITED NOISE BUDGET SCENARIO

In this section, we conduct a comprehensive evaluation of DiffusionGuard and all baseline methods using 5 different noise strength threshold values, and present the evaluation results in Table 4. As shown in the table, DiffusionGuard (noted as "Ours" in the table, highlighted in grey color) results in the best protection strength in every case across all noise perturbation threshold and across all evaluation metrics, providing better stealthiness compared to the baseline methods. We also visualize the qualitative results for each noise budget value in Fig. 12 and Fig. 13, in which DiffusionGuard sustains its protective strength even when the noise budget gets tighter (*i.e.* lower), while other methods quickly lose protection and results in successful edits as the noise budget value decreases. This makes DiffusionGuard especially more useful in real-life applications, in which small noise

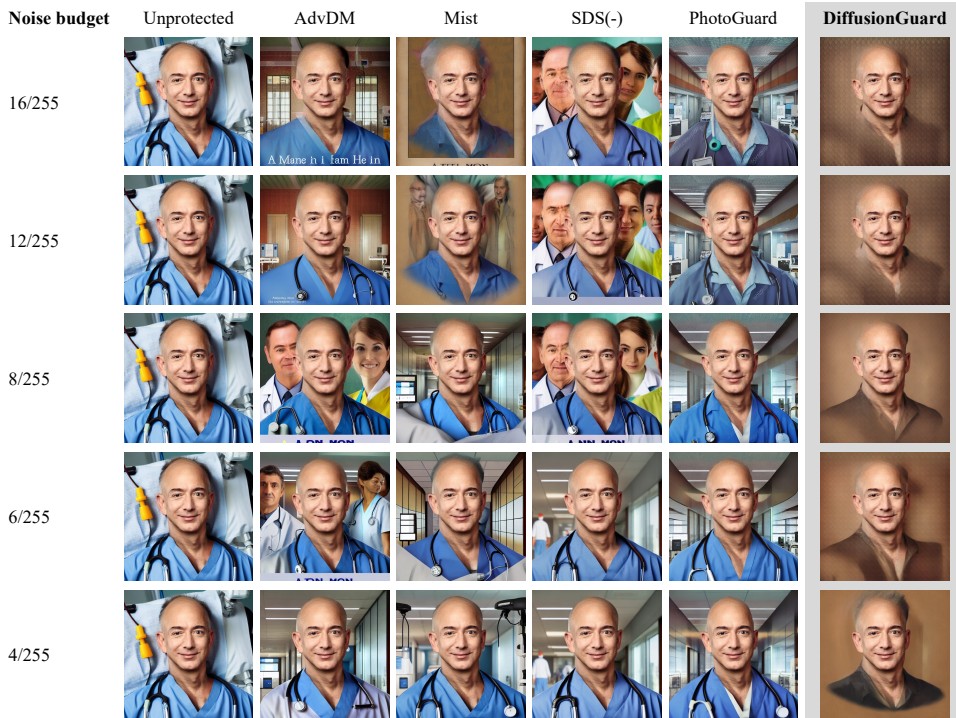

Figure 12: **Image editing results of each protection method using varying noise budget ($\|\delta\|_\infty$) values (`Seen` mask).** Editing prompt is `"A man in a hospital"`.

threshold is crucial for a less detectable image protection and for preserving the original image quality.

### E.2.2 LIMITED COMPUTE BUDGET SCENARIO

In this section, we evaluate DiffusionGuard and all baseline methods by applying projected gradient descent (PGD) optimization with varying time constraints for each method. Specifically, we optimize DiffusionGuard and all baseline methods with 5 different time durations, and present the evaluation results in Table 5. As shown in the table, DiffusionGuard ("Ours" in the table, highlighted in grey color) results in the best or the second best protection effectiveness in most cases across all most PGD optimization durations and across all evaluation metrics, exhibiting better resource efficiency compared to the baseline methods. This makes DiffusionGuard more useful in practical scenarios where multiple images need to be protected using a limited amount of computational resources.

We also visualize both limited compute budget and limited noise budget scenario measured in CLIP directional similarity in Fig. 14. In this figure, the CLIP directional similarity metric is shown against both the wall time and the noise budget. Notably, the protective effectiveness of DiffusionGuard in the `Unseen` set is similar to that of PhotoGuard in the `Seen`, which underlines the effectiveness of mask augmentation in generalizing to unseen mask inputs, resulting in a more mask-robust protection.

### E.3 ADDITIONAL ANALYSIS ON MASK REGION

In this section, we conduct additional analysis to further explore the behavior and the protective effectiveness of the methods related to mask regions selected during generation of the adversarial perturbation.

### E.3.1 USING DIFFUSIONGUARD AND BASELINE METHODS WITH A FIXED MASK

We evaluate whether the loss function of DiffusionGuard provides better protective effectiveness compared to the baseline methods when ruling out the influences of the mask, *i.e.* by using the same, fixed mask during optimization. Specifically, we fix the mask used for optimization to $M_{\tt tr}$ for all

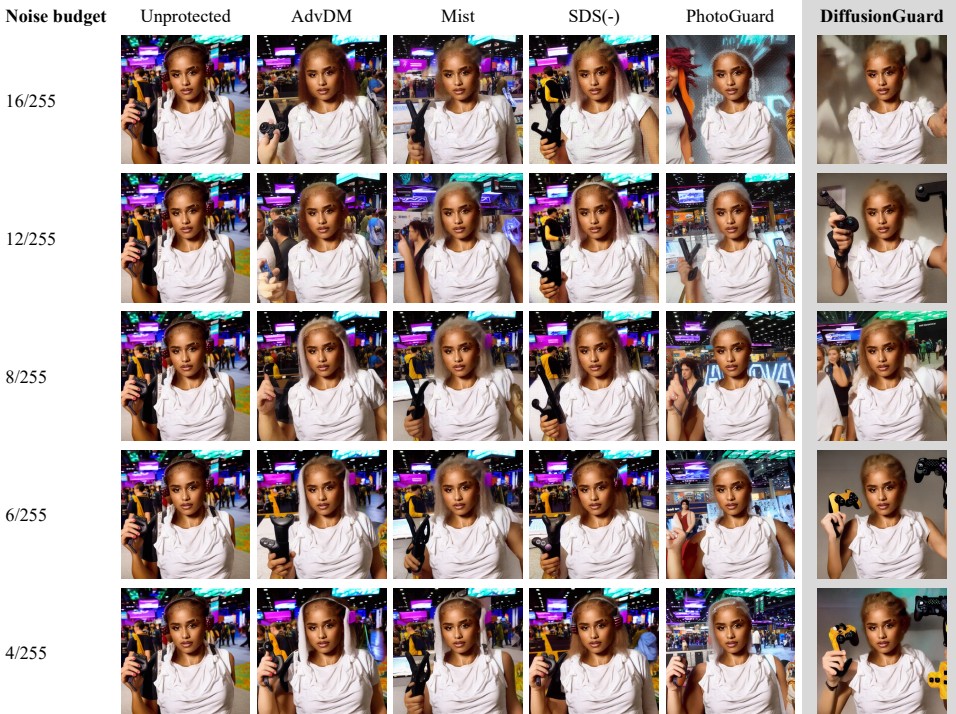

Figure 13: **Image editing results of each protection method using varying noise budget ($\|\delta\|_\infty$) values (`Seen` mask).** Editing prompt is `"A woman in a gaming convention"`.

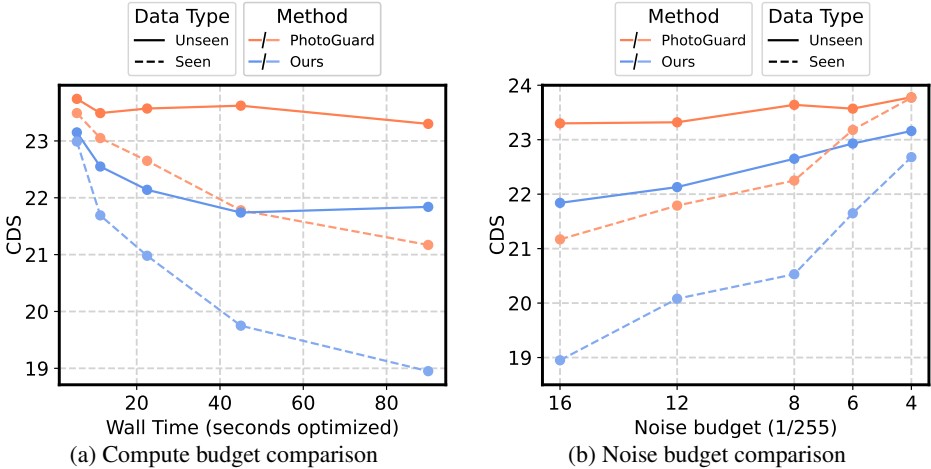

(a) Compute budget comparison

(b) Noise budget comparison

Figure 14: **(a) Comparison under limited compute budget, measured in CLIP directional similarity (CDS).** CDS values are presented per running time. **(b) Comparison under limited noise budget, measured in CLIP directional similarity (CDS).** With varying noise threshold values, we measure CDS values of each method. The protection is stronger if the CDS value is lower.

methods including DiffusionGuard and all baseline methods, and assess the performance on the `Seen` mask set (which only consists of $M_{\texttt{tr}}$).

Baseline methods like AdvDM (Liang et al., 2023), and methods that build on top of AdvDM such as Mist (Liang & Wu, 2023) or SDS(-) (Xue et al., 2024) originally propose to add a perturbation over the entire image, without considering specific mask regions, as discussed in Sec. 4. For this analysis, however, we adapt these methods to apply perturbations only within the mask region $M_{\texttt{tr}}$. Similarly, we modify DiffusionGuard by removing the mask augmentation component and fixing

Table 5: **Results of each protection method under various compute budget values.** The compute budget is defined by the time limit of PGD optimization, given in seconds. Our method achieves strong protection, exhibiting **best** or second-to-best protection strength in most evaluations even under tighter compute budget, in both `Seen` and `Unseen` set. All methods were trained using constraint of $|\delta|_\infty = 16/255$. Lower metric is better, indicating failed edits.

| Compute budget | Method | PSNR↓ | CDS↓ | IR↓ | CS↓ | PSNR↓ | CDS↓ | IR↓ | CS↓ |
|---|---|---|---|---|---|---|---|---|---|
| | | `Seen` (1 Mask, Train set) | | | | `Unseen` (4 Masks, Test set) | | | |
| 90 seconds | PhotoGuard | 12.87 | 21.17 | -1.537 | 27.89 | 14.53 | 23.30 | -1.357 | 30.30 |
| | AdvDM | 13.62 | 23.77 | -1.438 | 30.04 | 13.37 | 24.27 | -1.361 | 30.97 |
| | Mist | 14.22 | 24.25 | -1.368 | 30.27 | 14.51 | 23.93 | -1.307 | 30.79 |
| | SDS(-) | 14.44 | 23.42 | -1.337 | 29.89 | 14.32 | 23.85 | -1.237 | 30.78 |
| | Ours | **12.60** | **18.95** | **-1.807** | **26.55** | **13.19** | **21.84** | **-1.557** | **29.05** |
| 45 seconds | PhotoGuard | 13.15 | 21.78 | -1.487 | 28.50 | 14.86 | 23.62 | -1.325 | 30.43 |
| | AdvDM | 13.59 | 23.63 | -1.483 | 30.09 | 13.47 | 24.29 | -1.357 | 30.90 |
| | Mist | 14.36 | 23.94 | -1.417 | 30.09 | 14.46 | 23.93 | -1.291 | 30.71 |
| | SDS(-) | 14.46 | 23.80 | -1.336 | 30.04 | 14.31 | 24.02 | -1.240 | 30.83 |
| | Ours | **12.81** | **19.75** | **-1.763** | **27.06** | **13.43** | **21.74** | **-1.532** | **29.05** |
| 23 seconds | PhotoGuard | 13.51 | 22.65 | -1.448 | 29.02 | 15.35 | 23.57 | -1.324 | 30.39 |
| | AdvDM | 13.84 | 23.56 | -1.455 | 29.91 | **13.70** | 24.00 | -1.369 | 30.69 |
| | Mist | 14.18 | 23.82 | -1.462 | 30.04 | 14.33 | 24.09 | -1.306 | 30.70 |
| | SDS(-) | 14.44 | 23.37 | -1.403 | 29.90 | 14.26 | 24.02 | -1.267 | 30.79 |
| | Ours | **13.28** | **20.98** | **-1.654** | **27.73** | 13.76 | **22.14** | **-1.496** | **29.24** |
| 11 seconds | PhotoGuard | **14.06** | 23.05 | -1.446 | 29.07 | 16.19 | 23.49 | -1.331 | 30.23 |
| | AdvDM | 14.24 | 23.90 | -1.398 | 30.20 | 14.25 | 24.04 | -1.319 | 30.68 |
| | Mist | 14.23 | 23.58 | -1.395 | 29.79 | 14.32 | 23.97 | -1.300 | 30.60 |
| | SDS(-) | 14.43 | 23.53 | -1.373 | 29.97 | **14.24** | 23.93 | -1.288 | 30.73 |
| | Ours | **14.06** | **21.69** | **-1.564** | **28.28** | 14.44 | **22.55** | **-1.451** | **29.49** |
| 6 seconds | PhotoGuard | 15.06 | 23.49 | -1.391 | 29.36 | 17.61 | 23.74 | -1.339 | 30.33 |
| | AdvDM | 15.57 | 24.22 | -1.373 | 30.04 | 15.59 | 23.99 | -1.329 | 30.60 |
| | Mist | **14.19** | 23.73 | -1.407 | 29.83 | 14.37 | 23.95 | -1.303 | 30.54 |
| | SDS(-) | 14.37 | 23.54 | -1.393 | 29.92 | **14.18** | 24.08 | -1.306 | 30.72 |
| | Ours | 15.22 | **22.99** | **-1.509** | **29.04** | 15.56 | **23.15** | **-1.392** | **29.95** |

the mask to $M_{tr}$, as done for the baseline methods. This ensures a fair comparison by isolating the impact of the loss function after eliminating the mask factor. Note that no modifications were made to PhotoGuard (Salman et al., 2024), as it already applies perturbations exclusively to the mask region $M_{tr}$.

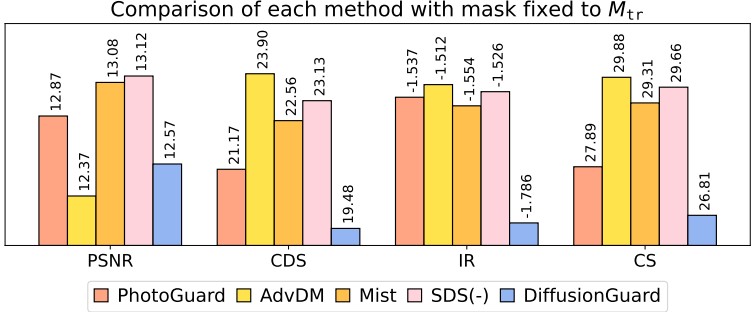

Figure 15: **Evaluation results on the `Seen` mask set after generating adversarial perturbation using each method with $M_{\mathbf{tr}}$ mask.** Note that `Seen` mask set consists only of $M_{tr}$ mask. We report PSNR, CLIP directional similarity (CDS), ImageReward (IR), and CLIP similarity (CS). Lower metrics indicate failed edits, representing better protection effectiveness of the defense method.

The evaluation results are visualized in Fig. 15. As shown in the figure, DiffusionGuard achieves significantly better protective effectiveness compared to the baseline methods across most metrics, verifying that even when the mask factor is isolated, DiffusionGuard exhibits the most effective protection strength.

### E.3.2 USING BASELINE METHODS IN COMBINATION WITH MASK AUGMENTATION

In this section, we evaluate the performance of each baseline method when combined with **mask augmentation**, a component of DiffusionGuard introduced in Sec. 3. As discussed in Appendix E.3.1, baseline methods can be adapted to use different masks. For AdvDM, Mist, and SDS(-), we modified them to add adversarial perturbations only within the mask region given by $M_{\tt tr}$. The purpose of this experiment was to analyze the effects of the loss function exclusively, after ruling out the mask-related factors. This section extends the previous analysis by applying the same mask augmentation algorithm from DiffusionGuard during the optimization process of all methods, while preserving their original loss functions.

As illustrated in Fig. 16, DiffusionGuard demonstrates superior performance compared to the baseline methods when the baselines are used in combination with mask augmentation. These experimental results, combined with those presented in Appendix E.3.1 (where the mask is fixed to $M_{\tt tr}$ to isolate the effects of the loss function), verify that (1) the early-stage perturbation loss proposed by DiffusionGuard provides the strongest protection, and (2) both components of DiffusionGuard work synergistically to provide effective protection against diffusion-based malicious image editing. We note that no changes were made to DiffusionGuard in this experiment, as it already incorporates mask augmentation.

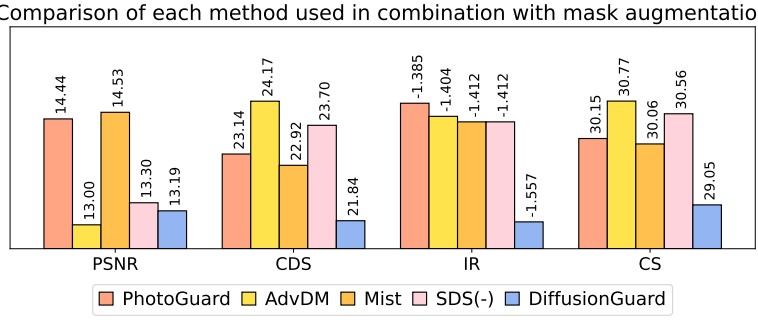

Figure 16: **Evaluation results on the `Unseen` mask set after generating adversarial perturbation using each method in combination with mask augmentation.** We report PSNR, CLIP directional similarity (CDS), ImageReward (IR), and CLIP similarity (CS). Lower metrics indicate failed edits, representing better protection effectiveness of the defense method.

### E.3.3 DIFFUSIONGUARD WITH AND WITHOUT MASK AUGMENTATION

In this section, we report a detailed analysis of the effect of mask augmentation on the performance of DiffusionGuard, specifically the performance of DiffusionGuard with and without mask augmentation in both `Seen` and `Unseen` sets of InpaintGuardBench. The results are illustrated in Fig. 17. As visualized, while mask augmentation does not have a notable impact or only marginally improves the performance of the protection in the case of `Seen` masks, it significantly improves the protection in the case of `Unseen` masks.

Comparison of DiffusionGuard with and without mask augmentation

**Figure 17: Evaluation results of DiffusionGuard and DiffusionGuard after removing mask augmentation.** We evaluate using the `Seen` set (left) and the `Unseen` set (right) of InpaintGuard-Bench. We report PSNR, CLIP directional similarity (CDS), ImageReward (IR), and CLIP similarity (CS). Lower metric indicates failed edits, representing better protection effectiveness of the defense method.

Table 6: **Editing results of each protection method after applying adversarial noise purification with various algorithms.** Our method achieves strong protection, exhibiting **best** or second-to-best protection strength in most evaluations even after applying noise purification, in both `Seen` and `Unseen` set, across all metrics. All methods were optimized using the constraint of $|\delta|_\infty = {}^{16}/_{255}$. Lower metric is better, indicating failed edits.

| Puri. | Method | PSNR ↓ | CDS ↓ | IR ↓ | CS ↓ | PSNR ↓ | CDS ↓ | IR ↓ | CS ↓ |
|---|---|---|---|---|---|---|---|---|---|
| | | | `Seen` (1 Mask, Train set) | | | | `Unseen` (4 Masks, Test set) | | |
| Adv-
Clean | PhotoGuard | 14.57 | 23.33 | -1.417 | 29.98 | 15.73 | 23.79 | -1.355 | 30.59 |
| | AdvDM | 14.37 | 23.74 | -1.460 | 30.31 | **14.34** | 24.12 | -1.354 | 31.04 |
| | Mist | 15.23 | 24.10 | -1.420 | 30.48 | 15.44 | 23.75 | -1.332 | 30.79 |
| | SDS(-) | 14.34 | 23.57 | -1.527 | 29.95 | 14.68 | 24.02 | -1.353 | 30.73 |
| | Ours | **13.81** | **22.10** | **-1.622** | **28.99** | 14.55 | **22.84** | **-1.451** | **29.94** |
| JPEG
(90) | PhotoGuard | 14.36 | 23.44 | -1.227 | 29.74 | 15.69 | 23.43 | -1.177 | 30.47 |
| | AdvDM | 14.24 | 23.89 | -1.236 | 30.04 | **14.18** | 23.89 | -1.168 | 30.80 |
| | Mist | 14.99 | 23.61 | -1.182 | 30.26 | 15.29 | 23.64 | -1.137 | 30.85 |
| | SDS(-) | 14.16 | 23.59 | -1.280 | 29.77 | 14.67 | 23.77 | -1.136 | 30.71 |
| | Ours | **13.72** | **21.61** | **-1.490** | **28.45** | 14.49 | **22.85** | **-1.279** | **29.84** |
| JPEG
(80) | PhotoGuard | 15.21 | 23.76 | -1.152 | 30.18 | 16.28 | 23.64 | -1.127 | 30.75 |
| | AdvDM | **14.64** | 23.90 | -1.154 | 30.20 | **14.68** | 23.70 | -1.143 | 30.77 |
| | Mist | 15.46 | 23.81 | -1.149 | 30.46 | 15.65 | 23.52 | -1.109 | 30.86 |
| | SDS(-) | 14.75 | 23.69 | -1.182 | 30.06 | 15.29 | 23.68 | -1.109 | 30.74 |
| | Ours | 14.68 | **22.74** | **-1.292** | **29.28** | 15.25 | **23.18** | **-1.171** | **30.33** |
| JPEG
(70) | PhotoGuard | 15.54 | 23.86 | -1.091 | 30.35 | 16.59 | 23.64 | -1.098 | 30.83 |
| | AdvDM | **15.07** | 23.62 | -1.157 | 30.12 | **15.07** | 23.70 | -1.104 | 30.86 |
| | Mist | 15.64 | 23.55 | -1.085 | 30.39 | 15.89 | **23.44** | -1.100 | 30.84 |
| | SDS(-) | 15.21 | 23.96 | -1.125 | 30.36 | 15.62 | 23.67 | -1.080 | 30.86 |
| | Ours | 15.27 | **23.45** | **-1.198** | **29.93** | 15.83 | 23.50 | **-1.133** | **30.66** |
| JPEG
(65) | PhotoGuard | 15.66 | 23.92 | -1.057 | 30.30 | 16.56 | 23.64 | -1.091 | 30.88 |
| | AdvDM | 15.33 | 23.57 | -1.087 | 30.11 | **15.21** | 23.63 | -1.094 | 30.81 |
| | Mist | 15.75 | 23.79 | -1.073 | 30.48 | 15.95 | **23.50** | -1.089 | 30.83 |
| | SDS(-) | **15.23** | 23.69 | -1.104 | 30.13 | 15.73 | 23.59 | -1.079 | 30.82 |
| | Ours | 15.51 | **23.51** | **-1.169** | **29.91** | 16.01 | 23.53 | **-1.099** | **30.69** |
| Crop&
Resize | PhotoGuard | 15.29 | 22.39 | -1.674 | 28.60 | 15.66 | 22.55 | -1.629 | 29.63 |
| | AdvDM | **14.42** | 22.22 | -1.778 | 28.62 | **14.07** | 22.68 | **-1.731** | 29.83 |
| | Mist | 14.96 | 22.15 | -1.719 | 28.52 | 15.00 | 22.12 | -1.655 | 29.48 |
| | SDS(-) | 14.56 | 22.12 | -1.761 | 28.55 | 14.64 | 22.41 | -1.689 | 29.74 |
| | Ours | 14.80 | **21.24** | **-1.814** | **27.75** | 15.11 | **22.01** | -1.707 | **29.19** |

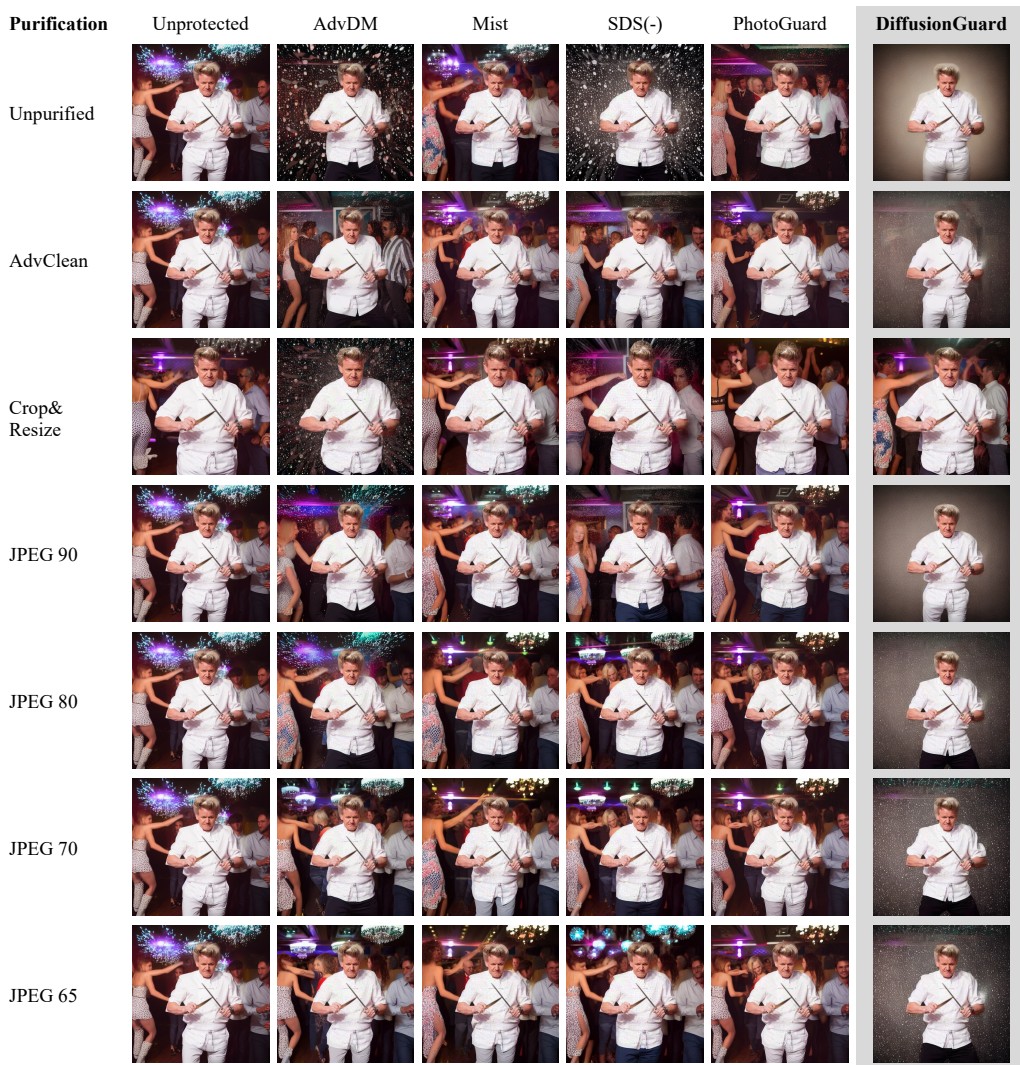

Figure 18: **Image editing results after applying purification method to images protected using each method (`Unseen` mask).** Editing prompt is `"A man dancing in a club"`.

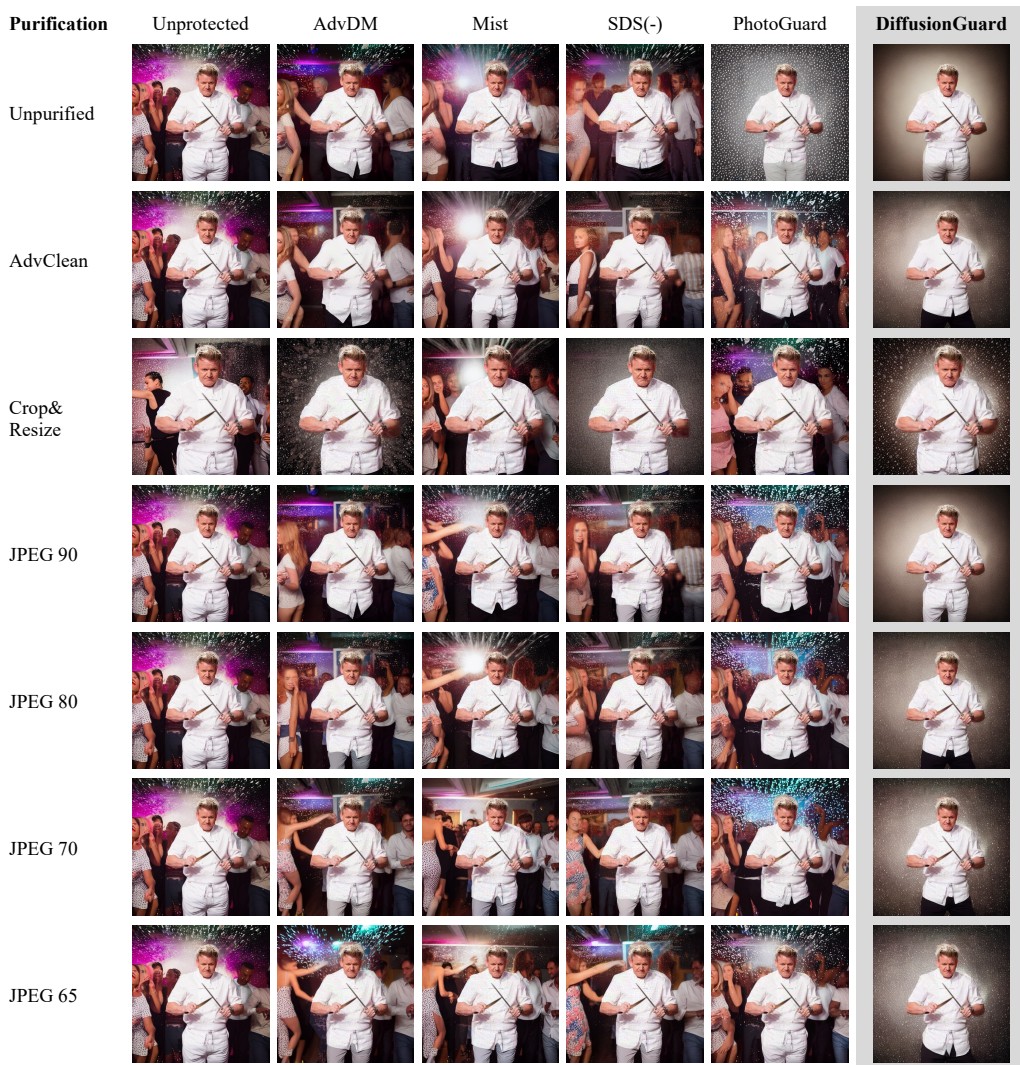

Figure 19: **Image editing results after applying purification method to images protected using each method (`Seen` mask).** Editing prompt is `"A man dancing in a club"`.

### E.4 EXPERIMENTS WITH INSTRUCTION-BASED EDITING MODEL

In this section, we compare the protective effectiveness of DiffusionGuard and the baseline methods when applied to an instruction-based editing model. Specifically, we edit images protected with DiffusionGuard and the baseline methods using InstructPix2Pix (Brooks et al., 2023), without providing a mask and only using the same editing prompts. For this experiment, we remove the mask augmentation component of DiffusionGuard and only use early-stage loss to generate adversarial perturbation against the model. Same applies for PhotoGuard, which adds the perturbation only in the mask region. The other three baseline methods were uses without any modification. The results are shown in Table 7.

Table 7: **Editing results of each protection method applied to InstructPix2Pix (Brooks et al., 2023).** Our method achieves strong protection, exhibiting **best** or second-to-best protection strength in most evaluations even when used with InstructPix2Pix, across all metrics. All methods were trained using constraint of $|\delta|_\infty = 16/255$. Lower metric is better, indicating failed edits.

| Method | PSNR ↓ | CDS ↓ | IR ↓ | CS ↓ |
|---|---|---|---|---|
| PhotoGuard | 17.19 | 15.02 | -1.508 | 22.95 |
| AdvDM | 14.53 | 22.15 | -1.234 | 27.18 |
| Mist | 14.35 | 22.82 | -1.204 | 27.48 |
| SDS(-) | **11.50** | 25.21 | -1.290 | 29.34 |
| DiffusionGuard | 17.42 | **14.07** | **-1.591** | **21.74** |

### E.5 EXPERIMENTS WITH MASKS LARGER AT TEST-TIME

Our proposed algorithm of mask augmentation shrinks the contour of the masks inwards to generate augmented masks. In this section, we verify whether such mask augmentation algorithm can be also generalized to when a mask larger than what was used for the generation of the adversarial perturbation is given at test-time, i.e. when a mask given at test time is larger than $M_{\mathtt{tr}}$ of DiffusionGuard. For this experiment, we algorithmically dilate the seen mask and one of the unseen masks to obtain two larger masks, until the masks became 26% larger on average compared to $M_{\mathtt{tr}}$. Then, directly test the images protected using each method from Section 4 with these new larger masks. Note that both masks are novel for all protection methods. The evaluation results for this experiments are presented in Table 8. As shown, even when a mask larger than what was used during the generation of the perturbation is given at test time, DiffusionGuard continues to demonstrate strong protective effectiveness, outperforming the baseline methods in most metrics. Even though mask augmentation of DiffusionGuard only shrinks the given mask and results in smaller masks, it is able to generalize to masks thar are larger as well.

Table 8: **Evaluation of DiffusionGuard and baseline methods using masks larger than the masks used during the generation of the perturbation (i.e. larger than $M_{\mathtt{tr}}$ of DiffusionGuard).** Our method still achieves strong protection, exhibiting **best** protection strength in most evaluations, across all metrics. All methods were trained using constraint of $|\delta|_\infty = 16/255$. Lower metric is better, indicating failed edits.

| Method | PSNR ↓ | CDS ↓ | IR ↓ | CS ↓ |
|---|---|---|---|---|
| PhotoGuard | 22.07 | -1.588 | 28.55 | 15.45 |
| AdvDM | 21.76 | -1.593 | 28.46 | **13.20** |
| Mist | 22.19 | -1.562 | 28.64 | 13.99 |
| SDS(-) | 21.29 | -1.587 | 28.20 | 13.96 |
| DiffusionGuard | **20.71** | **-1.709** | **27.86** | 14.72 |

### E.6 EXPERIMENTS WITH DIFFERENT SAMPLING STEPS AND DIFFERENT SAMPLER

As explained in Appendix D, we use DDIM (Song et al., 2021) sampler with 50 denoising steps for the generation of the edited images throughout this work. In this section, we verify whether DiffusionGuard is also effective when using DDIM sampler with different number of timesteps, or

even an entirely different sampler. This can be especially important as different timesteps or different samplers may start the denoising process with a different initial timestep. Specifically, we use DDIM sampler with 4 timesteps {25, 40, 50, 75}, and also experiment with DPM-Solver (Lu et al., 2022), which uses 25 denoising steps by default.

We report the comprehensive evaluation results in Table 9. As shown, DiffusionGuard consistently exhibits a strong protective effectiveness in all sampling setups.

Table 9: **Experiments with DDIM (Song et al., 2021) sampler with 3 additional denoising steps ({25, 40, 75}) and DPMS (Lu et al., 2022) sampler with 25 denoising steps.** Our method achieves strongest protection, exhibiting **best** protection strength in all evaluations, in both Seen and Unseen set, across all metrics. All methods were trained using constraint of $|\delta|_\infty = {}^{16}/_{255}$. Lower metric is better, indicating failed edits.

| Method | PSNR↓ | CDS↓ | IR↓ | CS↓ | PSNR↓ | CDS↓ | IR↓ | CS↓ |
|---|---|---|---|---|---|---|---|---|
| | Seen (1 Mask, Train set) | | | | Unseen (4 Masks, Test set) | | | |
| PhotoGuard DDIM (25) | 13.79 | 20.24 | -1.673 | 27.35 | 15.43 | 22.93 | -1.421 | 30.04 |
| AdvDM DDIM (25) | 14.43 | 23.50 | -1.462 | 29.89 | 14.16 | 23.92 | -1.382 | 30.73 |
| Mist DDIM (25) | 15.18 | 23.72 | -1.463 | 29.86 | 15.24 | 23.54 | -1.356 | 30.49 |
| SDS(-) DDIM (25) | 15.31 | 23.28 | -1.399 | 29.70 | 15.07 | 23.52 | -1.313 | 30.67 |
| Ours DDIM (25) | **13.45** | **18.86** | **-1.852** | **26.25** | **14.04** | **21.29** | **-1.596** | **28.59** |
| PhotoGuard DDIM (40) | 13.04 | 20.93 | -1.555 | 27.71 | 14.69 | 23.21 | -1.357 | 30.23 |
| AdvDM DDIM (40) | 13.73 | 23.76 | -1.449 | 30.07 | 13.52 | 24.27 | -1.358 | 30.94 |
| Mist DDIM (40) | 14.36 | 24.10 | -1.403 | 30.21 | 14.59 | 23.75 | -1.300 | 30.65 |
| SDS(-) DDIM (40) | 14.55 | 23.39 | -1.324 | 29.93 | 14.45 | 23.79 | -1.251 | 30.82 |
| Ours DDIM (40) | **12.72** | **19.02** | **-1.800** | **26.52** | **13.29** | **21.59** | **-1.562** | **28.90** |
| PhotoGuard DDIM (50) | 12.87 | 21.17 | -1.537 | 27.89 | 14.53 | 23.30 | -1.357 | 30.30 |
| AdvDM DDIM (50) | 13.62 | 23.77 | -1.438 | 30.04 | 13.37 | 24.27 | -1.361 | 30.97 |
| Mist DDIM (50) | 14.22 | 24.25 | -1.368 | 30.27 | 14.51 | 23.93 | -1.307 | 30.79 |
| SDS(-) DDIM (50) | 14.44 | 23.42 | -1.337 | 29.89 | 14.32 | 23.85 | -1.237 | 30.78 |
| Ours DDIM (50) | **12.60** | **18.95** | **-1.807** | **26.55** | **13.19** | **21.84** | **-1.557** | **29.05** |
| PhotoGuard DDIM (75) | 13.15 | 21.27 | -1.564 | 28.06 | 14.59 | 23.53 | -1.368 | 30.31 |
| AdvDM DDIM (75) | 13.69 | 23.70 | -1.434 | 30.04 | 13.36 | 24.20 | -1.376 | 30.90 |
| Mist DDIM (75) | 14.48 | 23.76 | -1.399 | 30.00 | 14.56 | 23.84 | -1.321 | 30.67 |
| SDS(-) DDIM (75) | 14.70 | 23.51 | -1.363 | 29.95 | 14.33 | 23.81 | -1.237 | 30.71 |
| Ours DDIM (75) | **12.87** | **19.07** | **-1.806** | **26.68** | **13.29** | **22.08** | **-1.572** | **29.12** |
| PhotoGuard DPMS (25) | 9.49 | 17.37 | -1.779 | 24.82 | 11.83 | 18.75 | -1.682 | 26.73 |
| AdvDM DPMS (25) | 10.70 | 18.29 | -1.776 | 25.75 | 10.63 | 19.37 | -1.744 | 27.27 |
| Mist DPMS (25) | 11.28 | 18.51 | -1.754 | 26.05 | 11.67 | 19.21 | -1.690 | 27.15 |
| SDS(-) DPMS (25) | 11.57 | 17.91 | -1.796 | 25.69 | 11.74 | 18.77 | -1.734 | 26.99 |
| Ours DPMS (25) | **9.31** | **15.58** | **-1.931** | **23.61** | **10.20** | **17.48** | **-1.816** | **25.53** |

## F    PROTECTION RESILIENCE AGAINST NOISE PURIFICATION

In this section, we report the protection effectiveness of each method after applying purification methods. It is known that adversarial perturbations can be "purified" by applying modification (*e.g.* JPEG compression, crop-and-resize) to the image which contains an adversarial perturbation. Because adversarial perturbations added by image protection methods are optimized in pixel-space using the PGD algorithm, any operation that modifies the image may impair the adversarial perturbation and decrease the protection effectiveness. Commonly used methods include the following methods.

- Crop-and-resize: The image is cropped at the center and is resized up to match the original resolution.

- JPEG compression: A universally adopted algorithmic image compression method. It accepts a quality parameter ranging from 0 to 100, where lower values result in worse image quality (hence greater noise removal and stronger purification).

- AdverseCleaner (Zhang, 2023): An algorithmic filter which removes adversarial perturbation targeted at diffusion models.

In this section, we conduct additional experiments extending from Sec. 4 to evaluate DiffusionGuard and the baseline methods after processing the protected images using each of the purification methods and present the full results in Table 6. As presented, As shown in the table, DiffusionGuard (noted as "Ours" in the table, highlighted in grey color) results in the best or the second best protection strength in most cases across all most PGD optimization durations and across all evaluation metrics, exhibiting better resource efficiency compared to the baseline methods. Additionally, we include the qualitative results for each protection method and each purification method in Fig. 18 and Fig. 19 (edited using `Unseen` and `Seen` in order). As shown in the figures, DiffusionGuard is resilient against various noise purification methods compared to the baselines, maintaining its protective strength even after a malicious user attempts to remove the adversarial perturbation.

Table 10: **Editing results of each protection method when black-box transferred to Stable Diffusion 2.0 Inpainting.** Our method achieves strong protection, exhibiting **best** or second-to-best protection strength in most evaluations even when transferred to a different model, in both `Seen` and `Unseen` set, across all metrics. All methods were trained using constraint of $|\delta|_\infty = {}^{16}/{}_{255}$. Lower metric is better, indicating failed edits.

| Method | PSNR ↓ | CDS ↓ | IR ↓ | CS ↓ | PSNR ↓ | CDS ↓ | IR ↓ | CS ↓ |
|---|---|---|---|---|---|---|---|---|
| | Seen (1 Mask, Train set) | | | | Unseen (4 Masks, Test set) | | | |
| PhotoGuard | **14.85** | 22.17 | -1.468 | 28.34 | 16.16 | 23.66 | -1.234 | 30.32 |
| AdvDM | 14.95 | 22.88 | -1.402 | 28.98 | **14.67** | 23.29 | -1.255 | 30.21 |
| Mist | 15.82 | 22.73 | -1.341 | 29.02 | 16.15 | 23.71 | -1.180 | 30.39 |
| SDS(-) | 15.82 | 22.33 | -1.325 | 28.70 | 15.65 | **22.88** | -1.191 | 29.99 |
| DiffusionGuard | 14.97 | **21.64** | **-1.616** | **28.03** | 15.12 | 23.21 | **-1.395** | **29.96** |

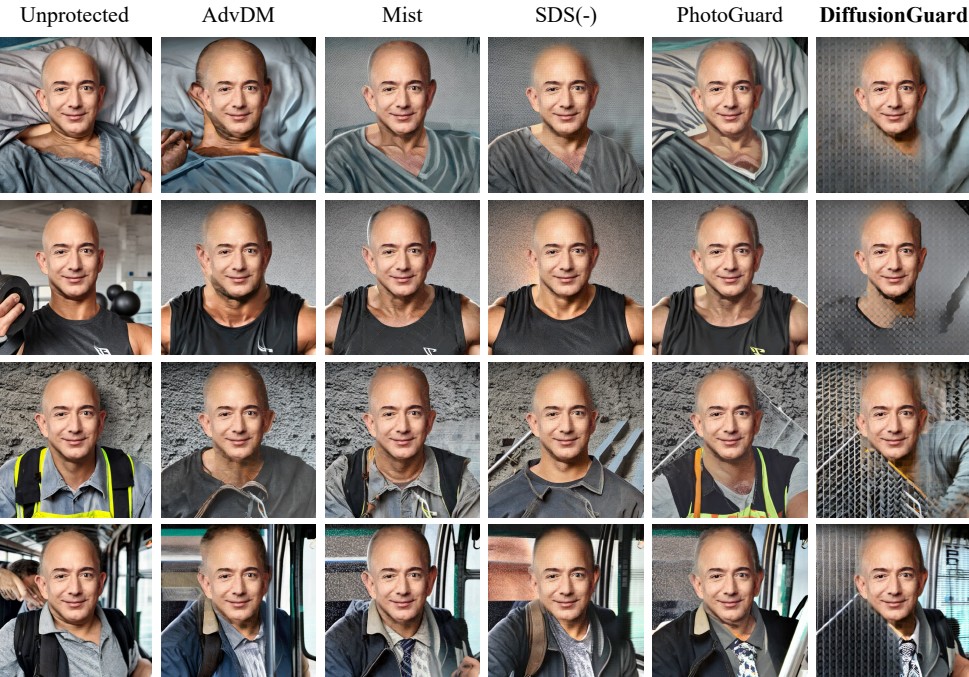

| Unprotected | AdvDM | Mist | SDS(-) | PhotoGuard | **DiffusionGuard** |

Figure 20: **Black-box transfer to Stable Diffusion 2.0 Inpainting from Stable Diffusion Inpainting, comparison between DiffusionGuard and baselines (`Unseen` mask).** Editing prompts for each row are `"A man in a hospital"`, `"A man in the gym"`, `"A photo of a construction worker"`, and `"A man getting on a bus"` in order.

| Unprotected | AdvDM | Mist | SDS(-) | PhotoGuard | **DiffusionGuard** |
|---|---|---|---|---|---|

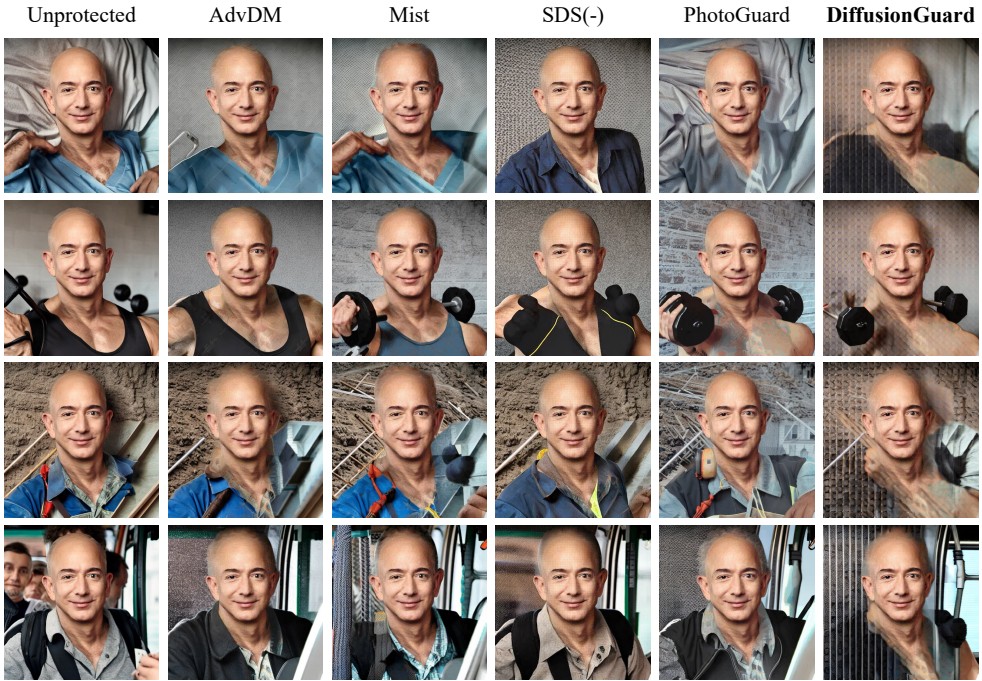

Figure 21: **Black-box transfer to Stable Diffusion 2.0 Inpainting from Stable Diffusion Inpainting, comparison between DiffusionGuard and the baseline methods (`Seen` mask).** Editing prompts for each row are `"A man in a hospital"`, `"A man in the gym"`, `"A photo of a construction worker"`, and `"A man getting on a bus"` in order.

## G TRANSFERABILITY TO BLACK-BOX MODELS

In this section, we conduct additional experiments to assess the black-box transfer effectiveness of DiffusionGuard and the baseline methods. Specifically, we use Stable Diffusion Inpainting 1.0 (Rombach et al., 2023) for generating adversarial examples, and test them on Stable Diffusion 2.0 Inpainting. We note that Stable Diffusion Inpainting 1.0 is a fine-tuned checkpoint of Stable Diffusion 1.2, and Stable Diffusion 2.0 Inpainting is a fine-tuned checkpoint of Stable Diffusion 2.0. Because Stable Diffusion 2.0 was pre-trained from scratch independently from the weights of Stable Diffusion 1.2, the weights of the two inpainting models are significantly different, making this evaluation setup a black-box transfer setting. The full evaluation results are presented in Table 10. As shown in the table, DiffusionGuard exhibits strong protection when transferred to a different model (*i.e.* used against a different editing model), achieving either best or second-to-best value in most cases across both `Seen` and `Unseen` of InpaintGuardBench. We also present the qualitative results in Fig. 20 and Fig. 21, respectively for `Unseen` and `Seen` masks. As visualized, DiffusionGuard maintains its protective effectiveness when transferred to a different model, while baseline methods fail to protect the image and results in successful edits.

## H    COMPARISON OF NOISE VISIBILITY

In this section, we both quantitatively and qualitatively compare the visibility, hence the stealthiness of each protection method. For the quantitative comparison, we compute distance metrics between the clean (unprotected) image and the protected image in order to verify how similar the protected image is to the clean image. We outline the qualitative comparison of the visibility of the adversarial perturbation generated by each noise in Fig. 22. As illustrated, the noise itself do not vary significantly in it visibility, as all methods including DiffusionGuard and the baselines were optimized using PGD with the same perturbation strength threshold ($|\delta|_\infty = 16/255$). In fact, DiffusionGuard and PhotoGuard are significantly more stealthy compared to the other three baselines as they add noise only within the small mask region, whereas the noise occupies the entire image in the case of Mist, AdvDM, and SDS(-).

The quantitative results presented in Table 11 are also aligned with these findings. DiffusionGuard achieves significantly higher SSIM and PSNR compared to the baseline methods, meaning it is closer to the clean image and thus has less visible noise. Additionally, we measure the L2 distance, which DiffusionGuard achieves the lowest value, representing similarity to the clean image. Finally, we report DreamSim (Fu et al., 2023), a perceptual distance metric based on an ensemble of foundational vision encoders such as CLIP (Radford et al., 2021). DiffusionGuard also achieves the lowest value in this metric as well.

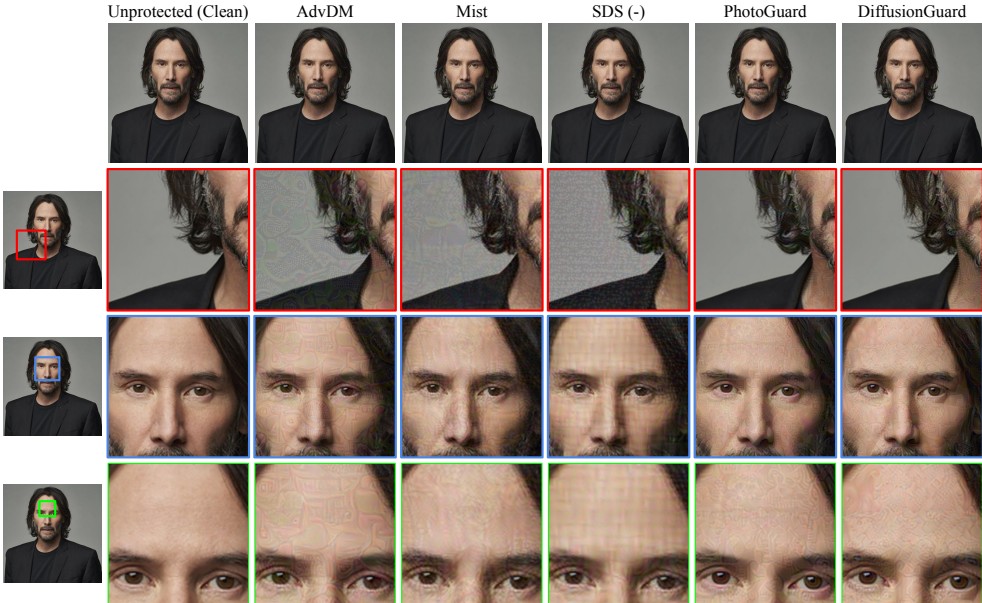

Figure 22: **Qualitative comparison of noise visibility of DiffusionGuard and the baseline methods.** First column represents the zoom-in location, and the other rows represent protection methods. As visible in the second row (red outlines), PhotoGuard and DiffusionGuard only add noise inside the mask, resulting in a less visible noise overall, whereas other three methods add noise to the entire image.

Table 11: **Quantitative comparison of the visibility of the adversarial perturbation generated using DiffusionGuard and baselines.** Each metric measures the similarity (PSNR, SSIM) or the distance (L2, DreamSim) of the protected image from the clean image.

| Method | PSNR ↑ | SSIM ↑ | L2 ↓ | DreamSim ↓ |
|---|---|---|---|---|
| AdvDM | 32.19 | 0.873 | 21.81 | 0.0419 |
| Mist | 32.61 | 0.883 | 20.80 | 0.0408 |
| SDS(-) | 31.61 | 0.851 | 23.33 | 0.0467 |
| PhotoGuard | 41.39 | 0.989 | 7.875 | 0.00693 |
| DiffusionGuard | **41.58** | **0.990** | **7.639** | **0.00689** |

## I    COMPARISON OF INSTRUCTION-BASED EDITING AND INPAINTING

In our work, we propose DiffusionGuard, a protection method specialized against inpainting methods and inpainting models. While we perform additional study in Appendix E.4 to verify that DiffusionGuard can also be used with non-inpainting editing methods such as instruction-based, our focus on inpainting models is based on the practical usefulness of these models compared to instruction-based models.

Instruction-based models are easier to use because they do not require binary masks, but they tend to preserve high-level structures such as body postures, large objects, backgrounds, and text. This limits their ability to make drastic edits, as illustrated in Fig. 23. For example, InstructPix2Pix often over-conditions on the original image structure, failing to follow instructions precisely. In the first row of Fig. 23, instead of showing a man being arrested, it generates a police officer due to the original posture conditioning. Additionally, it changes the face, limiting its potential as an identity-stealing privacy threat. In the third row, the inpainting result in the third row shows a woman dancing, while InstructPix2Pix retains her posture same as the source image.

Note that we modified the text prompt into an appropriate instruction-like form for InstructPix2Pix models (e.g. "A man dressed up in halloween costume" → "Make the man be dressed in halloween costume").

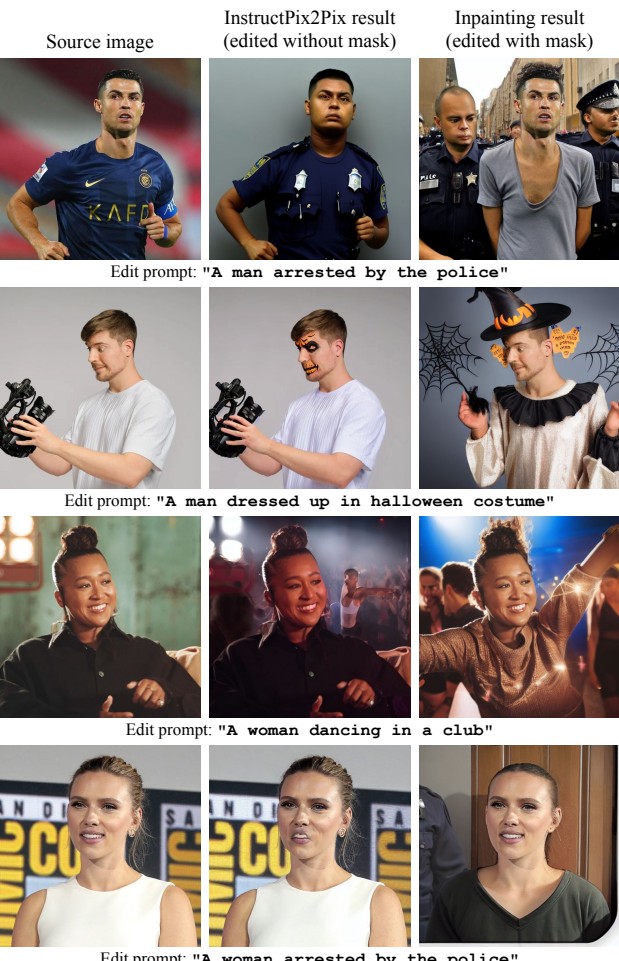

Figure 23: **Qualitative comparison instruction-based editing (InstructPix2Pix) and masked inpainting (Stable Diffusion Inpainting).** Inpainting allows a complete regeneration of the area designated by the mask, enabling a more flexible change of the overall structure of the image. In contrast, instruction-based editing preserves the overall structure of the image, resulting in images less aligned with the instruction especially if it requires a more drastic change.

## I.1 EDITING A MALICIOUS IMAGE TO CHANGE THE FACE

It is also possible to start an image that already contains a malicious context, such as an image of a person being arrested, and instruct an instruction-based editing model to change the face of the image to that of a desired person (e.g., a celebrity). In this section, we edit an image of a person being arrested and use InstructPix2Pix (Brooks et al., 2023) and Stable Diffusion Inpainting (Rombach et al., 2023) to edit it into a celebrity being arrested. Specifically, we use the instruction "Turn the man into celebrity name" with 10 different well-known celebrities that the editing model is aware of. We visualize the editing results in Fig. 24. As visualized in the figure, InstructPix2Pix results in less desirable results compared to Stable Diffusion Inpainting, modifying the unrelated area such as the face of the police officer (all images), or the background (Lionel Messi and Christiano Ronaldo). The generated face quality is also lower for InstructPix2Pix. In contrast, Stable Diffusion Inpainting results in more successful editing results as it receives a mask to designate which part of the image should be modified.

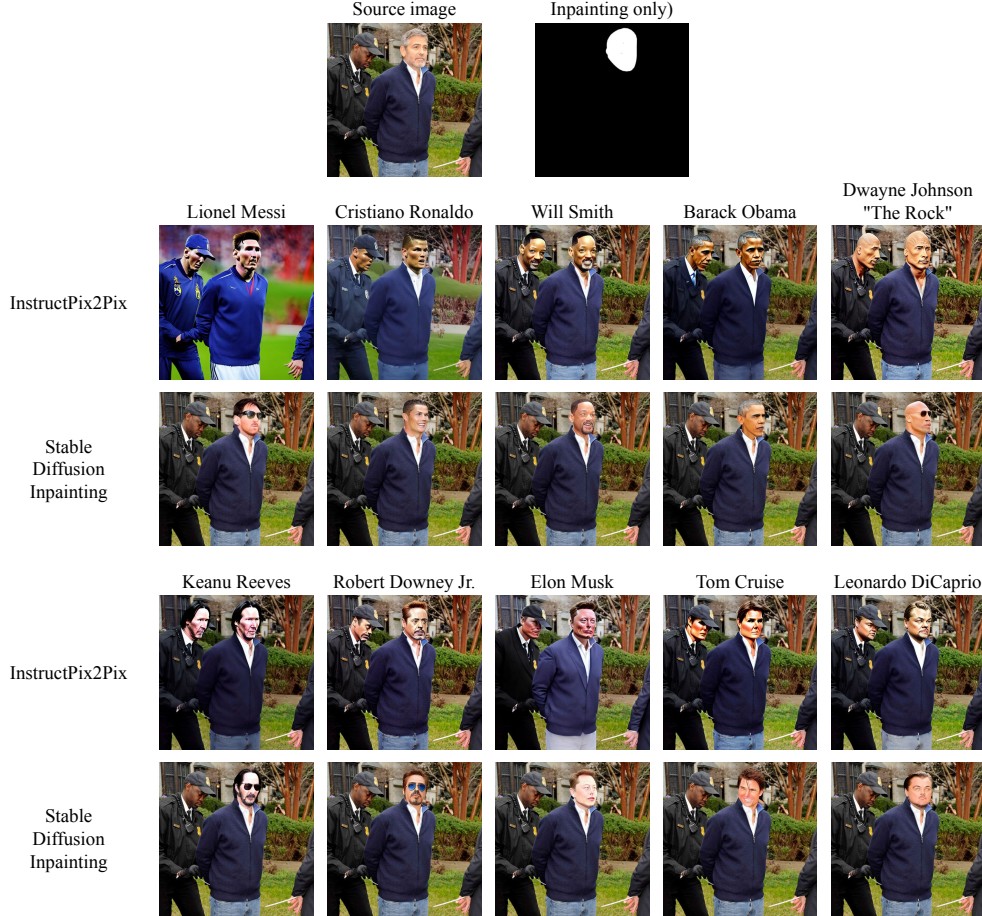

Figure 24: **Edited results starting from a person being arrested and changing the face using InstructPix2Pix and Stable Diffusion Inpainting using the prompt "Turn the man into {celebrity name}".** As shown, the images edited using InstructPix2Pix result in relatively more unrealistic images, always changing the police officer to the celebrity together with the person being arrested, and resulting in low-quality face images. Additionally, when the conditioning given by the celebrity name is strong (e.g., football players), it even changes the background similarly to a football stadium. In contrast, inpainting results in a consistent result as it uses masks to designate which area should be modified. (*Disclaimer: The results and examples presented in this paper are intended for academic and research purposes only. The edited images are not meant to misrepresent or defame any individuals, organizations, or professions depicted. The use of names and identities is strictly for illustrative purposes and does not imply endorsement, association, or real-life events.*)

## I.2 EDITING A CELEBRITY IMAGE TO CHANGE THE BACKGROUND

Another way of generating an image with malicious intent would be to modify the background of an image which depicts a specific identity. In this section, we use InstructPix2Pix (Brooks et al., 2023) and Stable Diffusion Inpainting (Rombach et al., 2023) to edit celebrity images by changing their background. For InstructPix2Pix, we used instruction "Change the background to jail" and for Stable Diffusion Inpainting, we used prompt "A photo of a person in jail", and a mask designating the face for each image. The results are illustrated in Fig. 25. As shown in the figure, InstructPix2Pix struggles with generating a successful edit, often changing the celebrity into a different person (Rows 1, 3, 4, 6). There area also cases where the model over-conditions on the source image such as the postures or letters inside them (Rows 2, 5, 6). In contrast, the inpainting model successfully generates accurate representations of celebrities in a jail setting in all cases.

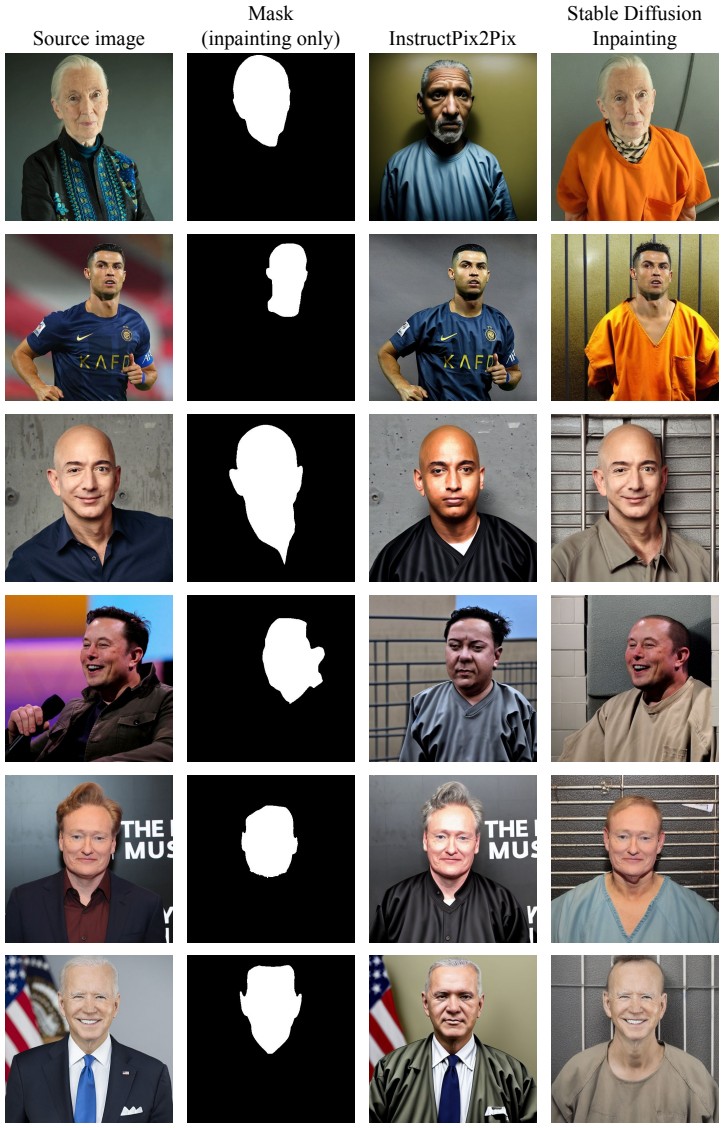

Figure 25: **Edited results starting from celebrity images and changing the background using InstructPix2Pix using the instruction "Change the background to jail", and Stable Diffusion Inpainting using the prompt "A photo of a person in jail".** InstructPix2Pix is less effective for generating a successful edit, often altering the celebrity into a different person (Rows 1, 3, 4, 6) or over-conditioning on the source image such as the postures or letters inside them (Rows 2, 5, 6), whereas Stable Diffusion Inpainting successfully generates accurate images of celebrities in a jail in all images. (*Disclaimer: The results and examples presented in this paper are intended for academic and research purposes only. The edited images are not meant to misrepresent or defame any individuals, organizations, or professions depicted. The use of names and identities is strictly for illustrative purposes and does not imply endorsement, association, or real-life events.*)

## J    EDITING THE INSIDE OF THE MASK

In this section, we explore whether DiffusionGuard is also able to protect images when inpainting the inside of the sensitive region, instead of inpainting the background behind the sensitive region (e.g. face of a person), which is what we focus on in this work. For example, a malicious user could aim to edit a specific sub-part inside the sensitive region, such as the eyes. To evaluate whether DiffusionGuard is able to protect the images against these cases as well, we take the images protected using DiffusionGuard using $M_{\text{tr}}$ (Sec. 4) and edit them with a novel mask containing the eyes. We outline the used source image, the mask, and the editing prompts in Fig. 26.

As shown in Fig. 26, DiffusionGuard is also able to protect the images from being edited inside the sensitive region. While the unprotected images result in plausible images of a man, DiffusionGuard protected images result in deformed and unrealistic eyes, making the image implausible. The reason why DiffusionGuard is able to protect against these cases is likely due to the fact that most of the noise added to the face still survives and is fed into the inpainting model, causing the edit to fail.

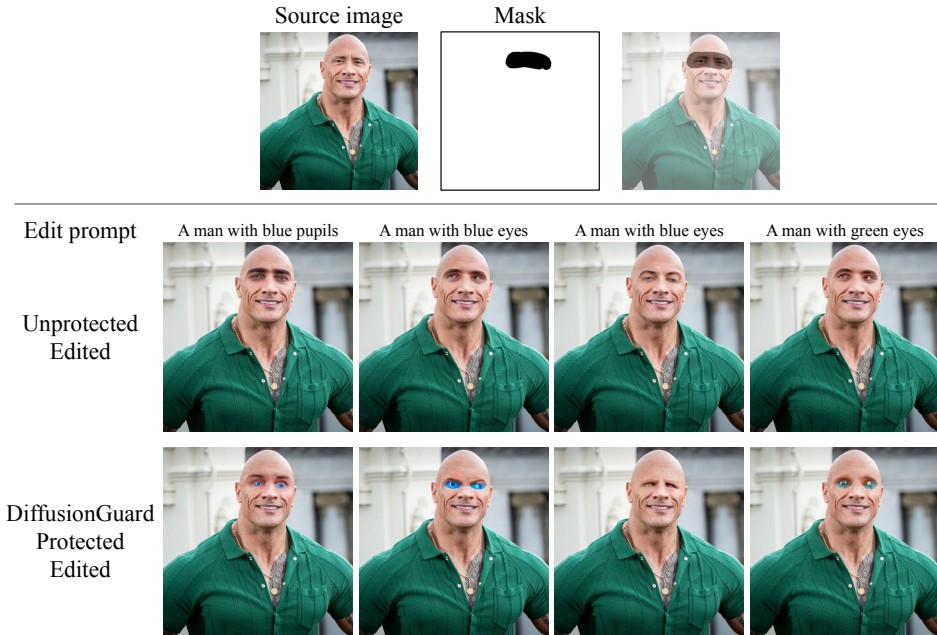

Figure 26: **Editing the inside of the mask.** We edit the source image using a mask containing only the eyes. The protected images were protected using the face for the generation of the adversarial perturbation. As shown in the third row, DiffusionGuard is also able to protect the images from being edited inside the mask.

### J.1    GENERATING SUB-PERTURBATIONS

In this section, we explore the possibility of generating parts of the perturbation separately and then unifying them to one to obtain a new perturbation. The motivation of this idea is the possibility of a malicious user trying to edit the inside of the face, and the regions which they would try to edit in this case would be based on human criteria (e.g., selecting facial features). For this section, we consider a simple case of a single portrait image. The overall flow is illustrated in Fig. 27. Given a sensitive region (e.g., face), we split the region into several parts (3 parts in the figure) based on an arbitrary criteria. We will call refer this splitted part as "sub-mask". For this experiment, we splitted the face into three sub-masks, as shown in the figure: 1) eyes and forehead, 2) rest of the face (cheeks, nose, and mouth), and 3) neck. After splitting, we applied DiffusionGuard to obtain adversarial perturbation for each sub-mask (obtaining sub-perturbation), and merged the perturbations into one by taking the union of the three sub-perturbations.

Then, we evaluate this new merged perturbation compared to a normal perturbation obtained with default DiffusionGuard. We evaluate under four scenarios: editing the eyes, editing the mouth, editing

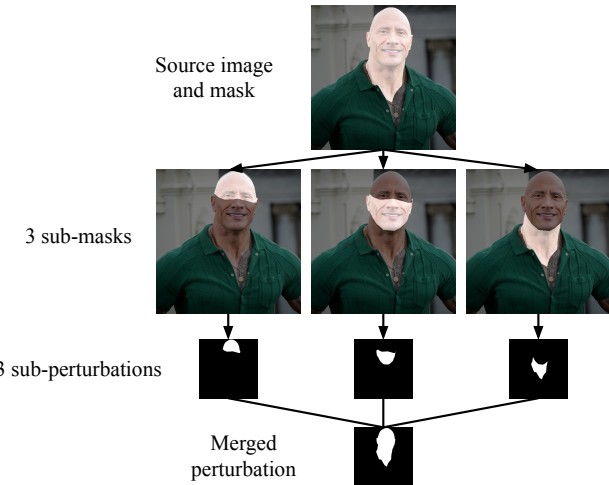

Figure 27: **Overview of obtaining sub-perturbations of a single perturbation and merging them into one.**

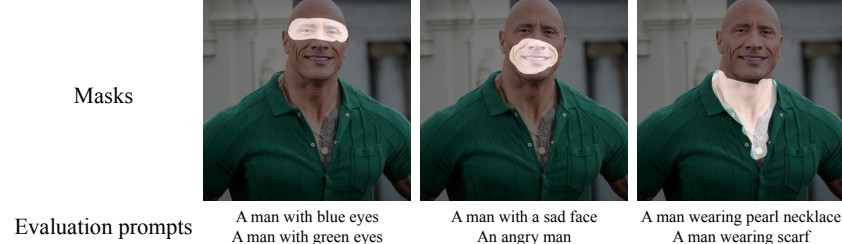

Figure 28: **Test set masks and the evaluation prompts used for each mask for the sub-perturbation experiments.**

the neck, and editing the entire image. The masks used for the first three scenarios are visualized in Fig. 28. For the entire image scenario, we use the same prompts as our main experiments (Sec. 4). We sample 32 images for each prompt with different seeds and report the average values. For the source image, we use only one image of Dwayne Johnson (same as Fig. 27). Same as our main experiments, we measure PSNR, CLIP directional similarity, CLIP similarity, and ImageReward as our quantitative metrics. The evaluation results are presented in Table 12. Interestingly, the merged perturbation has similar protective effectiveness as the original DiffusionGuard for localized edits, and is slightly better for full-image protection. Considering the inner workings of inpainting models, this is likely due to the fact that protective noise over targeted areas is not inputted to the inpainting model, the noise being omitted from the input.

Table 12: **Evaluation result of DiffusionGuard with and without using sub-perturbation.** CDS, CS, IR represent CLIP directional similarity, CLIP similarity, and ImageReward respectively.

| Method | PSNR ↓ | CDS ↓ | CS ↓ | UR ↓ |
|---|---|---|---|---|
| Mask: Eyes | | | | |
| Default DiffusionGuard | **33.45** | 1.48 | **20.28** | **-1.503** |
| Sub-perturbation DiffusionGuard | 34.45 | **0.67** | 20.31 | -1.442 |
| Mask: Mouth | | | | |
| Default DiffusionGuard | 30.71 | 8.03 | 17.91 | -1.363 |
| Sub-perturbation DiffusionGuard | **31.73** | **7.95** | **17.82** | **-1.400** |
| Mask: Neck | | | | |
| Default DiffusionGuard | **30.63** | **14.12** | **20.01** | **-0.269** |
| Sub-perturbation DiffusionGuard | 31.71 | 16.88 | 20.48 | 0.071 |
| Mask: Entire | | | | |
| Default DiffusionGuard | 37.47 | 2.34 | 19.51 | -1.98 |
| Sub-perturbation DiffusionGuard | **34.18** | **2.12** | **19.40** | **-1.99** |

## K   COMPARISON OF MASKS WITH DIFFERENT SIZES

In this section, we qualitatively compare editing results with varying sizes of masks and discuss insights from the editing results. The results are visualized in Fig. 29. We use 7 masks, 4 smaller than the head (Rows 2–5), 1 matching the head size (Row 1), and 2 larger than the head (Rows 6–7). The editing prompt used is "A man in a hospital".

One insight shown in the result is that masks larger than the head, which include regions outside the head (especially the background), restrict the editing flexibility. In the figure, for these larger masks, the background remains fixed as an empty entrance to a dark hallway (Rows 6–7). This is likely to align with the dark surroundings of the source image, forcing the model to generate a dark hallway where the background remains in the mask region. Also, the shirt color is consistently dark blue due to the original clothing. In contrast, the smaller masks (Rows 1–5) allow diverse backgrounds, such as a wall, hallway, or hospital ward. This suggests that larger masks, as they include the background, are less ideal for flexible editing and emphasize focusing on the head, especially the face.

Another observation is that for all 7 masks that are visualized, including the smoother backgrounds of Rows 6–7, the boundary between the original face and the edited background is rather visibly distinct in the protected images. This is likely due to the attack disrupting the inpainting model's internal processes, causing it to misinterpret colors and generate incorrect colors.

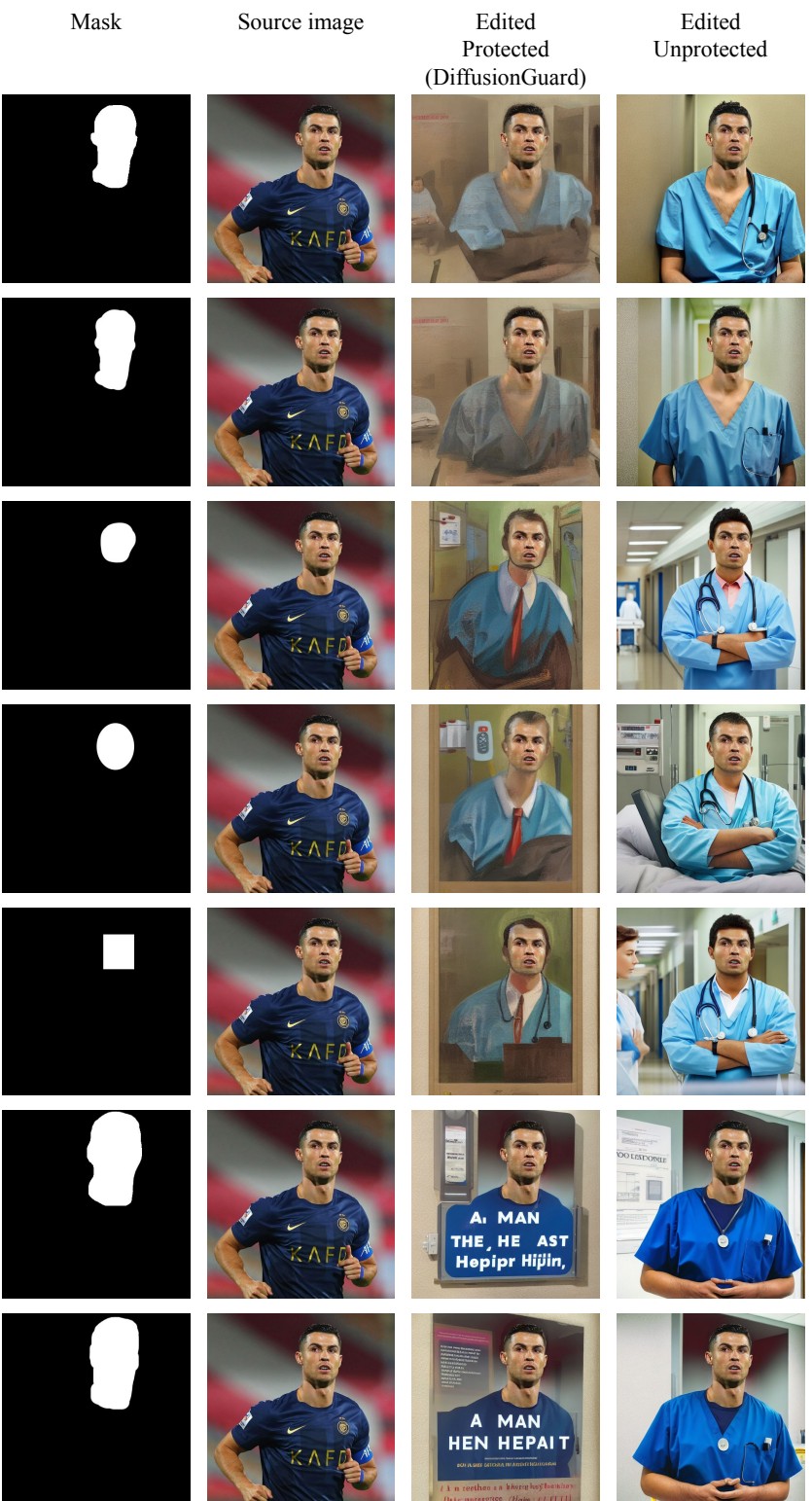

Figure 29: **Edited results using various shapes and sizes of masks, with prompt "A man in a hospital".** For rows 2, 3, 4, 5, the mask is smaller than the head, and for rows 6, 7, the mask is larger than the head. For row 1, the mask matches the size of the head. As visualized, DiffusionGuard successfully protects the image in all cases. Interestingly, larger mask causes the background around the face to leak in, forcing the generated image to have certain colors (and objects) around the face.

## L    LIMITATION

There are several limitations and interesting future directions in our work:

- **Black-box setups**: Although we demonstrate the effectiveness of DiffusionGuard in black-box settings in Sec. 4.6 and Appendix G, further investigations are required against more advanced closed models, such as DALL-E 3 (Betker et al., 2023).

- **Extension to personalization**: Text-to-image diffusion models have shown remarkable success in generating personalized subjects based on a few reference images (Ruiz et al., 2023). Because such personalized models can be misused to generate harmful content, developing defense methods against personalization methods would be an important direction for future research.

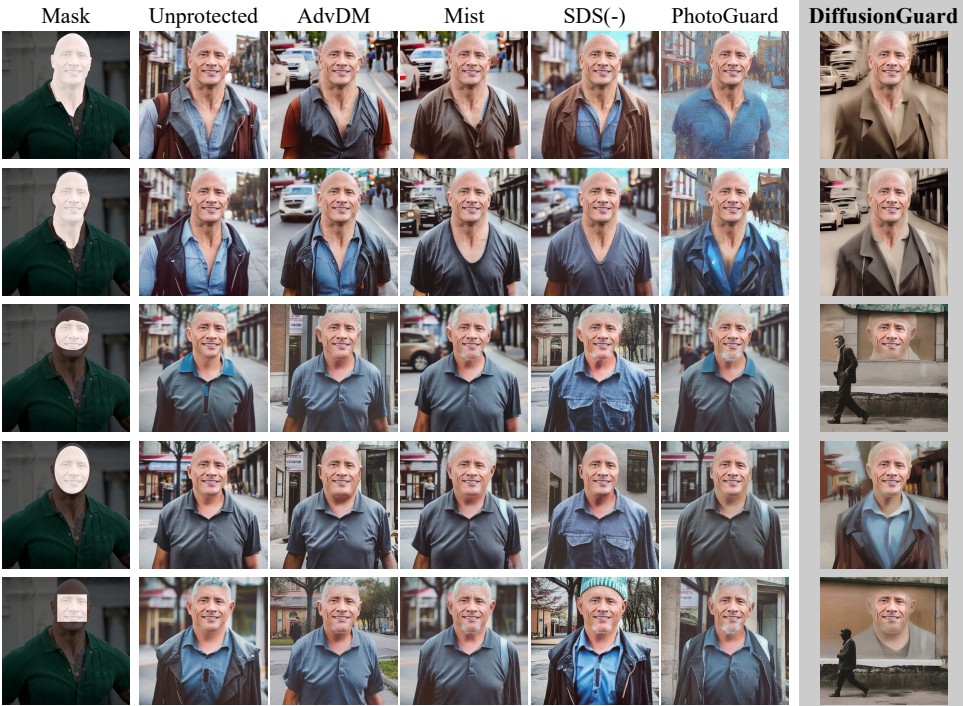

Figure 30: **Edited results after using DiffusionGuard and the baseline protection methods (all masks shown).** Editing prompt is `"A man walking in the street"`.

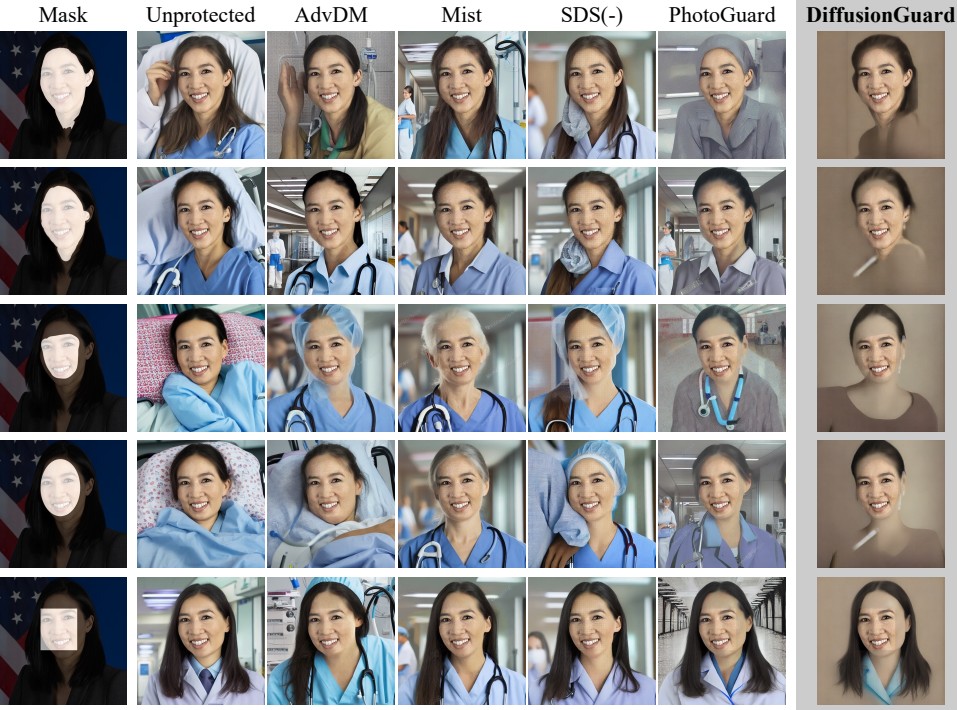

Figure 31: **Edited results after using DiffusionGuard and the baseline protection methods (all masks shown).** Editing prompt is `"A woman in a hospital"`.

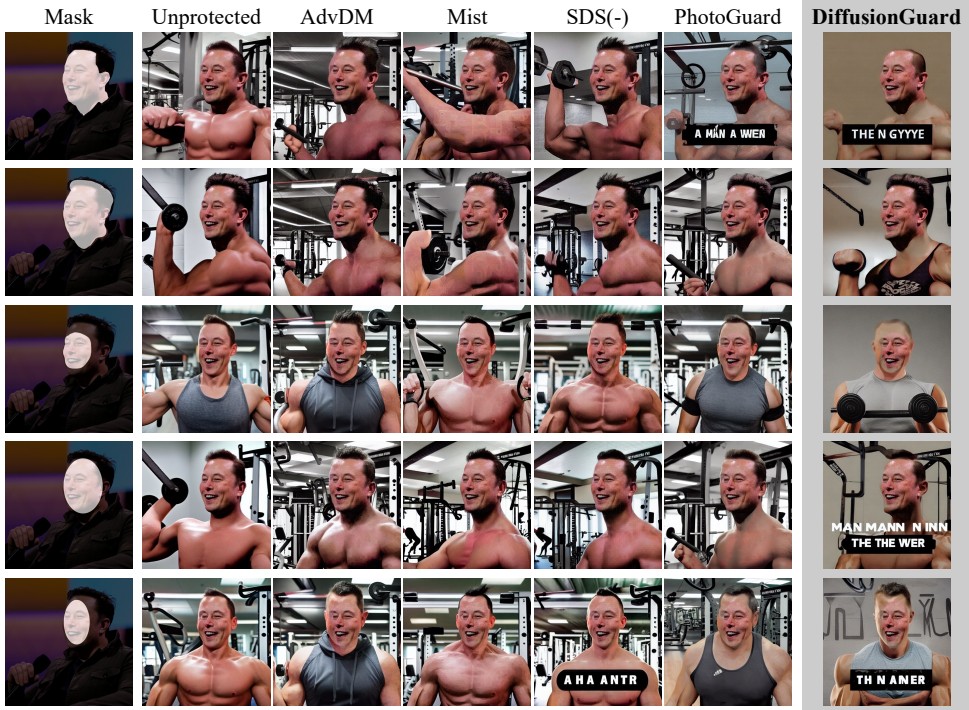

Figure 32: **Edited results after using DiffusionGuard and the baseline protection methods (all masks shown).** Editing prompt is `"A man in a gym"`.

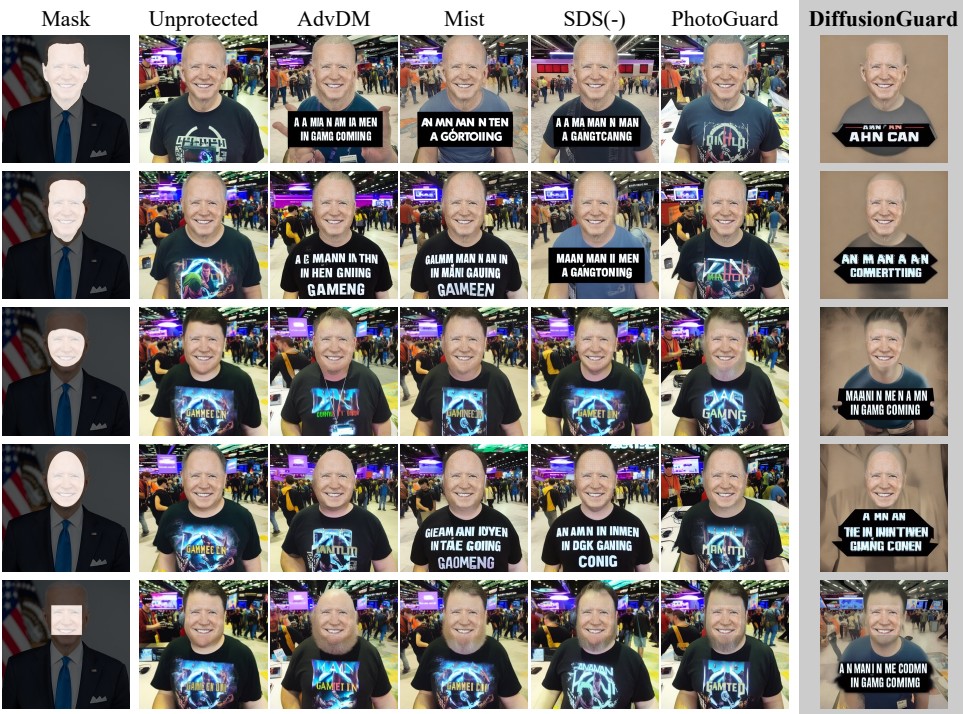

Figure 33: **Edited results after using DiffusionGuard and the baseline protection methods (all masks shown).** Editing prompt is `"A man in a gaming convention"`.

