# OpenReview forum: "DiffusionGuard: A Robust Defense Against Malicious Diffusion-based Image Editing"
_ICLR.cc/2025/Conference — ICLR 2025 Poster_

### Official Review · Reviewer_WYYe · 2024-10-22

**Soundness:** 3
**Presentation:** 3
**Contribution:** 3
**Rating:** 6
**Confidence:** 4

**Summary:**

The author proposed an attack method that targeting the LDM-based inpainting task. The method came with a new loss fucntion, and a new data agumentation for inpainting mask. The author also proposed a new benchmark for evaluate the anti-inpainting methods. The experiments showed the good performance. Generally it's a complete work.

**Strengths:**

1. The author captured the key problem that the global adversarial perturbation will loss its adversarial semantic in inpainting due to the mask
2. The proposed method only need to run one step of U-Net in each attack step
3. The experiment is quite comprehensive

**Weaknesses:**

1. The motivation for the loss is not clear
2. There is no ablation study for the hyperparameters

More details see questions

**Questions:**

1. In line 242, the author wrote "by minimizing the following loss ...". Should it be minimizing or maximizing? I was a little bit confused. I hope the author could clearify this.
2. The author proposed a new objective or loss function for the PGD-like attack. Could the author tell me why you choose to maximize the L2 norm of the predicted noise in the earlier step? Why not L1 or total variation or focal norm? I didn't see any explaination of why max(L2) works.
3. The author didn't show that the influence of different early step `t`. For example, from 1 to 10, there is no ablation about it. I'd like to see what's the influence of different `t`, and why.
4. Although the author has done the ablation for attack steps (comp. budget), the author didn't do the ablation for learning rate choice. Some previous papers mentioned the larger learning rate choice may cause better performance when attacking the generation task in some cases, which is counter-intuitive. I hope the author can do the ablation for this as well, to find the best hyperparameters settings.
5. regarding the line 175 to 183, the authors mention that they only apply perturbations in sensitive areas most commonly used by malicious users. This is reasonable in most cases, but if a malicious user only wants to edit the eyes in a face photo, and wants to change the shape of the pupil or the position where the line of sight is focused, will inpainting be successful in this case? I would be inclined to think that editing or inpainting would still succeed in this case.
    Therefore, I would like to ask the author to make a demo. For example, in a face image, the facial features are masked separately for attack, and then a perturbation composed of several sub-perturbations will be obtained. I want to see how effective a certain sub-perturbation is in this case. Whether the inpainting of the sub-entity will be successful and whether the editing of the entire entity will be successful.
    I expect the author can take series of experiments and show me the results, and it would be better if it could be analyzed quantitatively.
6. Some figures are too tiny to read such as Figure 7a, author may prefer to make the 7a wider to have a better visualization.

---

> ### Author Response · Authors · 2024-11-21
>
> Dear Reviewer WYYe,
>
> We sincerely appreciate your insightful comments. They were incredibly helpful in improving our draft. We have addressed each comment in detail below.
>
> ---
>
> **[Q1] Clarification of loss maximization**
>
> Thank you very much for your careful review, and we apologize for the confusion. The loss should indeed be maximized, as stated in Eq. 5. We have clarified this in the revised draft.
>
> ---
>
> **[Q2] Reason for choosing L2 norm of the predicted noise.**
>
>
> In developing our method, we experimented with various loss function designs, including the L1 norm and total variation, as you mentioned. We found that maximizing the L2 norm empirically provided the most effective protection results. Table A (below) reports the performance of DiffusionGuard when using the L1 norm and total variation of the predicted noise as the loss function.
>
> **Table A. DiffusionGuard with L1 or total variation loss**
>
> | **Method**                              | **CLIP Dir. Sim ↓** | **CLIP Sim. ↓** | **ImageReward ↓** | **PSNR ↓** |
> |-----------------------------------------|----------------------|------------------|--------------------|------------|
> | **Seen mask**                           |                      |                  |                    |            |
> | DiffusionGuard, L1 loss                 | 19.37               | -1.789           | 26.71              | 12.62      |
> | DiffusionGuard, total variation loss    | 20.79               | -1.766           | 27.47              | 12.88      |
> | **DiffusionGuard, L2 loss (default)**   | **18.95**           | **-1.807**       | **26.55**          | **12.60**  |
> | **Unseen mask**                         |                      |                  |                    |            |
> | DiffusionGuard, L1 loss                 | 21.97               | -1.549           | 29.14              | 13.23      |
> | DiffusionGuard, total variation loss    | 22.60               | -1.538           | 29.47              | 13.22      |
> | **DiffusionGuard, L2 loss (default)**   | **21.84**           | **-1.557**       | **29.05**          | **13.19**  |
>
> As shown, using the L2 norm of the predicted noise results in the best (lowest) value for all metrics for both seen and unseen masks.
>
> ---
>
> **[Q3] Influence of the early step T value**
>
> Thank you for your constructive comment. As per your suggestion, we conducted additional ablation experiments by selecting T after splitting the timesteps into 5 equal intervals due to resource constraints. The results are presented in the Table B below.
>
> **Table B. Ablation of different early step values**
>
> | **Method**                              | **CLIP Dir. Sim ↓** | **CLIP Sim. ↓** | **ImageReward ↓** | **PSNR ↓** |
> |-----------------------------------------|----------------------|------------------|--------------------|------------|
> | **Unseen mask**                         |                      |                  |                    |            |
> | DiffusionGuard, step T/5                | 24.20               | -1.285           | 31.09              | 14.36      |
> | DiffusionGuard, step 2T/5               | 23.59               | -1.388           | 30.54              | 14.32      |
> | DiffusionGuard, step 3T/5               | 22.29               | -1.498           | 29.63              | 13.42      |
> | DiffusionGuard, step 4T/5               | 21.88               | -1.552           | 29.24              | 13.03      |
> | DiffusionGuard, step 5T/5               | 21.84               | -1.557           | 29.05              | 13.19      |
>
>
> We observed that T values around the first and second intervals yield the best protection strength, which we attribute to the significance of early denoising steps in inpainting models. This finding aligns with the motivation and approach of our method, which emphasizes early-stage (closer to pure Gaussian noise, i.e., T) loss.

---

> ### Author Response · Authors · 2024-11-21
>
> **[Q4] Ablation of learning rate choice**
>
> Thank you for your thoughtful comment. We agree that identifying the best hyperparameters for DiffusionGuard is important for its effectiveness. Therefore, we conducted additional experiments using various learning rate (PGD step size) values, and the results are presented below (Table C).
>
> **Table C. Learning rate (step size) hyperparemeter search of DiffusionGuard**
>
> | **Method**                              | **CLIP Dir. Sim ↓** | **CLIP Sim. ↓** | **ImageReward ↓** | **PSNR ↓** |
> |-----------------------------------------|----------------------|------------------|--------------------|------------|
> | **Seen mask**                           |                      |                  |                    |            |
> | DiffusionGuard, lr=0.5                  | 19.21               | -1.777           | 26.80              | **12.52**  |
> | DiffusionGuard, lr=1.0                  | **18.95**           | -1.807           | **26.55**          | 12.60      |
> | DiffusionGuard, lr=2.0                  | 19.45               | **-1.816**       | 26.90              | 12.52  |
> | DiffusionGuard, lr=3.0                  | 20.10               | -1.724           | 27.29              | 12.71      |
> | DiffusionGuard, lr=4.0                  | 20.75               | -1.677           | 27.72              | 12.80      |
> | DiffusionGuard, lr=8.0                  | 22.23               | -1.541           | 28.79              | 13.47      |
> | **Unseen mask**                         |                      |                  |                    |            |
> | DiffusionGuard, lr=0.5                  | 22.00               | **-1.560**       | 29.23              | **13.13**  |
> | DiffusionGuard, lr=1.0                  | **21.84**           | -1.557           | **29.05**          | 13.19      |
> | DiffusionGuard, lr=2.0                  | 22.13               | -1.513           | 29.30              | 13.15      |
> | DiffusionGuard, lr=3.0                  | 22.18               | -1.481           | 29.28              | 13.26      |
> | DiffusionGuard, lr=4.0                  | 22.59               | -1.463           | 29.54              | 13.39      |
> | DiffusionGuard, lr=8.0                  | 23.06               | -1.372           | 30.01              | 14.03      |
>
> As shown, we find that the learning rate of 0.5/255 or 1.0/255 yields the best performance, and the performance gets worse as the learning rate increases. The hyperparameter search suggests that DiffusionGuard performs better when the learning rate is smaller, which is better aligned with the general intuition of optimization, unlike some previous paper which stated that larger learning rates may cause better performance.
>
> ---
>
> **[Q5-1] When malicious user edits only the eyes in a face photo**
>
> To address the reviewer's concern, we conducted additional demonstration experiments in which only certain facial features are inpainted (e.g., shape of pupil, line of sight, shape of mouth), by crafting new masks that correspond to these new setups. We include the qualitative demo in Figure 24 of Appendix J, in which DiffusionGuard still causes the editing to fail, maintaining its protective effectiveness.
>
> We believe this is because the noise is applied all over the face, most of the noise survives even when certain facial features are inpainted, resulting in a failed edit.

---

> ### Author Response · Authors · 2024-11-21
>
> **[Q5-2] Effectiveness of dividing perturbation into sub-perturbations**
>
> This is a very intriguing idea! As suggested, we conducted a series of experiments and detailed them in Appendix J.1 of our updated draft.
>
> As shown in Figure 25 of Appendix J.1, we divided a face in a given image into three subparts: (1) eyes and forehead, (2) mouth and cheeks, and (3) neck. Correspondingly, we split the original mask into three masks, each targeting a subpart. Using DiffusionGuard, we generated adversarial perturbations for each subpart, then merged them into a single perturbation matching the original mask's shape.
>
> We compared this merged perturbation with the original DiffusionGuard perturbation under four scenarios: editing the eyes, mouth, neck, and the entire image. The quantitative results are presented in Table D below. Interestingly, the merged perturbation performed similarly to the original for localized edits and slightly better for full-image protection.
>
> **Table D. Sub-perturbation quantitative results**
>
> | **Method**                              | **CLIP Dir. Sim ↓** | **CLIP Sim. ↓** | **ImageReward ↓** | **PSNR ↓** |
> |-----------------------------------------|----------------------|------------------|--------------------|------------|
> | **Mask: Eye**                           |                      |                  |                    |            |
> | Default DiffusionGuard                  | 1.48                | **20.28**        | **-1.503**         | **33.45**  |
> | Sub-perturbation DiffusionGuard         | **0.67**            | 20.31            | -1.442             | 34.45      |
> | **Mask: Mouth**                         |                      |                  |                    |            |
> | Default DiffusionGuard                  | 8.03                | 17.91            | -1.363             | 30.71      |
> | Sub-perturbation DiffusionGuard         | **7.95**            | **17.82**        | **-1.400**         | **31.73**  |
> | **Mask: Neck**                          |                      |                  |                    |            |
> | Default DiffusionGuard                  | **14.12**           | **20.01**        | **-0.269**         | **30.63**  |
> | Sub-perturbation DiffusionGuard         | 16.88               | 20.48            | 0.071              | 31.71      |
> | **Mask: Entire face**                   |                      |                  |                    |            |
> | Default DiffusionGuard                  | 2.34                | 19.51            | -1.98              | 37.47      |
> | Sub-perturbation DiffusionGuard         | **2.12**            | **19.40**        | **-1.99**          | **34.18**  |
>
> While the idea of generating sub-perturbation has great potential for development, it did not consistently enhance protection in this experiment, likely because the protective noise over targeted areas is not inputted to the inpainting model. Despite this, the approach shows promise, especially when inpainting the background, instead of inside the face. For instance, if a malicious user regenerated everywhere except the eyes and the nose, this idea could be more effective than default perturbations.
>
> Thank you again for suggesting this fascinating idea about sub-perturbations, and we will continue to explore any possible directions to enhance its effectiveness.
>
> ---
>
> **[Q6] Figure 7a too small**
>
> Thank you for your careful review. As per your suggestion, we have updated Figure 7a to make it wider, improving its visualization. This change has been incorporated into the updated draft for your convenience.

---

> ### Comment · Reviewer_WYYe · 2024-11-21
> **request more clarification of the Re-Q5-1 and maybe some explanation of Q2**
>
> ## Q1
> I was confused about that XD. Now this concern is solved.
>
> ## Q2
> Thanks for tons of experiments. Choosing $L_2$ due to better performance, that make sense, but more explaination or analysis will benefit. For example, I'm not sure if it's correct, maybe the $L_2$ is related to the reconstruction loss or something else, so it have better performance? I'm expecting a convincing explanation.
>
> ## Q3+Q4
> This solved my concern about the ablation study, which will provide a basic understanding of different parameters.
>
> ## Q5-1
> Just for double check, could you clarify the adversarial sample was generated with the previous mask (main object) or the new mask (eye)?
>
> ## 5-2
> Very interesting findings. I'm surprised that the merged perturbation is a little bit better than the default one. I think this could be a potential research direction that atomize the image into several sections, then optimize for them. A good speed up strategy such as optimize all blocks at the same time will be a huge innivation.

---

> ### Author Response · Authors · 2024-11-22
>
> Dear reviewer WYYe,
>
> Thank you for your thoughtful feedback and insightful questions. We deeply appreciate the time you’ve taken to engage with our work and offer such valuable perspectives. Below, we address each of your points in detail:
>
> **[R1] Question about L2 loss**
>
> Thank you for the question. We agree with your insight that L2 leads to better performance because diffusion models are trained for the MSE loss (e.g., between target noise and its prediction). More specifically, this training would effectively define the likelihood of the diffusion model with MSE as well; for instance, [1] defines the conditional log-likelihood using the L2 distance of the input image and a diffusion-denoised image.
>
> Therefore, attacking the diffusion model using the training loss would be the most effective, and other losses could be less effective. For example, an adversarial perturbation generated to maximize the L1 loss might not maximize the L2 loss. If a model trained using the L2 loss receives such input, the model would process it more accurately than when it receives an adversarial perturbation generated using the L2 loss.
>
> **[R2] Resolved concern about ablation study**
>
> We are delighted that our additional ablation studies addressed your concern and provided clarity on different parameters. Thank you for your thoughtful feedback and encouraging words.
>
> **[R3] Clarification about adversarial perturbation for eye editing experiments**
>
> To clarify, the adversarial perturbations for the example in [W5-1] (Figure 26) were generated using the previous mask (main object). In other words, the adversarial perturbation from our main experiment was directly reused, making this a transferred setup. We have updated our draft to reflect this clarification.
>
> **[R4] About sub-perturbations**
>
> We fully agree that atomizing an image into several sections and optimizing for each individually presents a promising research direction. One interesting application of this approach could be for images containing multiple independent entities to protect—for example, an image with multiple people. In such cases, generating adversarial perturbations for each face separately and then merging them might prove more effective than simply optimizing for all faces simultaneously.
>
> We deeply appreciate your insightful suggestion and are excited about the possibilities of future research direction. This discussion has been immensely thought-provoking and has further motivated us to explore these ideas in future work.
>
> [1] Jaini, P., Clark, K., and Geirhos, R., 2024. Intriguing properties of generative classifiers. International Conference on Learning Representations, 2024.

---

> ### Comment · Reviewer_WYYe · 2024-11-22
>
> Thank author for thorough response during the rebuttal period. I am pleased to see that all my previous concerns have been adequately addressed, and I have accordingly increased my rating to 6.
>
> While I find this paper to be methodologically sound and well-executed, I should note that although I have experience with `attack against LDM` and `LDM-based defenses`, I am not specifically an expert in inpainting. Nevertheless, my first impression of attacking inpainting is, there could be a solution that `atomize image into minimum semantic sections` as I mentioned previously, which might lead to more robust attack methods against inpainting. This consideration somewhat tempers my enthusiasm for the novelty of the approach, preventing me from assigning a higher score of 8 or 10.
>
> That said, I want to commend the experimental section, which comprehensively addresses all my concerns. The authors' diligent efforts during the rebuttal period have successfully clarified the vast majority of my questions. The thorough responses and additional analyses have significantly strengthened my confidence in the work's validity and contribution to the field.

---

> > ### Author Response · Authors · 2024-11-23
> >
> > Dear Reviewer WYYe,
> >
> > We are very delighted that our comments and experiments were able to adequately address your previous concerns, and thank you for the constructive comments and for increasing the score.
> >
> > We agree that atomizing an image into minimum semantic sections could indeed lead to a more robust solution against inpainting, especially under various mask inputs. This is indeed an interesting direction for future research, and we appreciate you bringing it to our attention. While our current approach focuses on exploiting the specific behavior of inpainting models and augmenting a mask and to design effective and robust attacks, incorporating information about semantic subparts in an image could further enhance robustness against various masks.
> >
> > We are also grateful for your recognition of the experimental section and our efforts to provide comprehensive analyses. Your feedback has been invaluable in refining our work. Thank you again for your thoughtful and encouraging review.

---

### Official Review · Reviewer_ZPiB · 2024-10-24

**Soundness:** 4
**Presentation:** 4
**Contribution:** 3
**Rating:** 6
**Confidence:** 4

**Summary:**

This work reveals that inpainting models generate fine details in the very early stages of the denoising process, leading to the development of a defense method against unauthorized image inpainting. A mask augmentation technique is proposed to enhance robustness. Additionally, a benchmark is introduced to evaluate the effectiveness of protection against unauthorized image inpainting.

**Strengths:**

1. An insight is provided that inpainting models generate fine details during the very early stages of the denoising process.
2. A new objective specifically designed to prevent image inpainting is introduced.
3. A benchmark is introduced.

**Weaknesses:**

1. The mask augmentation is achieved by shrinking the contours inward. If malicious users provide masks larger than those used during training, will this affect performance?

2. The diffusion model's sampling can begin from different timesteps, and various sampling schedulers may start at different timesteps. For example, when sampling with 50 steps of DDIM, T is typically around 981, whereas for 25 steps of DPM-Solver, T might be around 961. If the user uses a different sampler from the one used during training, or the same sampler but starts from a different timestep T, will the proposed algorithm still work in this case?

3. The problem setting may be somewhat narrow. While the title suggests it is "against image editing," the method is only effective for a specific type of editing—image inpainting. It remains unclear whether the method can prevent other forms of editing that don't involve masks, such as instruction-guided editing [1][2].

4. Several recent references [3,4,5] on harmful concept removal are missing.

---
[1] InstructPix2Pix: Learning to Follow Image Editing Instructions

[2] MagicBrush: A Manually Annotated Dataset for Instruction-Guided Image Editing

[3] One-dimensional Adapter to Rule Them All: Concepts, Diffusion Models and Erasing Applications

[4] MACE: Mass Concept Erasure in Diffusion Models

[7] Separable Multi-Concept Erasure from Diffusion Models

**Questions:**

Please refer to the weaknesses.

If the authors address my concerns during the rebuttal, I would be open to adjusting my score.

---

> ### Author Response · Authors · 2024-11-21
>
> Dear Reviewer ZPiB,
>
> We sincerely appreciate your insightful comments. They were incredibly helpful in improving our draft. We have addressed each comment in detail below.
>
> ---
>
> **[W1] Effects of larger masks at test time**
>
> This is an interesting question! We conducted additional experiments to test this scenario. Specifically, we generated two larger masks by dilating the existing masks (one seen and one unseen), ensuring they are significantly larger, with an average increase of 26% in mask area. As shown in Table A (below), we found that DiffusionGuard maintains strong protective effectiveness compared to the baselines, even when a larger mask is provided at test time.
>
> **Table A. Using a larger mask at test-time**
>
> | Method           | CLIP Dir. Sim ↓ | CLIP Sim. ↓ | ImageReward ↓ | PSNR ↓  |
> |------------------|-----------------|-------------|---------------|---------|
> | PhotoGuard       | 22.07          | -1.588      | 28.55         | 15.45   |
> | AdvDM            | 21.76          | -1.593      | 28.46         | **13.20** |
> | Mist             | 22.19          | -1.562      | 28.64         | 13.99   |
> | SDS(-)           | 21.29          | -1.587      | 28.20         | 13.96   |
> | **DiffusionGuard** | **20.71**      | **-1.709**  | **27.86**     | 14.72   |
>
> We explain this in detail in Appendix E.5 and Table 8 in our updated draft.
>
> ---
>
> **[W2] Inference starting with different timesteps or using different sampler**
>
> Thank you for the intriguing question. To address this, we conducted two additional experiments: testing DiffusionGuard with DDIM using different steps, and testing with a different sampler (DPM-Solver) using 25 steps. Specifically, we tested DDIM with three additional inference steps (25, 40, 75), corresponding to T values of 961, 976, and 963, respectively. Additionally, we tested DPM-Solver with 25 steps. In all cases, we found that DiffusionGuard operates independently of inference steps and consistently outperforms baseline methods, as shown in Table B below. As shown in the table, DiffusionGuard consistently shows the best performance without being particularly sensitive to the number of sampling steps or the sampler type.
>
> **Table B. Evaluation using different number of DDIM inference steps or sampler**
>
> | **Method**                                      | **CLIP Dir. Sim↓** | **CLIP Sim.↓** | **ImageReward↓** | **PSNR↓** |
> |-------------------------------------------------|--------------------|----------------|------------------|-----------|
> | **Unseen mask**                                 |                    |                |                  |           |
> | PhotoGuard, DDIM 25 steps                       | 22.93              | -1.421         | 30.04            | 15.43     |
> | PhotoGuard, DDIM 40 steps                       | 23.21              | -1.357         | 30.23            | 14.69     |
> | PhotoGuard, DDIM 50 steps (default)             | 23.30              | -1.357         | 30.30            | 14.53     |
> | PhotoGuard, DDIM 75 steps                       | 23.53              | -1.368         | 30.31            | 14.59     |
> | PhotoGuard, DPM-Solver 25 steps                  | 18.75              | -1.682         | 26.73            | 11.83     |
> | AdvDM, DDIM 25 steps                             | 23.92              | -1.382         | 30.73            | 14.16     |
> | AdvDM, DDIM 40 steps                             | 24.27              | -1.358         | 30.94            | 13.52     |
> | AdvDM, DDIM 50 steps (default)                   | 24.27              | -1.361         | 30.97            | 13.37     |
> | AdvDM, DDIM 75 steps                             | 24.20              | -1.376         | 30.90            | 13.36     |
> | AdvDM, DPM-Solver 25 steps                        | 19.37              | -1.744         | 27.27            | 10.63     |
> | **DiffusionGuard, DDIM 25 steps**               | **21.29**          | **-1.596**     | **28.59**        | **14.04** |
> | **DiffusionGuard, DDIM 40 steps**               | **21.59**          | **-1.562**     | **28.90**        | **13.29** |
> | **DiffusionGuard, DDIM 50 steps (default)**     | **21.84**          | **-1.557**     | **29.05**        | **13.19** |
> | **DiffusionGuard, DDIM 75 steps**               | **22.08**          | **-1.572**     | **29.12**        | **13.29** |
> | **DiffusionGuard, DPM-Solver 25 steps**          | **17.48**          | **-1.816**     | **25.53**        | **10.20** |
>
> Note that while we presented two main baselines and unseen mask set only in this table due to space limit, DiffusionGuard outperformed all other baselines in a similar manner in both seen and unseen mask sets. We include the full detail of the experiments in Table 9 of Appendix E.6.

---

> > ### Comment · Reviewer_ZPiB · 2024-11-22
> > **About larger masks**
> >
> > Thanks for the detailed responses.
> >
> > About masks, could the authors provide some visual results for reference? For example, providing a figure containing several cases using DiffusionGuard with maybe 6-8 different sizes (some larger some smaller) of masks on the same case, to visually see the effect of different sizes of the mask. And to see if any insight can be provided for readers.

---

> ### Author Response · Authors · 2024-11-21
>
> **[W3] Somewhat narrow problem setting and evaluating against instruction-guided editing**
>
> Thank you for your constructive feedback. Our proposed method can be used with instruction-based models such as InstructPix2Pix as well, especially the early-stage loss component. Although mask augmentation cannot be used as instruction-based models do not accept any mask input, our loss can still be applied.
>
> Our focus on inpainting methods stems from their superior practical usefulness and flexibility in editing. While instruction-based methods like InstructPix2Pix can edit images without a mask, they tend to preserve high-level structures, such as body posture, which limits their capacity for drastic modifications (see Figure 23 in Appendix I for failure cases of InstructPix2Pix). In contrast, masked inpainting allows for the complete regeneration of designated areas. Consequently, we believe that inpainting-based editing methods, which are the focus of this paper, offer more practical value for complex scenarios than instruction-based (non-inpainting) editing methods.
>
> Furthermore, to strengthen our work, we have extended our evaluations to include instruction-based methods such as InstructPix2Pix. Our results show that DiffusionGuard provides superior protective effectiveness compared to existing methods when applied to InstructPix2Pix, as detailed in the table below.
>
> **Table C. Comparison using InstructPix2Pix**
>
> | **Method**         | **CLIP Dir. Sim↓** | **CLIP Sim.↓** | **ImageReward↓** | **PSNR↓**    |
> |---------------------|--------------------|----------------|------------------|--------------|
> | PhotoGuard          | 15.02             | -1.508         | 22.95           | 17.19        |
> | AdvDM               | 22.15             | -1.234         | 27.18           | 14.53        |
> | Mist                | 22.82             | -1.204         | 27.48           | 14.35        |
> | SDS(-)              | 25.21             | -1.290         | 29.34           | **11.50**    |
> | **DiffusionGuard**  | **14.07**         | **-1.591**     | **21.74**       | 17.42    |
>
> We have included these results in Table 7 of Appendix E.4 with more details.
>
> ---
>
> **[W4] Missing references on harmful concept removal**
>
> Thank you for the constructive feedback. We have updated the draft as per your suggestion, and included the recent references on harmful concept removal in Section 5 (Related works).

---

> > ### Comment · Reviewer_ZPiB · 2024-11-22
> > **About different sampling schedulers (or T)**
> >
> > Thanks for the detailed experiments.
> >
> > About different sampling schedulers (or T), could authors provide some discussion or rationale as to why this training method can be generalized to different samplers or T? If my understanding is not wrong, only one T is seen during training.

---

> > ### Comment · Reviewer_ZPiB · 2024-11-22
> > **About instruction-guided editing**
> >
> > About instruction-guided editing, I am satisfied with the results that DiffusionGuard can protect images from them.
> >
> > About failure cases of Instruct-pix2pix, if users start from a bad image and then input prompts like replace the face with xxx, where xxx is a famous person that sd can generate, can it be successful?
> >
> > Or, if users start from a photo of a famous person and then input a prompt like changing the background as in jail, can it be successful?

---

> ### Author Response · Authors · 2024-11-22
>
> Dear reviewer ZPiB,
>
> Thank you for your thoughtful feedback and insightful questions. We deeply appreciate the time you’ve taken to engage with our work and offer such valuable perspectives. We addressed each point in detail in the replies below.
>
> **[R1] Qualitative examples for edits using various masks**
>
> Thank you for a constructive comment. Based on your new suggestion, we added a new visualization in Figure 29 in Appendix K, including different masks and their corresponding edit results.
>
> We used a total of 7 masks: 4 smaller than the head (Rows 2–5), 1 matching the head size (Row 1), and 2 larger than the head (Rows 6–7). In all cases, DiffusionGuard results in a successful protection, causing the edit to result in unrealistic images.
>
> A notable insight is that masks larger than the head, which include regions outside the head (especially the background), restrict editing flexibility. In the figure, for these larger masks, the background remains fixed as an empty entrance to a dark hallway (Rows 6–7) to align with the dark surroundings of the source image, and the shirt color is consistently dark blue due to the original clothing. In contrast, the smaller masks (Rows 1–5) allow diverse backgrounds, such as a wall, hallway, or hospital ward. This suggests that larger masks, as they include the background, are less ideal for flexible editing and emphasize the need to focusing on the head, especially the face.
>
> Another observation is that for all 7 masks, including the smoother backgrounds of Rows 6–7, the boundary between the original face and the edited background is visibly distinct in the protected images. This likely results from the attack disrupting the inpainting model's internal processes, causing it to misinterpret colors. This effect could be useful for future research directions.

---

> ### Author Response · Authors · 2024-11-22
>
> **[R2] About using different timesteps or schedulers**
>
> We believe that optimizing for a single integer timestep (e.g., 1 to 1000) generalizes to nearby timesteps because timesteps are converted into real-number sinusoidal embeddings, similar to Transformer positional embeddings, as described in the DDPM [1] paper: "Diffusion time t is specified by adding the Transformer sinusoidal position embedding into each residual block."
> While diffusion model libraries like Diffusers represent timesteps as discrete integers (e.g., `981`), this is arbitrary, as timesteps exist on a continuous spectrum. Optimizing for one timestep influencing nearby ones is reasonable, given the common practices like interpolating positional embeddings (e.g., Vision Transformer [2]) or adjusting the number of timesteps dynamically (e.g., Consistency Models [3]).
>
> Another example can be found in research about protection against diffusion models as well. In AdvDM [4], adversarial perturbations are generated by randomly sampling timesteps uniformly from all of {1, 2, ..., 1000}, rather than specific values like {21, 41, ..., 981}.
>
> [1] Ho, J., Jain, A., and Abbeel, P., 2020. Denoising diffusion probabilistic models. Advances in Neural Information Processing Systems, 2020.
>
> [2] Dosovitskiy, A., Beyer, L., Kolesnikov, A., et al., 2021. An image is worth 16x16 words: Transformers for image recognition at scale. International Conference on Learning Representations, 2021.
>
> [3] Song, Y., Dhariwal, P., Chen, M., et al., 2023. Consistency models. International Conference on Machine Learning, 2023.
>
> [4] Liang, C., Wu, X., Hua, Y., et al., 2023. Adversarial example does good: Preventing painting imitation from diffusion models via adversarial examples. International Conference on Machine Learning, 2023.

---

> ### Author Response · Authors · 2024-11-22
>
> **[R3] Additional experiments with InstructPix2Pix**
>
> Thank you for the detailed response. Based on your feedback, we conducted two editing experiments with InstructPix2Pix, one by starting from a malicious image and changing the face, and one by starting with a celebrity image and changing the background.
> - **Starting with malicious image and changing the face to celebrity (Figure 24 of Appendix I.1):**
>   We started with a person being arrested and instructed InstructPix2Pix and Stable Diffusion Inpainting to replace the subject's face with that of a celebrity. Using the prompt "Turn the man into {celebrity name}", we edited 10 images of well-known figures recognized by Stable Diffusion Inpainting. As a control, we applied the same prompt to an inpainting model with a mask designating the face area.
>
>   The results reveal two key limitations of InstructPix2Pix: (1) the instruction affects unintended areas, such the police officer in the image, always turning them into the celebrity in all cases, or even the background, and (2) the generated face quality is lower than that of the inpainting model. For highly conditioned celebrities like football players (e.g., Lionel Messi, Cristiano Ronaldo), even the background changes to match their context, such as resembling a football stadium.
>
>
> - **Starting with celebrity image and changing the background (Figure 25 of Appendix I.2):**
>   Starting with celebrity images, we used InstructPix2Pix with the instruction "Change the background to jail" and Stable Diffusion Inpainting with the prompt "A photo of a person in jail". For inpainting, we used a mask designating the face region.
>
>   As shown in Figure 25, InstructPix2Pix struggles with this task, often altering the celebrity into a different person (Rows 1, 3, 4, 6) or over-conditioning on the source image such as the postures or letters inside them (Rows 2, 5, 6). In contrast, the inpainting model successfully generates accurate representations of celebrities in a jail setting in all cases.

---

> > ### Comment · Reviewer_ZPiB · 2024-11-22
> >
> > I appreciate the authors' patient explanations and thorough experiments. The results regarding large masks are both intriguing and reasonable, the discussion on generalization is well-articulated, and the insights into instruct-pix2pix are thoughtful and commendable.
> >
> > All my concerns have been effectively addressed, and I am pleased to raise the score to 6.

---

> > > ### Author Response · Authors · 2024-11-22
> > >
> > > Dear Reviewer ZPiB,
> > >
> > > Thank you for your thoughtful feedback and for raising the score. We are delightful that our additional experiments and explanations were able to engage with the thoughtful ideas you raised. Your insights and comments were invaluable in improving our work, and we greatly appreciate the constructive discussion.

---

### Official Review · Reviewer_QU4Q · 2024-11-03

**Soundness:** 3
**Presentation:** 3
**Contribution:** 3
**Rating:** 6
**Confidence:** 4

**Summary:**

The paper proposes a effective and robust method against malicious diffusion-based image editing. The method is interesting and insightful. With the proposed benchmark, the paper shows the superior results compared to baseline methods.

**Strengths:**

1. The observations, that the inpainting models produce fine details of masked region at early steps, are interesting and insightful.

2. Using augmented masks is a reasonable and effective method to improve robustness.

3. The paper proposes a benchmark to evaluate different methods. Extensive results show the effectiveness and robustness of the method.

**Weaknesses:**

There are two main concerns.

1. Did the authors try some specifically designed purification methods for such perturbations in diffusion models? Such as the method in [1].

2. Only focusing on mask-based image editing may be a little limited. Currently many editing methods do not require such masks, such as InstructPix2Pix[2]. Can the proposed method be used in these methods? Will the proposed method still be more effective and robust?




[1] Bochuan Cao et al. IMPRESS: Evaluating the Resilience of Imperceptible Perturbations Against Unauthorized Data Usage in Diffusion-Based Generative AI, 2023.

[2] Tim Brooks et al. InstructPix2Pix: Learning to Follow Image Editing Instructions, 2023.

**Questions:**

1. Will different editing prompts have effects on the results? Are the perturbations generated with a single prompt or several different text prompts?

For other questions, please see the Weaknesses part.

---

> ### Author Response · Authors · 2024-11-21
>
> Dear Reviewer QU4Q,
>
> We sincerely appreciate your insightful comments. They were incredibly helpful in improving our draft. We have addressed each comment in detail below.
>
> ---
>
> **[W1] Testing against purification methods specifically designed for diffusion models**
>
> We appreciate your constructive comment. Following your suggestion, we conducted additional evaluations of DiffusionGuard and the baseline models against IMPRESS, a purification method specifically designed for defense against diffusion models. The results, as detailed in Table A below, demonstrate that DiffusionGuard remains more resilient compared to the baseline methods, achieving best (bold) or second-to-best (underline) results in all metrics. This shows the robustness of DiffusionGuard against targeted purification strategies.
>
> **Table A. Evaluation after purifying each protection method with IMPRESS**
>
> | **Method**              | **CLIP Dir. Sim ↓** | **CLIP Sim. ↓**  | **ImageReward ↓** | **PSNR ↓**   |
> |--------------------------|---------------------|-------------------|--------------------|--------------|
> | **Seen mask**           |                     |                   |                    |              |
> | PhotoGuard              | *23.45*            | -1.387            | *29.84*           | 14.28        |
> | AdvDM                   | 23.57              | *-1.458*          | 30.04             | **13.57**    |
> | Mist                    | 23.69              | -1.417            | 30.07             | 14.14        |
> | SDS(-)                  | 23.49              | -1.334            | 29.94             | 14.46        |
> | **DiffusionGuard**      | **22.37**          | **-1.500**        | **29.10**         | *13.78*      |
> | **Unseen mask**         |                     |                   |                    |              |
> | PhotoGuard              | 23.33              | -1.348            | *29.94*           | 14.58        |
> | AdvDM                   | 23.28              | **-1.442**        | 30.00             | **13.55**    |
> | Mist                    | *23.24*            | -1.407            | 30.09             | 14.08        |
> | SDS(-)                  | 23.87              | -1.231            | 30.80             | 14.32        |
> | **DiffusionGuard**      | **22.81**          | *-1.418*          | **29.64**         | *13.92*      |
>
> ---
>
>
> **[W2] DiffusionGuard against InstructPix2Pix**
>
> Thank you for your constructive feedback. Our proposed method can be used with instruction-based models such as InstructPix2Pix as well, especially the early-stage loss component. Although mask augmentation cannot be used as instruction-based models do not accept any mask input, our loss can still be applied.
>
> Our focus on inpainting methods stems from their superior practical usefulness and flexibility in editing. While instruction-based methods like InstructPix2Pix can edit images without a mask, they tend to preserve high-level structures, such as body posture, which limits their capacity for drastic modifications (see Figure 23 in Appendix I for failure cases of InstructPix2Pix). In contrast, masked inpainting allows for the complete regeneration of designated areas. Consequently, we believe that inpainting-based editing methods, which are the focus of this paper, offer more practical value for complex scenarios than instruction-based (non-inpainting) editing methods.
>
> Furthermore, to strengthen our work, we have extended our evaluations to include instruction-based methods such as InstructPix2Pix. Our results show that DiffusionGuard provides superior protective effectiveness compared to existing methods when applied to InstructPix2Pix, as detailed in Table B below.
>
> **Table B. Comparison using InstructPix2Pix**
>
> | **Method**              | **CLIP Dir. Sim ↓** | **CLIP Sim. ↓**  | **ImageReward ↓** | **PSNR ↓**   |
> |--------------------------|---------------------|-------------------|--------------------|--------------|
> | PhotoGuard              | 15.02              | -1.508            | 22.95             | 17.19        |
> | AdvDM                   | 22.15              | -1.234            | 27.18             | 14.53        |
> | Mist                    | 22.82              | -1.204            | 27.48             | 14.35        |
> | SDS(-)                  | 25.21              | -1.290            | 29.34             | **11.50**    |
> | **DiffusionGuard**      | **14.07**          | **-1.591**        | **21.74**         | 17.42        |
>
> We have included these results in Table 7 of Appendix E.4 with more details.

---

> ### Author Response · Authors · 2024-11-21
>
> **[Q1] Effect of editing prompts on the generation of perturbations**
>
> Thank you for this interesting question. In our experiments, we generated perturbations by setting the text prompt to an empty string ("" in Python) to maintain neutrality and ensure that they are not biased towards any specific test-time prompts, following PhotoGuard [1].
>
> Regarding the effects of different prompts, our observations indicate that perturbations generated with a non-empty text prompt are more tailored to that specific prompt. This makes them more effective at protecting against similar prompts due to their targeted nature but less effective against prompts that are unrelated.
>
> We have clarified this in Appendix D (Experimental details) in our updated draft.
>
> [1] Salman, H., Khaddaj, A., Leclerc, G., Ilyas, A., and Madry, A., 2023. Raising the cost of malicious AI-powered image editing. International Conference on Machine Learning, 2023.

---

> ### Author Response · Authors · 2024-11-26
>
> Dear Reviewer QU4Q,
>
> We greatly appreciate the time and efforts in reviewing our paper.
>
> As the discussion period draws close, we kindly remind you that seven days remain for further comments or questions. We would appreciate the opportunity to address any additional concerns you may have before the discussion phase ends.
>
> Thank you very much!
>
> Many thanks,
>
> Authors

---

### Official Review · Reviewer_ScsB · 2024-11-03

**Soundness:** 2
**Presentation:** 3
**Contribution:** 2
**Rating:** 6
**Confidence:** 2

**Summary:**

The paper proposes DiffusionGuard, an image-cloaking algorithm to defend against malicious diffusion-based text-guided inpainting. Compared to previous works, it has two main proposals. First, instead of optimizing any denoising step using either image-space loss or reconstruction loss, DiffusionGuard only optimizes at the early stage (t = T) and aims to increase the norm of the noise. Second, it employs mask augmentation to improve the robustness of the proposed algorithm for different mask variations at test time. Experiments verified that DiffusionGuard outperforms the previous baselines on this task.

**Strengths:**

- The proposal of mask augmentation is sensible.
- DiffusionGuard outperforms the baselines in all metrics. Qualitative figures show that it often causes the inpainting models to generate plain and blurry inpainted output backgrounds.

**Weaknesses:**

- The title is misleading. The work only focuses on diffusion-based text-guided inpainting, e.g., Stable Diffusion Inpainting. It does not consider other diffusion-based image editing methods such as Instruct-Pix2Pix, MasaCtrl... The authors should revise the title to better specify the scope of the work.
- The work only tests with Stable Diffusion Inpainting variants. Recent inpainting models, e.g., MagicBrush [1], should be mentioned and tested.
- L191-200: the mentioned "unique behavior" of inpainting models sounds misleading. In the early denoising stage, the fine details only appear on the unchanged region, which is basically copied from the input. The inpainting regions, i.e., the background, still do not have fine details and behave as in normal diffusion models.
- L200: The reason for targeting the early steps is not convincing. From the presented results, the proposed method affects only the inpainting regions outside of the face, which have similar behavior as in normal diffusion models. In the ablation studies, the author should add an extra experiment to test the case when Eq (4) is applied in all time steps instead of only in the early one.
- In mask augmentation, the mask is shrunk to be smaller. What happens if the mask used at test time is bigger?
- The PSNR metric used in Table 1 and Fig. 5 is not reliable. Given the same image and mask, we can have different editing results that match the input prompt. Hence, a small PSNR does not necessarily imply a successful defense; good editing can still produce a low PSNR score.
- The test set is small, with only 42 images. It is better to test on a much larger set of images.
- The authors ran experiments with 5 masks per testing image. From Fig.4, the masks are pretty similar; hence, the effect of changing the mask is not significant. I would trade the number of masks and prompts to have more testing images.
- The authors should provide a qualitative figure showcasing the cloaked images to see whether the added noise is obvious or not. Quantitative numbers (PSNR, SSIM) for it are also recommended.
- Fig.4: The first 3 examples are very good; the inpainted backgrounds are plain and blurry. However, the last example does not show that behavior. The authors should explain why. Fig.5a confirms that DiffusionGuard is not always that good and still loses to Photoguard 22-25% of the time.

[1]. Zhang, K., Mo, L., Chen, W., Sun, H. and Su, Y., 2024. Magicbrush: A manually annotated dataset for instruction-guided image editing. Advances in Neural Information Processing Systems, 36.

**Questions:**

See weaknesses.

---

> ### Author Response · Authors · 2024-11-21
>
> Dear Reviewer ScsB,
>
> We sincerely appreciate your insightful comments. They were incredibly helpful in improving our draft. We have addressed each comment in detail below.
>
> ---
>
> **[W1] Misleading title and lack of comparison to other editing methods (e.g., InstructPix2Pix)**
>
> Thank you for your constructive feedback. We acknowledge that the title may not fully reflect the scope of our work. We plan to revise it to "DiffusionGuard: A Robust Defense Against Malicious Diffusion-based **Inpainting**”, which addresses your concern more accurately.
>
> Our focus on inpainting methods stems from their superior practical usefulness and flexibility in editing. While instruction-based methods can edit images without a mask, they tend to preserve high-level structures, such as body posture, which limits their capacity for drastic modifications (see Figure 23 in Appendix I for failure cases of InstructPix2Pix). In contrast, masked inpainting allows for the complete regeneration of designated areas. Consequently, we believe that inpainting-based editing methods, which are the focus of this paper, offer more practical value for complex scenarios than instruction-based (non-inpainting) editing methods.
>
> Furthermore, to strengthen our work, we have extended our evaluations to include instruction-based methods such as InstructPix2Pix. Our results show that DiffusionGuard provides superior protective effectiveness compared to existing methods when applied to InstructPix2Pix, as shown in Table A below.
>
> **Table A. Comparison using InstructPix2Pix**
>
> | Method           | CLIP Dir. Sim ↓ | CLIP Sim. ↓ | ImageReward ↓ | PSNR ↓  |
> |------------------|-----------------|-------------|---------------|---------|
> | PhotoGuard       | 15.02          | -1.508      | 22.95         | 17.19   |
> | AdvDM            | 22.15          | -1.234      | 27.18         | 14.53   |
> | Mist             | 22.82          | -1.204      | 27.48         | 14.35   |
> | SDS(-)           | 25.21          | -1.290      | 29.34         | **11.50** |
> | **DiffusionGuard** | **14.07**      | **-1.591**  | **21.74**     | 17.42   |
>
>
> We have included these results in Table 7 of Appendix E.4 with more details.
>
> ---
>
> **[W2] Lack of comparison to MagicBrush**
>
> Thank you for your comment. However, it is important to note that MagicBrush does not propose a new inpainting model but rather focuses on instruction-based editing without requiring an inpainting mask. This is explicitly stated by the authors: *"We fine-tune InstructPix2Pix on MagicBrush and show that…"* Although the dataset in the MagicBrush paper includes a mask for each edit instance, the authors clarify that no inpainting-specific models are trained using this dataset.
>
> Our work evaluates defense against inpainting using Stable Diffusion (SD) Inpainting 1.0 and 2.0, as they are the most widely used inpainting models. Most related studies also test primarily on these two models, as very few other public inpainting models are available.
>
> ---
>
> **[W3] Misleading description of the unique behavior of inpainting models**
>
> To clarify, both the inside and outside of the mask in Fig. 2 are generated solely by the denoiser, starting from pure Gaussian noise, without any external copy-and-paste operation. Starting from pure Gaussian noise, the denoiser of the inpainting model is able to sharply denoise the masked region in the early steps by observing the source image which is given as an additional input. In contrast, a normal diffusion model with the same inputs does not exhibit this behavior. This indicates that this copy-and-paste-like behavior of inpainting models is a learned behavior during their fine-tuning process and we refer to this as a "unique behavior".
>
> As you noted, the rest of the image is generated similarly to normal diffusion models, lacking fine details in the early stages. This observation inspired our early-stage loss, based on the hypothesis that the early completion of the masked region may influence the generation of the rest of the image.

---

> ### Author Response · Authors · 2024-11-21
>
> **[W4] Reason for targeting early steps not sufficiently convincing**
>
> As clarified in [W3], the model does generate the inside of the face as well, and not just the outside, at the very early denoising step. This distinct behavior from normal diffusion models motivated targeting the early denoising stages.
>
> Following the common practice done by the baselines (PhotoGuard, AdvDM, Mist, SDS), we post-processed the generated results solely for demonstration purposes by copying and pasting the inside of the mask from the source image. This post-processing occluded the raw generated image and made the inside of the masks appear clean. We apologize for the confusion, and we have updated our draft to clearly state this. We also included raw generated images without post-processing in Figure 11 of Appendix D.
>
> Additionally, we also conducted additional experiments following your suggestion, by applying Eq. 4 to multiple timesteps at the same time. Specifically, we applied the loss at timesteps $\textbraceleft\frac{T}{10}, \frac{2T}{10}, ..., T\textbraceright$ or $\textbraceleft\frac{T}{5}, \frac{2T}{5}, ..., T\textbraceright$ simultaneously. As shown in Table B (below), the performance degraded when not focusing solely on the early timestep, supporting our claims.
>
> **Table B. Ablation of applying our loss over multiple steps at the same time**
>
> | Method                                      | CLIP Dir. Sim ↓ | CLIP Sim. ↓ | ImageReward ↓ | PSNR ↓  |
> |--------------------------------------------|-----------------|-------------|---------------|---------|
> | **Seen mask**                              |                 |             |               |         |
> | DiffusionGuard, {T/10, 2T/10, ..., T}      | 23.02           | -1.530      | 29.32         | 14.15   |
> | DiffusionGuard, {T/5, 2T/5, ..., T}        | 23.76           | -1.454      | 30.00         | 14.97   |
> | **DiffusionGuard (single early step)**     | **18.95**       | **-1.807**  | **26.55**     | **12.60** |
> | **Unseen mask**                            |                 |             |               |         |
> | DiffusionGuard, {T/10, 2T/10, ..., T}      | 23.36           | -1.379      | 30.25         | 14.27   |
> | DiffusionGuard, {T/5, 2T/5, ..., T}        | 24.07           | -1.319      | 30.74         | 15.51   |
> | **DiffusionGuard (single early step)**     | **21.84**       | **-1.557**  | **29.05**     | **13.19** |
>
> ---
>
> **[W5] Effects of bigger masks at test time**
>
> This is an interesting question! We conducted additional experiments to test this scenario. Specifically, we generated two larger masks by dilating the existing masks (one seen and one unseen), ensuring they are significantly larger, with an average increase of 26% in mask area. As shown in Table C below, DiffusionGuard continues to demonstrate strong protective effectiveness, outperforming the baselines even with larger test masks.
>
> **Table C. Using a larger mask at test-time**
>
> | Method           | CLIP Dir. Sim ↓ | CLIP Sim. ↓ | ImageReward ↓ | PSNR ↓  |
> |------------------|-----------------|-------------|---------------|---------|
> | PhotoGuard       | 22.07          | -1.588      | 28.55         | 15.45   |
> | AdvDM            | 21.76          | -1.593      | 28.46         | **13.20** |
> | Mist             | 22.19          | -1.562      | 28.64         | 13.99   |
> | SDS(-)           | 21.29          | -1.587      | 28.20         | 13.96   |
> | **DiffusionGuard** | **20.71**      | **-1.709**  | **27.86**     | 14.72   |
>
> ---
>
> **[W6] Reliability issue of PSNR**
>
> We agree that PSNR may not fully capture a successful defense. Still, we have incorporated three additional metrics that more accurately reflect the effectiveness of our defense strategy: CLIP similarity, CLIP directional similarity, and ImageReward. These metrics assess how well the edited images align with the editing instructions and the overall editing performance, respectively.
>
> Following your suggestion, we also added a new chart (Figure 14 of Appendix E.2.2) by replacing PSNR with CLIP directional similarity in Figure 5b, 5c. The results are still aligned with our findings: DiffusionGuard consistently outperforms PhotoGuard across different compute and noise budgets.

---

> ### Author Response · Authors · 2024-11-21
>
> **[W7,8] Suggestion to use a larger test set, and trade masks and prompts for more images**
>
> Thank you for your constructive feedback. While our original benchmark size aligns with most existing works in image protection against diffusion-based editing (42 images, each with 5 masks and 10 prompts), we agree that expanding the test set is important. For reference, SDS [1] employs a test set of 100 portrait images, each with 1 mask and 1 prompt, while PhotoGuard [2] tests on a smaller number of images but with up to 60 prompts each.
>
> In response, we collected 688 new images from the FFHQ [3] dataset, each paired with two human-validated, automatically generated masks (one seen, one unseen). This significantly expands the test set compared to existing works. Our evaluation on this larger dataset shows that DiffusionGuard outperforms baseline methods by a large margin. Quantitative results are provided in Table D below. For this experiment, we used all 10 prompts without reducing the number of prompts, totaling 13760 edited instances for each protection method.
>
> **Table D. Evaluation of each method on a new test set with 688 images**
>
> | Method           | CLIP Dir. Sim ↓ | CLIP Sim. ↓ | ImageReward ↓ | PSNR ↓  |
> |------------------|-----------------|-------------|---------------|---------|
> | **Seen mask**    |                 |             |               |         |
> | PhotoGuard       | 24.31          | -1.986      | 26.05         | 12.79   |
> | AdvDM            | 26.69          | -2.018      | 28.03         | 13.40   |
> | Mist             | 26.39          | -1.962      | 27.84         | 14.09   |
> | SDS(-)           | 25.93          | -1.939      | 27.64         | 14.31   |
> | **DiffusionGuard** | **22.48**      | **-2.176**  | **24.43**     | **12.70** |
> | **Unseen mask**  |                 |             |               |         |
> | PhotoGuard       | 26.28          | -1.948      | 27.88         | 14.19   |
> | AdvDM            | 27.22          | -1.967      | 28.78         | 13.24   |
> | Mist             | 26.78          | -1.898      | 28.39         | 14.00   |
> | SDS(-)           | 26.43          | -1.843      | 28.38         | 14.13   |
> | **DiffusionGuard** | **24.26**      | **-2.112**  | **26.22**     | **12.63** |
>
> We plan to collect more images to further extend our dataset and release it publicly soon. We have also included the new dataset in the supplementary materials, resized and compressed due to attachment size limits.
>
> [1] Xue, H., Liang, C., Wu, X., and Chen, Y., 2024. Toward effective protection against diffusion-based mimicry through score distillation. International Conference on Learning Representations, 2024.
>
> [2] Salman, H., Khaddaj, A., Leclerc, G., Ilyas, A., and Madry, A., 2023. Raising the cost of malicious AI-powered image editing. International Conference on Machine Learning, 2023.
>
> [3] Karras, T., Laine, S., and Aila, T., 2019. A style-based generator architecture for generative adversarial networks. Conference on Computer Vision and Pattern Recognition, 2019.

---

> > ### Comment · Reviewer_ScsB · 2024-11-21
> >
> > Thanks the authors for their rebuttal. It well addressed most of my concerns, so I have increased the score.

---

> > > ### Author Response · Authors · 2024-11-22
> > >
> > > Dear Reviewer ScsB,
> > >
> > > Thank you for your thoughtful feedback and for increasing the score. We appreciate your comments, which greatly helped improve the depth of our work. Your insights were invaluable, and we look forward to further refining our research.

---

### Author Response · Authors · 2024-11-21

Dear reviewers,

We sincerely appreciate the time and effort you have dedicated to reviewing our paper. Your insightful comments have been incredibly valuable in refining and improving our work.

As reviewers highlighted, our paper addresses an important ethical issue concerning diffusion-based image editing (ScsB, QU4Q, ZPiB, WYYe) and presents a novel method to protect images against inpainting models based on an interesting insight about inpainting models (QU4Q, ZPiB). This approach is validated through comprehensive evaluations (ScsB, QU4Q, ZPiB) and complemented by clear and detailed explanations (QU4Q, WYYe).

In response to your feedback, we have conducted additional experiments and incorporated the findings into the revised draft. Below is an outline of the key updates:

- **Evaluation of DiffusionGuard against instruction-based editing methods, including InstructPix2Pix** (Appendix E.4, Table 7; ScsB, QU4Q, ZPiB).
- **Comparison of effectiveness under larger test masks** (Appendix E.5, Table 8; ScsB, ZPiB).
- **Analysis of inference robustness across different samplers and timesteps** (Appendix E.6, Table 9; ZPiB).
- **Enhanced visualization and measurement of the visibility of the adversarial perturbations** (Appendix H, Figure 22; ScsB)
- **Additional references on recent studies about harmful concept removal** (Section 5; ZPiB)
- **Exploration of sub-perturbation approaches** (Appendix J.1, Figure 25; WYYe).

We believe that DiffusionGuard can be a useful addition to the ICLR community, guided by your feedback helping us enhance the clarity and depth of our work.

Thank you once again for your constructive reviews and support.

Authors.

---

### Meta-Review · Area_Chair_FnSA · 2024-12-21

**Metareview:**

This work introduces a novel adversarial noise-based defense method designed to protect images from unauthorized edits by diffusion-based image editing techniques. The authors propose an objective that targets the early stages of the diffusion process to enhance attack performance, complemented by a mask-augmentation technique that further improves robustness. The reviewers unanimously support the paper's acceptance in positive ratings (i.e., 6.0 on average), recognizing its contribution to addressing the challenges of diffusion-based image manipulation with an efficient and robust protection method. They also commend the innovative approach of crafting adversarial noise by focusing on the early denoising stages. Additionally, reviewers WYYe, QU4Q, and ScsB highlight the effectiveness of the mask augmentation method in enhancing robustness during test time.

Based on these positive evaluations, we have decided to accept the paper.

**Additional Comments On Reviewer Discussion:**

During the discussion periods, the reviewers request the thorough evaluations (ScsB, QU4Q) and detailed experiments for new insights in the aspects of enhancements in protection effectiveness, robustness, and efficiency (ZPiB, WYYe). The authors do properly address these concerns and provide the results of requested evaluations, including the more experiments with instruction-based editing  (ScSB, QU4Q), on a more extensive test set (ScsB), for the robustness evaluation against diffusion-based purification (QU4Q), and with different hyperparameters (WYYe, ZPiB).

---

### Decision · Program_Chairs · 2025-01-22

Accept (Poster)